# Secondary reactions of aromatics-derived oxygenated organic molecules lead to plentiful highly oxygenated organic molecules within an intraday OH exposure

Yuwei Wang[1], Chuang Li[1], Ying Zhang[1], Yueyang Li[1], Gan Yang[1], Xueyan Yang[1], Yizhen Wu[1], Lei Yao[1,2], Hefeng, Zhang[3*], Lin Wang[1,2,4,5,6]*

[1] Shanghai Key Laboratory of Atmospheric Particle Pollution and Prevention (LAP[3]), Department of Environmental Science and Engineering, Jiangwan Campus, Fudan University, Shanghai 200438, China

[2] Shanghai Institute of Pollution Control and Ecological Security, Shanghai 200092, China

[3] State Environmental Protection Key Laboratory of Vehicle Emission Control and Simulation, Vehicle Emission Control Center of Ministry of Ecology and Environment, Chinese Research Academy of Environmental Sciences, Beijing 100012, China

[4] IRDR International Center of Excellence on Risk Interconnectivity and Governance on Weather/Climate Extremes Impact and Public Health, Fudan University

[5] National Observations and Research Station for Wetland Ecosystems of the Yangtze Estuary, Shanghai, China

[6] Collaborative Innovation Center of Climate Change, Nanjing, 210023, China

*Corresponding Author: H.Z., email, zhanghf@craes.org.cn; phone, +86-10-84915586*

*L.W., email, lin_wang@fudan.edu.cn; phone, +86-21-31243568*

**ABSTRACT.** Highly oxygenated organic molecules (HOMs) can participate in new particle formation (NPF) and enhance growth of newly formed particles partially because of their low volatility. Previous studies have shown formation of HOMs via autoxidation reactions of $RO_2$ intermediates generated by OH-initiated oxidation of anthropogenic volatile organic compounds (VOCs). It was also suggested that multi-generation OH oxidation could be an important source for aromatics-derived HOMs. However, our understanding on the generation of aromatics-derived HOMs are still insufficient, especially for their formation mechanisms, which determine molar yields of HOMs and are essential to the establishment of global chemical box models related to HOMs. In this study, with a potential aerosol mass oxidation flow reactor (PAM OFR), two series of OH-initiated oxidation experiments of 1,3,5-trimethylbenzene (1,3,5-TMB) were conducted to investigate the formation of aromatics-derived HOMs. In the first series, the evolution of oxidation products of 1,3,5-TMB in an OH exposure range of $(0.5 – 5.0)\times10^{10}$ molecules $cm^{-3}$ s, equivalent to an OH exposure of $0.7 – 6.9$ hours at an OH concentration ([OH]) of $2\times10^6$ molecules $cm^{-3}$, was investigated by a nitrate-based chemical ionization mass spectrometer and a Vocus proton-transfer-reaction mass spectrometer, indicating significant secondary OH chemistry during the ageing of stabilized

first generation oxygenated products within an intraday OH exposure and formation of various HOMs with lower double bond equivalence (DBE). In addition, organonitrates, formed after the introduction of $NO_x$ into the reaction systems, further confirmed the existence of such secondary reactions. The second series of experiments was conducted with same residence time but much lower [OH], which also shows the generation of multi-generation HOMs with an [OH] as low as $1.06 \times 10^7$ molecules $cm^{-3}$ for 53 s, i.e., an OH exposure of around $5.86 \times 10^8$ molecules $cm^{-3}$ s. Our study suggests an important role of secondary OH chemistry in the oxidation of aromatics, if these oxygenated products survived long enough in the ambient, and elucidates detailed formation mechanisms of certain HOM products.

## 1 Introduction

OH radicals can react with volatile organic compounds (VOCs) in the atmosphere, converting primary pollutants to secondary ones. Generated from oxidation of VOCs, oxygenated organic molecules (OOMs) are crucial in a variety of atmospheric chemical processes, contributing efficiently to the formation of secondary organic aerosols (SOAs) and ground-level $O_3$ (Ng et al., 2010; Wang et al., 2022; Qu et al., 2021). Among the enormous number of oxygenated VOCs (OVOCs), highly oxygenated organic molecules (HOMs) have recently attracted significant attention (Bianchi et al., 2019). Most of HOMs are low volatility organic compounds (LVOCs) or extremely low volatility organic compounds (ELVOCs), and thus are able to drive the initial formation of nucleated particles under certain conditions and contribute to the subsequent growth of newly-formed particles, which finally enhance SOA formation (Tröstl et al., 2016; Lehtipalo et al., 2018; Stolzenburg et al., 2018; Mohr et al., 2019; Qiao et al., 2021).

Formation of HOMs is triggered by oxidation of VOCs in the gas phase. Peroxy radicals ($RO_2$) are generated at the initial step and will undergo an intramolecular hydrogen atom shift forming a hydroperoxide functionality and an alkyl radical. A molecular oxygen will rapidly attach to this alkyl radical and form a new and more oxidized $RO_2$. This reaction is called as autoxidation and the newly formed $RO_2$ can go through another autoxidation or bimolecular termination reactions to form a stabilized product (Crounse et al., 2013). Autoxidation is suggested to be responsible for widely detected HOMs in the atmosphere, because it can form highly oxygenated $RO_2$ in a short time scale. In terms of biomolecular reactions, $RO_2$ reacts appreciable only with hydroperoxyl radical ($HO_2$), NO, and another $RO_2$. The $RO_2$ reaction chain in polluted areas is largely terminated by NO, which prohibits generation of compounds with high oxidation levels and reduces yields of HOMs (Bianchi et al., 2019).

Nevertheless, autoxidation reactions alone are not enough to explain the large numbers of
oxygen atoms and low double bond equivalence (DBE, calculated as $nC - \frac{nH+nN}{2} + 1$ where
$nC$, $nH$, and nN stand for the number of carbon, hydrogen, and nitrogen atoms, respectively, in
a molecular) for HOMs observed in laboratory experiments and ambient campaigns. Take
alkylbenzenes as an example, previous studies suggest that the main products of OH-initiated
oxidation of alkylbenzenes ($C_xH_{2x-6}$, x=7, 8, or 9), i.e., bicyclic peroxy radicals (BPR, $C_xH_{2x-5}O_5\cdot$, x=7, 8, or 9) (Jenkin et al., 2003), can undergo an autoxidation reaction and form a new
$_5O_5\cdot$, x=7, 8, or 9) (Jenkin et al., 2003), can undergo an autoxidation reaction and form a new
peroxy radical, $C_xH_{2x-5}O_7\cdot$ (x=7, 8, or 9) (Wang et al., 2017).Autoxidation of BPR could be
very fast if it has a favorable structure, as found in a previous study (Wang et al., 2017). On the
other hand, the structure of the resulting $C_xH_{2x-5}O_7\cdot$ is strongly different from that of BPR,
whose autoxidation reaction rate can be as low as the order of 0.001 s$^{-1}$, since it lacks
enhancements from favorable transition state geometries and substitutes or resonance structures
(Bianchi et al., 2019; Otkjær et al., 2018). Such a slow autoxidation reaction rate cannot explain
the extensive existence of HOM monomers with more than 7 oxygen atoms and HOM dimers
with more than 10 oxygen atoms, which are the maximum numbers of oxygen atoms in
stabilized first generation monomer and dimer products, respectively, formed from $C_xH_{2x-5}O_7\cdot$ (Molteni et al., 2018; Wang et al., 2020b; Mentel et al., 2015; Berndt et al., 2018b).
$_5O_7\cdot$ (Molteni et al., 2018; Wang et al., 2020b; Mentel et al., 2015; Berndt et al., 2018b).
Another possibility is the formation of a second oxygen bridge after the hydrogen shift of BPR
(Molteni et al., 2018), but this reaction pathway would not allow a further oxygenation reaction
without a breakage of the carbon ring, which is also unpromising. A very recent investigation
offers new insights into the formation mechanism of these products, indicating the molecular
rearrangement of BPR can initiate a series of autoxidation (Iyer et al., 2023). However, the
formation mechanism of HOMs with a large hydrogen atom number, i.e., low DBE, is still
vague. For example, monomer products with 16 hydrogen atoms in the OH-initiated oxidation
of TMB and those with 14 hydrogen atoms in the OH-initiated oxidation of xylene were
observed in the laboratory, both with a DBE of 2 lower than their precursors' (Molteni et al.,
2018), but their formation mechanisms cannot be explained by any known mechanisms with
only one OH attack.
Multigeneration reactions of VOCs complicate HOMs' formation. Previous studies
indicate that HOMs can also be formed by sequential oxidation of stabilized first-generation
products of benzene and toluene (Garmash et al., 2020; Cheng et al., 2021). Garmash et al.
(2020) conducted OH oxidation experiments of benzene and toluene with an OH exposure
equivalent to atmospheric oxidation times of 10 hours – 15 days at OH concentrations of ~10$^6$
molecules cm$^{-3}$. Cheng et al. (2021) simulated oxidation of benzene and toluene with an OH
exposure equivalent to 2.4 – 19.4 days of atmospheric photochemical ageing. Certainly, such
extremely high OH exposures favor secondary OH chemistry and help to facilitate our
understanding on product distributions, but such a long timescale limits atmospheric
implications of their results, given the complex physical and chemical processes that occur at
night.
Compared to benzene and toluene, trimethylbenzene (TMB) is a compound characterized
with much larger HOM molar yields when reacted with OH, and the abundance of TMB in the
atmosphere is unignorable (Molteni et al., 2018; Yuan et al., 2012). Previous laboratory
experiments on TMB-derived HOMs mainly focused on the autoxidation reactions of BPR and
the influences of $NO_x$, and the quantity of experiments was very finite, restricting the
application of their conclusions to atmospheric relevant conditions (Tsiligiannis et al., 2019;
Wang et al., 2020b). From the mechanism perspective, a number of HOM monomers with more
than 7 oxygen atoms detected in the OH-initiated oxidation of TMB were previously assumed
to be generated via multiple autoxidation reactions (Molteni et al., 2018). Nevertheless, a
subsequent OH oxidation of the first-generation oxygenated products might be more plausible
for the formation of HOM monomers with more than 7 oxygen atoms from the present point of
view. Indeed, laboratory experiments show that $RO_2$ formed during the second-generation OH
oxidation of the stabilized first-generation oxidation products can also undergo autoxidation
reactions, which entangles reaction mechanisms potentially involved in the formation of those
HOMs and justifies more investigations on the multigeneration OH oxidation of aromatics
(Wang et al., 2020b). Atmospheric OH concentration ([OH]) up to $6\times10^6 - 2.6\times10^7$ molecule
$cm^{-3}$, which is several times higher than the typical average atmospheric [OH], $1.5\times10^6$
molecule $cm^{-3}$ (Jacob, 1999), has been frequently observed in both urban and suburban
environments in China (Tan et al., 2019; Lu et al., 2012), leading to a realistic implication of
multigeneration OH oxidation. Therefore, it is imperative to study chemical characteristics of
aromatics-derived HOMs at different OH exposures, especially those that are less than or
equivalent to one day of atmospheric oxidation.
In this study, two series of laboratory experiments on OH-initiated oxidation of 1,3,5-TMB,
selected as an example of anthropogenic VOCs, were conducted. One was conducted with [OH]
ranging from $9.32\times10^7$ to $1.03\times10^9$ molecule $cm^{-3}$, corresponding to an OH exposure equivalent
to atmospheric oxidation times of roughly $0.7 - 6.9$ hours at an average daytime [OH] of $2.0\times$
$10^6$ molecules $cm^{-3}$. A nitrate-based chemical ionization mass spectrometer (nitrate CIMS) and
a Vocus proton-transfer-reaction mass spectrometer (Vocus PTR) were deployed to measure
the oxidation products and the precursor, respectively. We explored the evolution of oxidation
products to investigate the secondary OH chemistry of stabilized first-generation oxidation
products generated by the oxidation of 1,3,5-TMB. Furthermore, the influence of NO on the
formation of HOMs was investigated by introducing $N_2O$ into the reaction system. In addition,
another series of experiments under atmospheric relevant [OH] were conducted to confirm the
applicability of the above-developed multi-generation OH oxidation mechanisms in the
ambient atmosphere.
**2 Methods**

OH-initiated oxidation of 1,3,5-TMB was investigated in a potential aerosol mass

oxidation flow reactor (PAM OFR) system at $T = 298 \pm 1$ K and a pressure of 1 atm (Lambe et
al., 2015). Two series of experiments were conducted, one under high [OH] conditions and the
other under low [OH] conditions. Hereafter, we refer to the series of high [OH] experiments as
'the 1st-round experiments' and the low [OH] ones as 'the 2nd-round experiments', respectively.
The $i$th experiment in the 1st-round experiments is labelled as 1-$i$ and the one in the 2nd-round
experiments as 2-$i$, where $i$ stands for its serial number. The experimental settings in this study
differed slightly from what were used previously (Wang et al., 2020b). In the 1st-round
experiments, forty OH experiments without $NO_x$ (Exp. 1-1 – 1-40) and twenty-eight
experiments with $NO_x$ (Exp. 1-41 – 1-68) were performed. Seven experiments were conducted
in the 2nd-round, four without $NO_x$ (Exp. 2-1 – 2-4) and three with $NO_x$ (Exp. 2-5 – 2-7). The
experimental conditions are summarized in Table S1, including concentrations of the precursor,
ozone, and NO and $NO_2$. The equivalent OH exposure in the OFR for each experiment was
estimated according to the precursor consumption, and also listed in Table S1. OH exposures
in the OFR were in the range of $(5.2 – 48.7)\times10^9$ and $(0.6 – 5.5)\times10^9$ molecules $cm^{-3}$ s in the
1st-round and 2nd-round experiments, respectively.

A home-made 1,3,5-TMB/$N_2$ cylinder was used as a stable gaseous precursor source in the

experiments, from which the flow rate of 1,3,5-TMB/$N_2$ varied between 1 – 3 sccm (standard
cubic centimeter per minute), leading to $7.08\times10^{11} – 1.54\times10^{12}$ molecule $cm^{-3}$ of 1,3,5-TMB in
the 1st-round experiments, and $7.55\times10^{11}$ or $8.45\times10^{11}$ molecule $cm^{-3}$ of 1,3,5-TMB in the 2nd-
round experiments, respectively (Table S1). A total flow of 15 slpm (standard liters per minute)
zero-gas generated by a zero-gas generator (model 737-13, Aadco Instruments Inc.), together
with the 1,3,5-TMB/$N_2$ flow, was introduced into the OFR. The reaction time in both series of
experiments was kept at around 53 s and the flow reactor was kept as a plug flow one in both
series. The flow in the PAM OFR is laminar with a very low axial mixing, as characterized
with a Taylor dispersion model in a previous study (Lambe et al., 2011). 6 slpm out of the 15
slpm zero-gas was initially passed through a Nafion humidifier (Perma Pure Model FC100-80-
6MSS) filled with ultra-pure water and finally converged with the main flow into the OFR to
achieve and keep a desired RH of $20.0 \pm 2.5$ % in the OFR throughout all the experiments, and
2 slpm was initially passed through a separate ozone chamber, resulting in an initial ozone
concentration of around $1.05 \times 10^{13} - 2.16 \times 10^{13}$ molecule cm$^{-3}$ in the OFR in the 1$^{st}$-round
experiments and $3.01 \times 10^{12} - 3.72 \times 10^{12}$ molecule cm$^{-3}$ in the 2$^{nd}$-round experiments,
respectively. The OFR was operated with only the 254 nm lights on, under which the primary
oxidant production reactions in the OFR were $O_3 + hv\ (254\ nm) \rightarrow O_2 + O(^1D)$ and
$O(^1D) + H_2O \rightarrow 2OH$. After turning on of UV lights, a HOM compound is believed to be
generated if its signal is more than 3 standard deviations of its background signal. If the
fluctuations in the 1-min-averaged signals of both TMB in the Vocus PTR and typical HOMs
(i.e., $C_9H_{14}O_7(NO_3)^-$) in the nitrate CIMS are within 2% during a 10-min period, a steady state
was assumed to be reached. It usually took around no more than 2 minutes for the signals of
HOMs to stabilize after the adjustment of UV lights. We typically monitored the reaction
products for around 20 minutes for each experiment. An ozone monitor (Model 106-M, 2B
technologies) and a trace-gas analyzer for NO-NO$_2$-NO$_x$ (Thermo, 42i-TL) were placed at the
exit of the OFR to measure concentrations of ozone and NO$_x$, respectively.

Non-tropospheric VOC and OVOC photolysis is a typical issue that should be taken into

account when evaluating the OFR settings, especially under the high UV light dose settings in
the 1$^{st}$-round experiments. Our evaluation on photolysis of the precursor and HOMs shows that
photolysis was not a contributor to our observation on C9 and C18 HOM formation. The
photolysis rate of 1,3,5-TMB can be estimated based on the absorption cross-sections of 1,3,5-
TMB at 254 nm (Keller-Rudek et al., 2013) and UV photon fluxes estimated by a chemistry
model discussed in the following sections. The ratio of photolysis-to-OH reaction for 1,3,5-
TMB in our 1$^{st}$-round experiments was merely $0.010 - 0.033$. Hence, photolysis of 1,3,5-TMB
was insignificant in the OFR. For stabilized products such as C9 and C18 HOMs, the cross
sections of organic molecules are usually $\sim 3.9 \times 10^{-18} - 3.9 \times 10^{-17}$ cm$^2$ (Peng et al., 2016), while
the reaction rate between OH and the stabilized first-generation products are estimated to be
around $1.28 \times 10^{-10}$ molecule$^{-1}$ cm$^3$ s$^{-1}$, as suggested by Master Chemical Mechanism (MCM)
(Jenkin et al., 2003). Hence, the ratio of photolysis rates of C9 and C18 HOMs to their
secondary OH oxidation rates is estimated to be around $0.020 - 0.056$ in the 1$^{st}$-round
experiments. In the 2$^{nd}$-round, the influences of photolysis should be even lower due to the
much lower light intensity.

For experiments with NO$_x$ in the 1$^{st}$-round experiments, 350 sccm N$_2$O (99.999%, Air

Liquid) was added into the OFR to produce and sustain NO$_x$ mixing ratios at levels that were
sufficiently high to be a competitive sink for RO$_2$ radicals. NO and NO$_2$ were produced via the
reaction $N_2O + O(^1D) \rightarrow 2NO$, followed by the reaction $NO + O_3 \rightarrow NO_2 + O_2$. Two sets of
irradiance intensities were chosen for NO$_x$ experiments, generally resulting in two NO$_x$ levels,
$4.41 \times 10^{10}$ molecule cm$^{-3}$ NO + $1.72 \times 10^{12}$ molecule cm$^{-3}$ NO$_2$ (Exp. 1-41 – 1-54) and $1.18 \times 10^{11}$
molecule cm$^{-3}$ NO + $2.94 \times 10^{12}$ molecule cm$^{-3}$ NO$_2$ (Exp. 1-55 – 1-68) at the exit of the OFR.
With the aim to slightly modify OH exposure but keep NO$_x$ concentrations constant among
each set of experiments, the initial concentrations of 1,3,5-TMB were adjusted in a large range
from $4.09 \times 10^{11}$ to $2.06 \times 10^{12}$ molecule cm$^{-3}$ while RH and irradiances were not changed, as an
increase in the precursor concentration corresponds to a larger sink for OH. In the 2$^{nd}$-round
experiments, due to the lower O($^1$D) in the PAM OFR, 2.5 slpm pure N$_2$O was utilized instead,
whereas the total flow rate was kept the same as that in the 1$^{st}$-round. We lowered the light
intensity to obtain a lower [OH] in the PAM OFR, which also resulted in fluctuations in the NO
concentrations ([NO]) from $3.19 \times 10^{10}$ to $1.74 \times 10^{11}$ molecule cm$^{-3}$ and the NO$_2$ concentrations
([NO$_2$]) from $2.70 \times 10^{11}$ to $9.31 \times 10^{11}$ molecule cm$^{-3}$.
A nitrate CIMS (Ehn et al., 2014; Eisele and Tanner, 1993) and a Vocus PTR (Krechmer
et al., 2018) were deployed at the exit of the OFR to measure the oxidation products of 1,3,5-
TMB in the 1$^{st}$-round experiments. These two mass spectrometers have been well characterized
in a previous study (Wang et al., 2020b).
The sample flow rate for the nitrate CIMS in the 1$^{st}$ round-experiments was 8 slpm through
a Teflon tube with an outer diameter (OD) of 1/4 in. and a length of 70 cm. The sheath flow for
the nitrate CIMS was supplied by a zero-gas generator at a flow rate of 15 slpm. Mass resolution
was approximately 8000 for ions with m/z larger than 200 Th. HOMs generated from TMB
oxidation were charged in the ambient pressure interface region by collisions with nitrate
clusters, $(HNO_3)_x \cdot NO_3^-$ ($x = 0 - 2$), and detected by nitrate CIMS as clusters with NO$_3^-$, i.e.,
HOM·NO$_3^-$ (Hyttinen et al., 2015). In addition, HOMs' signals were corrected with relative
transmission efficiencies of our nitrate CIMS (Heinritzi et al., 2016). We followed the same
sampling method of PAM OFR as those in previous studies, in order to obtain a similar flow
tube residence time distributions (RTDs) and thus validate usage of a modified PAM_chem_v8
model to estimate concentrations of radicals in the OFR as discussed below.
Vocus PTR was applied to quantify precursor concentrations. The focusing ion-molecule
reactor (FIMR) was heated up and its temperature was maintained at 100 °C during the
experiments. The FIMR can be operated under 2.0 mbar without a strong interference from
corresponding water clusters when ionizing the neutral compounds. The Vocus front and back
voltages were 650 V and 15 V, respectively, forming an axial voltage of 635V and a reduced
electrical field (*E/N*, where E is the electric field strength and N is the number density of the
buffer gas in FIMR) of 180 Td. The radio frequency (RF) voltages and frequency were set to
be 450 V and 1.3 MHz, respectively. The sample flow was introduced to the Vocus PTR
through a Teflon tube with an OD of 1/4 in. and a length of 120 cm from the OFR. A total
sample flow of 1.4 slpm was maintained by a pump with an orifice to minimize the delay time
of sampling, from which approximately 125 sccm was sampled into the FIMR through a
capillary tube.

In the 2nd-round experiments, a Vocus CI-TOF (Towerk AG, Switzerland) equipped with

a Vocus Aim inlet and the same nitrate-ion chemical ionization source as adopted in the 1st-
round experiments was utilized to measure oxidation products, hereafter referred as nitrate CI-
TOF. The nitrate CI-TOF was characterized with a flat transmission efficiency between m/z 60
Th and m/z 500 Th, as well as a mass resolution of 10000 at m/z 200 Th. In this series of
experiments, the reaction products were sampled from the PAM OFR via a 30 cm-long Teflon
tube with a 1/2 in. OD to our nitrate CI-TOF. The Vocus PTR and the ozone monitor were
connected to the PAM OFR from a separate port via a 120 cm-long Teflon tube with a 1/4 in.
OD.

We did not quantify HOMs' concentrations. Since the inner diameters of PAM OFR,

sampling tube, and the nitrate CIMS inlet were different, and two reducing unions were used
during sampling, the estimation of the penetration efficiency and sampling efficiency of HOMs
are thus of a significant uncertainty. The initial concentrations of TMB utilized in both sets of
experiments fluctuated slightly, which resulted from sample preparation processes and were
more obvious in the 1st-round experiments. Therefore, in the discussion on the data of the 1st-
round experiments, we tried to minimize potential influences of the differences in the initial
TMB concentrations on the signals of HOMs by normalizing the HOMs signals with the initial
TMB signal. To precisely illustrate changes in the abundance of HOMs at different OH
exposures, a normalized signal was chosen to present the abundance of detected HOMs, which
is defined as the ratio of the signals of HOMs in the nitrate CIMS normalized by the reagent
ions and the initial signal of 1,3,5-TMB, i.e., $S(HOMs)/S(TMB)$. $S(HOMs)$ is the signal of
HOM detected by the nitrate CIMS normalized with the signal of reagent ions, whereas
$S(TMB)$ is the initial signal of 1,3,5-TMB detected by the Vocus PTR.

To compare chemical regimes of two series of experiments and the ambient atmosphere,

a PAM chemistry model (PAM_chem_v8), utilized widely in previous studies, were chosen
with the latest updates to calculate radical profiles in our OFR (Li et al., 2015; Cheng et al.,
2021; Wang et al., 2020b; Mehra et al., 2020; Lambe et al., 2015, 2018; Peng and Jimenez,
2020; Lambe et al., 2017). This model is based on a photochemical box model that includes
chemistry of photolysis of oxygen, water vapor, and other trace gases by the primary
wavelengths of mercury lamps, and simplified VOC and $RO_2$ chemistry, but further reactions
of the first-generation stabilized products and the second-generation organic radicals are not
considered. The reactions and corresponding kinetics utilized in this model were summarized
in Table S2. In this work, autoxidation and accretion of 1,3,5-TMB-derived BPR, as well as
subsequent reactions of the autoxidation product of BPR, i.e., $C_9H_{13}O_7\cdot$, are newly implemented
or modified in this model (Reaction No. 46 – 62 in Table S2). These two radicals were the most
significant $RO_2$ in the system and represented the whole $RO_2$ pool in the PAM chemistry model
simulation. The pathways of peroxy radicals and their kinetics are discussed below. $NO_x$-related
reactions are also included in the model. When experiments without $NO_x$ are simulated, these
$NO_x$-related reactions do not contribute to the simulation results.
The detailed reactions involved with $RO_2$ include:
$$RO_2 + R'O_2 \rightarrow RO + R'O + O_2 \qquad\qquad (R1)$$
$$RO_2 + R'O_2 \rightarrow R=O + R'OH + O_2 \qquad\qquad (R2)$$
$$RO_2 + R'O_2 \rightarrow ROH + R'=O + O_2 \qquad\qquad (R3)$$
$$RO_2 + R'O_2 \rightarrow ROOR' + O_2 \qquad\qquad (R4)$$
$$RO_2 + HO_2 \rightarrow ROOH + O_2 \qquad\qquad (R5)$$
$$RO_2 + OH \rightarrow Products \qquad\qquad (R6)$$
$$RO_2 \xrightarrow{isomerization} Products \qquad\qquad (R7)$$
$$RO_2 + NO \rightarrow RO + NO_2 \qquad\qquad (R8)$$
$$RO_2 + NO \rightarrow RONO_2 \qquad\qquad (R9)$$
$$RO_2 \rightarrow physical\ loss \qquad\qquad (R10)$$
$R1$, $R2$, and $R3$ are reactions of $RO_2 + RO_2$, forming alkoxy radicals, carbonyl termination
products, and hydroxyl termination products, respectively. $R4$ is the accretion reaction,
forming dimers via combination of two monomeric $RO_2$. $R5$ is the reaction between $RO_2$ and
$HO_2$, forming hydroperoxyl radicals. The reaction rate constants for $RO_2$ in $R1 - R5$ are
obtained by MCM or previous investigations (e.g., Jenkin et al., 2003; Berndt et al., 2018; Peng
and Jimenez, 2020). We treat $R1 - R3$ as a total reaction with a reaction rate constant of $8.8\times10^{-13}$
molecule$^{-1}$ cm$^3$ s$^{-1}$, and branching ratios of $R1 - R3$ of 0.6, 0.2, and 0.2, respectively, as
suggested by MCM (Jenkin et al., 2003). The reaction rate constants of BPR and $C_9H_{13}O_7\cdot$ for
$R4$ are $1.7\times10^{-10}$ and $2.6\times10^{-10}$ molecule$^{-1}$ cm$^3$ s$^{-1}$, respectively (Berndt et al., 2018b). The
reaction rate constants for $R5$ is $1.5\times10^{-11}$ molecule$^{-1}$ cm$^3$ s$^{-1}$ (Jenkin et al., 2003).
$R6$ is the reaction between OH and $RO_2$, whose reaction rate constant is $1\times10^{-10}$ molecule$^{-1}$
cm$^3$ s$^{-1}$ according to previous studies (Bossolasco et al., 2014; Yan et al., 2016; Assaf et al.,
2016, 2017; Peng and Jimenez, 2020). Current knowledge on the reaction products for the
reaction of $CH_3O_2\cdot + OH$, the most studied $RO_2 + OH$ reaction, is summarized in Table S3. The
products of this reaction are suggested to include a Criegee intermediate ($CH_2O_2\cdot$), a stabilized
methylhydrotrioxide ($CH_3OOOH$), an alkoxy radical ($CH_3O\cdot$), and methanol ($CH_3OH$) (Yan et
al., 2016; Fittschen, 2019; Caravan et al., 2018; Müller et al., 2016). Müller et al. (2016) and
Caravan et al. (2018) suggested that the formation of $CH_2O_2\cdot$ is actually infeasible, and Yan et
al. (2016) estimated an upper limit branching ratio of 5% for this pathway. The branching ratios
of stabilized products $CH_3OH$ and $CH_3OOOH$ are 6 – 7% (Caravan et al., 2018; Müller et al.,
2016) and 7% (Müller et al., 2016), respectively. The most significant product of this reaction
is the alkoxy radical ($CH_3O\cdot$), with a branching ratio of more than 86% (Müller et al., 2016).
In the absence of $NO_x$, $CH_3OH$ and $CH_3O\cdot$ can also be formed via the traditional unimolecular
reaction between $CH_3O_2\cdot$ and $RO_2$, i.e., $R1$ and $R3$. The possible role of this reaction of large
$RO_2$, i.e., BPR and other $C9\text{-}RO_2$, with OH has not yet been investigated. However, according
to the branching ratios for the reaction of $CH_3O_2\cdot$ + OH, this reaction is likely to form RO
instead of stabilized C9 products. Hence, we assume that the branching ratios of hydrotrioxide
(ROOOH), RO, and ROH are 0.07, 0.86, and 0.07, respectively, for BPR + OH and $C9\text{-}RO_2$ +
OH.

$R7$ is the unimolecular reactions of $RO_2$ in the PAM OFR. $RO_2$ isomerization rate

coefficients are highly dependent on their structures, spanning from $10^{-3} – 10^6$ $s^{-1}$ (Bianchi et
al., 2019; Crounse et al., 2013; Knap and Jørgensen, 2017; Praske et al., 2018). However, only
some substituted acyl $RO_2$ can undergo rapid isomerization at a reaction rate of $10^6$ $s^{-1}$ (Knap
and Jørgensen, 2017). 1,3,5-TMB-derived BPR and its autoxidation product, $C_9H_{13}O_7\cdot$, do not
belong to this group of substituted acyl $RO_2$ (Molteni et al., 2018; Tsiligiannis et al., 2019). The
most important unimolecular reactions for 1,3,5-TMB-derived BPR is likely autoxidation while
the precise autoxidation reaction rates of 1,3,5-TMB-derived BPR and other $RO_2$ in this system
are currently unclear (Bianchi et al., 2019; Molteni et al., 2018). Previous theoretical
investigations suggest that more than 90% BPR generated by the oxidation of 1,3,5-TMB
possess a structure favoring autoxidation and thus their overall autoxidation reaction rate is
relatively fast (Wang et al., 2017). We follow quantum calculation results on the autoxidation
reaction of a methyl group adjacent to the $RO_2$ functionality group (Wang et al. 2017), and time
the suggested rate (0.026 $s^{-1}$) by 3 due to the symmetry with three methyl groups in our parent
compound. The obtained autoxidation reaction rate is 0.078 $s^{-1}$.

$R8$ and $R9$ are the reactions between NO and $RO_2$, generating alkoxy radicals and

organonitrates, respectively. The reaction rate for the sum of these two reactions is $8.5\times10^{-12}$
molecule$^{-1}$ cm$^3$ s$^{-1}$. The branching ratios of these two reactions are 0.843 and 0.157, respectively,
according to MCM (Jenkin et al., 2003).

Alkoxy radicals, RO, will be generated in $R1$, $R6$, and $R8$. The widely used near-explicit

mechanism, MCM, assumes that RO formed via the alkoxy channel of BPR ($R1$) will
decompose into small molecules. Recently, Xu et al. (2020) probed the chemical fates of BPR-
derived RO, hereafter referred to as bicyclic alkoxy radical (BCP-oxy), in the oxidation of

benzene by laboratory experiments and model calculations, which can be taken as a reference to induce the mechanism of 135-TMB-derived BCP-oxy. BCP-oxy can undergo two reactions, i.e., ring-breakage and ring-closure, and a new calculation result suggests that the branching ratio of ring-breakage reaction is larger than 98% (Wang et al., 2013). 56% of ring-breakage reactions will break benzene-derived BCP-oxy into butenedial and glyoxal, and the rest 44% will generate a C6 alkyl radical by a 1,5-aldehydic H-shift. The latter C6 alkyl radical will further undergo other reactions, including a 93% branching ratio for decomposition reactions that results in a reduction of carbon atom number (Xu et al., 2020). Therefore, most of benzene-derived BCP-oxy will likely decompose into compounds with fewer carbon atoms. We assume that 1,3,5-TMB-derived BCP-oxy will undertake these decomposition reactions with a similar branching ratio, which means that these radicals cannot form a large number of stabilized products that can influence the distributions of stabilized C9 products in nitrate CIMS.

$R10$ is the physical loss of $RO_2$. The physical loss of $RO_2$ in the PAM OFR consists of the condensation loss to the aerosol particles and the diffusion loss to the OFR walls. In our experiments, measurement results by a long-SMPS show that the aerosol particles presented in the PAM OFR were few. The long SMPS consisted of a long-DMA (TSI model 3081) and a CPC (TSI model 3787), covering a particle number size distribution from 13.6 nm to 736.5 nm. Thus, though not detected in this study, we cannot absolutely deny the possibility that particles might have been generated, resulting in a larger physical loss of HOMs. This part of physical loss might be underestimated. The first-order loss rate of HOMs to the OFR walls, $k_{wall}$, is limited by eddy diffusion and can be calculated with the following function (Cheng et al., 2021; Palm et al., 2016; McMurry and Grosjean, 1985):

$$k_{wall} = \frac{A}{V} \cdot \frac{2}{\pi} \cdot \sqrt{k_e D_g} \qquad (Eq1)$$

where the OFR surface-area-volume ratio ($A/V$) is 25 m$^{-1}$ and the coefficient of eddy diffusion ($k_e$) is 0.0042 s$^{-1}$, as estimated by the method utilized in a previous study (Brune, 2019) and given in $Eq2$.

$$k_e = 0.004 + 10^{-2.25} V^{0.74} \qquad (Eq2)$$

where $V$ is the enclosure volume (m$^3$). The molecular diffusion coefficient, $D_g$, is estimated with the method as described by Fuller et al. (1966) and is around $5 \times 10^{-6}$ m$^2$ s$^{-1}$ with 1,3,5-TMB derived BPR as an example. Hence, $k_{wall}$ is around 0.0023 s$^{-1}$ in the PAM OFR.

Other kinetic data in the modified PAM_chem_v8 model are obtained from the IUPAC (International Union of Pure and Applied Chemistry) dataset (https://iupac-aeris.ipsl.fr, last access: 26 October 2023) and the MCM dataset (MCM v3.3.1, https://mcm.york.ac.uk/MCM/, last access: 9 October 2023).

For the 1$^{st}$-round experiments, the input parameters of temperature, mean residence time,

water vapor concentration, $O_3$ concentration, and the initial 1,3,5-TMB concentration are 25 °C,
53 s, 0.63%, $1.23 \times 10^{13}$ molecule cm$^{-3}$, and $1.23 \times 10^{12}$ molecule cm$^{-3}$, respectively, as measured
directly. For the 2$^{nd}$-round experiments, the input parameters of $O_3$ concentration and the initial
1,3,5-TMB concentration were updated as $3.68 \times 10^{12}$ molecule cm$^{-3}$ and $7.55 \times 10^{11}$ molecule
cm$^{-3}$, respectively. In the NO$_x$ experiments, the input flow rate of N$_2$O is 350 sccm in the 1$^{st}$-
round experiments and 2.5 slpm in the 2$^{nd}$-round experiments, respectively. The actinic flux at
254 nm, $I_{254}$, is constrained by comparing OH exposures by model output and OH exposures
estimated by the consumption of 1,3,5-TMB as measured by the Vocus PTR. Consumption of
$O_3$ estimated by the model agrees well with the measured results, with discrepancies being
always within 10% at different OH exposures.
**3 Results and discussion**
**3.1 Comparison of chemical regimes**

Concentration profiles of OH, RO$_2$, and HO$_2$ as a function of OH exposures in our high

[OH] experiments without NO$_x$, i.e., the 1$^{st}$-round experiments, are illustrated in Figure S1a.
According to the modified PAM_chem_v8 model, when [OH] increased from $9.32 \times 10^7$ to
$1.03 \times 10^9$ molecule cm$^{-3}$, [HO$_2$] increased from $7.25 \times 10^8$ to $2.79 \times 10^9$ molecule cm$^{-3}$, whereas
[RO$_2$] concentrations increased from $5.17 \times 10^9$ to $9.5 \times 10^9$ molecule cm$^{-3}$. The radical
concentrations in high [OH] experiments with NO$_x$ (Figure S1b) varied in a similar range, with
[RO$_2$] ranging from $4.38 \times 10^9$ to $9.13 \times 10^9$ molecule cm$^{-3}$, HO$_2$ ranging from $4.47 \times 10^9$ to
$6.47 \times 10^9$ molecule cm$^{-3}$, and OH ranging from $3.86 \times 10^8$ to $7.82 \times 10^8$ molecule cm$^{-3}$,
respectively. The ratios of between HO$_2$/OH and RO$_2$/OH in the 1$^{st}$-round experiments were
generally in the same order of magnitude as those in with the ambient atmosphere (Whalley et
al., 2021).

Radical concentrations were also estimated by the PAM_chem_v8 model to illustrate the

chemical regimes in the 2$^{nd}$-round experiments (Table S4). The average [HO$_2$], [OH], and [RO$_2$]
were $9.7 \times 10^7$, $1.64 \times 10^7$, and $1.69 \times 10^9$ molecule cm$^{-3}$, respectively, in Exp. 2-3, and were
$6.7 \times 10^7$, $1.04 \times 10^7$, and $1.34 \times 10^9$ molecule cm$^{-3}$, respectively, in Exp. 2-4, both of which
generally differ by no more than a factor of 3 from the summer daytime ambient ones in polluted
atmospheres (Tan et al., 2017, 2018, 2019; Whalley et al., 2021; Lu et al., 2012). The average
[HO$_2$], [OH], and [RO$_2$], as well as the NO and NO$_2$ concentrations in Exp. 2-7 are generally
very close to those in the same environment (Tan et al., 2019).

We take Exp. 1-12 ([OH] = ~$8.47 \times 10^8$ molecule cm$^{-3}$ and NO$_x$ = 0) and Exp. 2-3 ([OH] =

~$1.64 \times 10^7$ molecule cm$^{-3}$ and NO$_x$ = 0) as representative examples and compare simulation
results with those from the ambient atmosphere, since $NO_x$ in the ambient is believed not to
impact relative ratios for $R1 – R3$, $R5$, and $R6$. In the ambient atmosphere, the average $[HO_2]$,
$[OH]$, and $[RO_2]$ were $2.7 \times 10^8$, $8.0 \times 10^6$, and $1.4 \times 10^9$ molecule $cm^{-3}$, respectively, around
summertime noon in urban Beijing (Whalley et al. 2021), and $(4 – 28) \times 10^8$, $(0.8 – 2.4) \times 10^7$,
and $1.2 \times 10^9$ molecule $cm^{-3}$ (modeled) at a suburban site in Yangtze River Delta (Ma et al. 2022).
As shown in Figure 1a, for the most important $RO_2$, BPR, the fractions of monomeric
termination reactions of $RO_2 + RO_2$ ($R1 – R3$), $RO_2 + HO_2$ ($R5$), and $RO_2 + OH$ ($R6$) were
6.2%, 29.3%, and 64.5%, respectively, in Exp.1-12. In contrast, the fractions were 32.5%,
31.8%, and 35.7%, respectively, in Exp. 2-3, whereas the values were 20.3%, 66.6%, and 13.2%,
respectively, for summertime, urban Beijing.
**(a)**

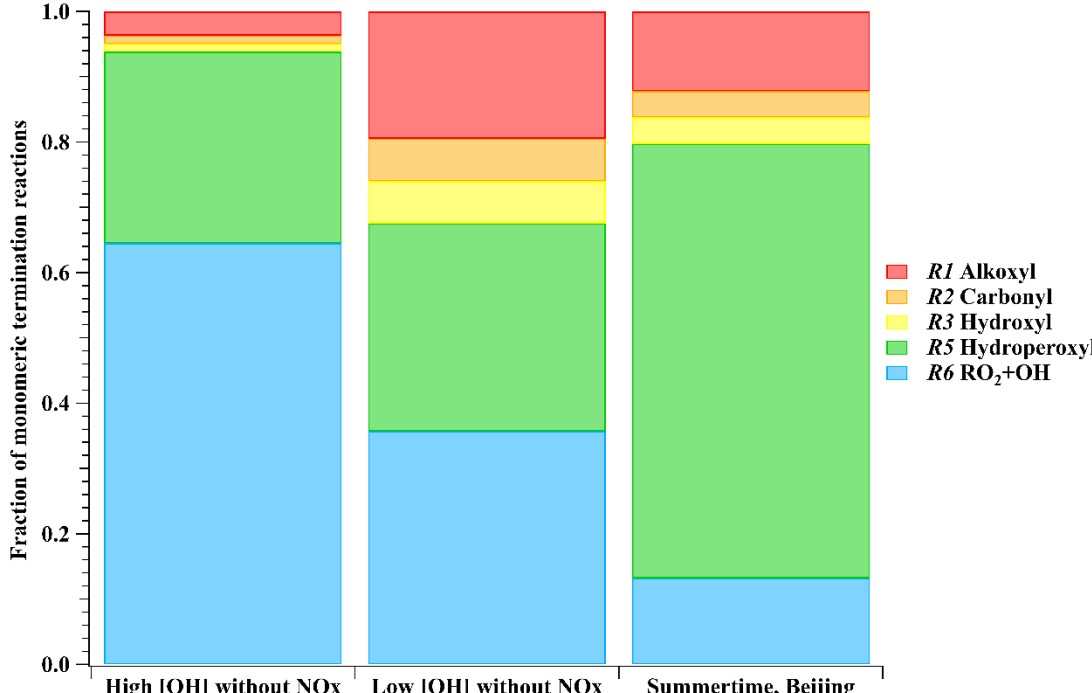


**(b)**

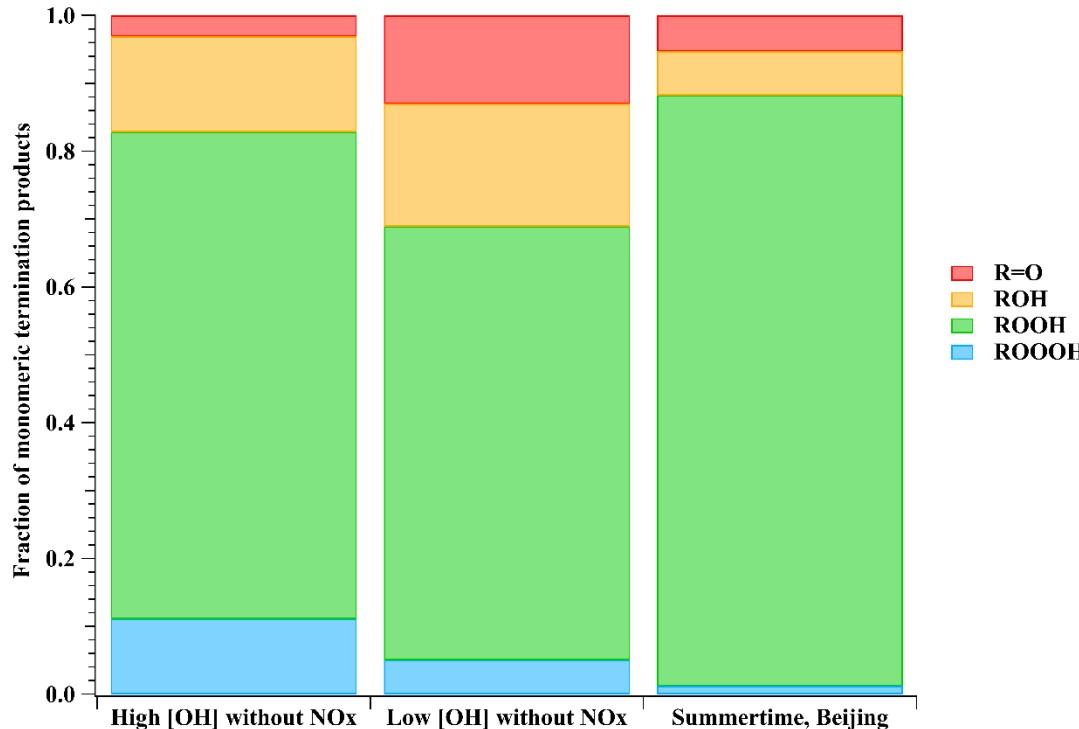


**(c)**

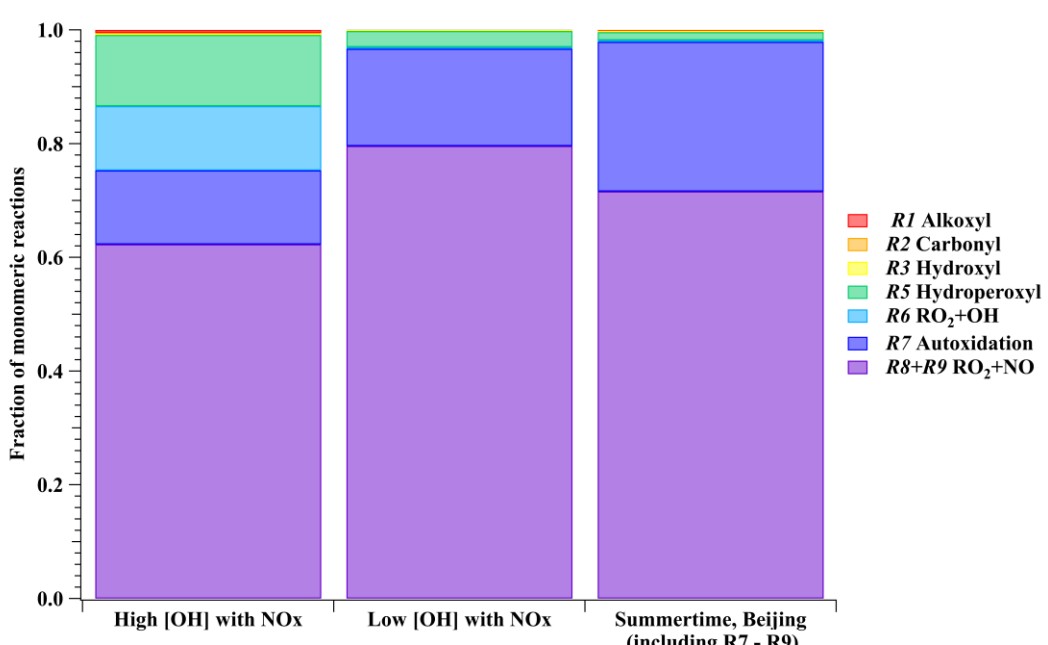


**(d)**

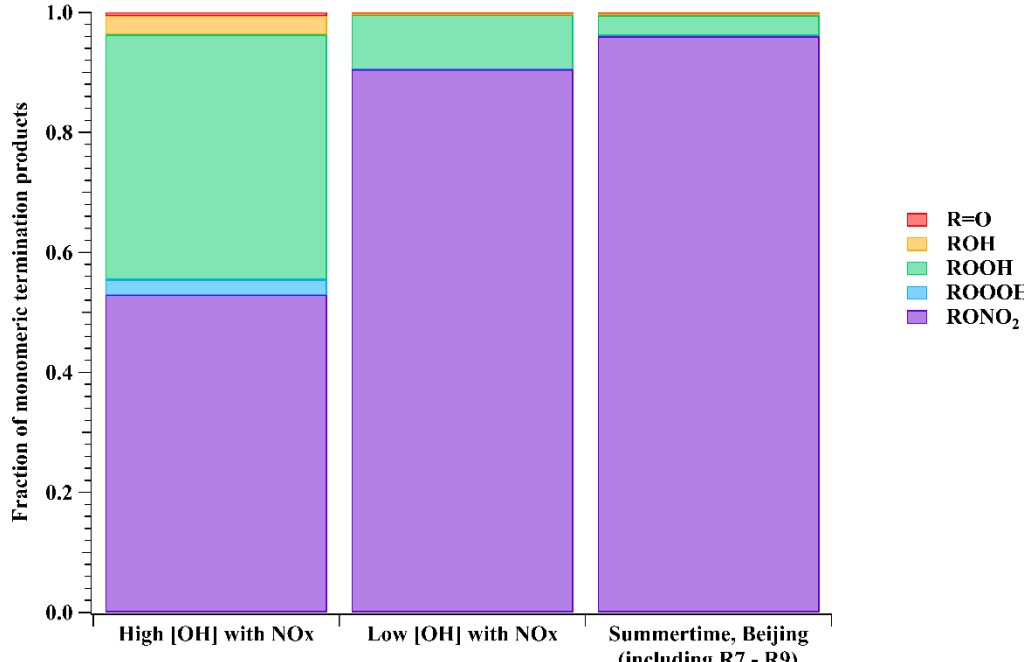


**Figure 1.** (a) The fraction of monomeric termination reactions and (b) monomeric termination
products of BPR in a representative high [OH] experiment without $NO_x$ (Exp. 1-12), a
representative low [OH] experiment without $NO_x$ (Exp. 2-3), and summertime, urban Beijing
(Whalley et al. 2021). $NO_x$ related reactions and products for the Beijing study are not included
for a better comparison. (c) The fraction of monomeric reactions ($R1 – R3$ and $R5 – R9$) and
(d) monomeric termination products of BPR in a representative high [OH] experiment with
$NO_x$ (Exp. 1-48), a representative low [OH] experiment with $NO_x$ (Exp. 2-7), and summertime,
urban Beijing (Whalley et al. 2021). Reactions and kinetic rate coefficients used in the
calculations are provided in Table S2.

Our $NO_x$-free experiments are characterized with an inherent drawback that the proportion
of the $HO_2$ termination pathway ($R5$) is actually lower than that under ambient conditions,
which is similar to most other laboratory experiments (Bianchi et al., 2019). In our high [OH]
experiments without $NO_x$, the reaction rates of unimolecular reactions, e.g., autoxidation
reaction ($R7$) and condensation ($R10$) did not change with [OH] that increased in our
experiments relative to that in the ambient. As a result, relative proportions of autoxidation and
condensation were lowered. On the other hand, 1,3,5-TMB-derived BPR was suggested to
undergo autoxidation ($R7$) at a reaction rate of 0.078 s$^{-1}$ (Wang et al., 2017), which represented
36.8%, 94.4%, and 92.8% of the overall rates of $R1 – R3$ and $R5 – R7$ in Exp. 1-12, Exp. 2-3,
and summertime, urban Beijing, respectively. Because of its dominant proportion in Exp. 2-3
and the ambient, the autoxidation channel is not included for clarity in Figure 1a. Autoxidation

did possess a lower significance in our high [OH] experiments due to the other accelerated bimolecular reactions. However, it would only influence the oxygen content of our products but would not change the DBE. Both accretion reaction ($R4$) and condensation ($R10$) have been taken into account in the model, but they would not influence the distributions of monomeric stabilized products. We will specifically discuss these two pathways in the following sections because of their complexity between the laboratory and ambient conditions.

$RO_2$ other than BPR and $C_9H_{13}O_7\cdot$ existed in the PAM OFR, which were not included in the model simulation. Their reaction rates of the accretion reaction ($R4$) and the autoxidation reaction ($R7$) should be different from BPR and $C_9H_{13}O_7\cdot$ due to the strong dependence of these two reaction rates on the molecular structure. Rates for the other reaction channels, on the other hand, should be the same as those of BPR and $C_9H_{13}O_7\cdot$. Therefore, their fates in terms of the monomeric termination reactions ($R1 - R3$, $R5 - R6$, and $R8 - R9$) should be similar as BPR and $C_9H_{13}O_7\cdot$.

Calculated from yields of stabilized monomeric termination products of BPR, the fractions of monomeric termination reaction products in Exp. 1-12, Exp. 2-3, and summertime, urban Beijing (Whalley et al. 2021) are presented in Figure 1b, showing a lot of similarities between these conditions. The fractions of R=O, ROH, ROOH, and ROOOH in Exp. 1-12 were 3.1%, 14.1%, 71.7%, and 11.1%, respectively. These fractions were 13.0%, 18.1%, 63.9%, and 5.0%, respectively, in the Exp. 2-3, and were 5.3%, 6.5%, 87.0%, and 1.2%, respectively, in the summertime Beijing case. Among them, the majority of products are always ROOH and ROH, with ROOH being the most abundant. Therefore, the monomeric termination products of BPR in our experiments are atmospheric relevant. In addition, only the R=O product has a DBE higher than the reacted $RO_2$, but merely accounted for a limited proportion. All the other stabilized termination products have a DBE that is 1 lower than the precursor, and are the majority in both laboratory and ambient conditions. This indicates that the majority of the first-generation products typically have a DBE that is 1 lower than that of 1,3,5-TMB, whereas the majority of subsequent-generation products typically have a DBE that is 2 lower than that of 1,3,5-TMB. Once a monomeric compound with a DBE that is at least 2 lower than that of 1,3,5-TMB was observed, multi-generation OH reactions have happened in the system.

In laboratory experiments in absence of $NO_x$ (e.g., Exp.1-12), the proportions of $R8 - R9$, i.e., the NO channel in the urban atmosphere were attributed to termination reactions of $R1 - R6$, i.e., $RO_2 + RO_2$, accretion reaction, $RO_2 + HO_2$, and $RO_2 + OH$. By expanding proportions of these termination reactions, laboratory investigations on product distributions can be facilitated, as the detection of certain HOM products became more precise and the mass spectra became simplified.

In experiments with $NO_x$, the chemical fates of BPR in high [OH] experiments (Exp. 1-48

as an example, [OH] = ~$6.77\times10^8$ molecule $cm^{-3}$, NO = ~$4.73\times10^{10}$ molecule $cm^{-3}$. $NO_2$ =
~$1.67\times10^{12}$ molecule $cm^{-3}$), low [OH] experiments (Exp. 2-7 as an example, [OH] = ~$1.69\times10^7$
molecule $cm^{-3}$, NO = ~$3.19\times10^{10}$ molecule $cm^{-3}$. $NO_2$ = ~$2.70\times10^{11}$ molecule $cm^{-3}$), and the
summertime, urban Beijing are compared. As shown in Figure 1c, in all three conditions, $RO_2$
reactions with NO were always the most significant pathway, with autoxidation being the
second most significant.
Accounting for at least 52% of monomeric termination products under all conditions,
organonitrates were always the most important termination products, as shown in Figure 1d.
On the other hand, based on the formulae of organonitrates, the detailed formulae of monomer
$RO_2$ could be probed, which can help us better understand the chemical reactions inside the
system. Alkoxy radicals generated in the NO termination channel will unlikely influence the
distributions of C9 stabilized products since they tend to get decomposed in the subsequent
reactions, as discussed in our previous discussion on the fate of alkoxy radicals in Section 2.
Due to the complexity of ambient $RO_2$ pool, it is difficult to estimate the detailed fraction
of accretion reactions $R4$. In the laboratory experiments, $RO_2$ pool mainly consists of BPR and
its autoxidation reaction product $C_9H_{13}O_7\cdot$, which both can undergo accretion reaction rapidly
(Berndt et al., 2018b). The concentrations of these two radicals were estimated by
PAM_chem_v8. The reaction rate of accretion ($R4$) for BPR was around 1.61 $s^{-1}$ in Exp.1-12,
being 88.4% of $R1 - R7$, and was 0.29 $s^{-1}$ in Exp.2-3, equivalent to 77.7% of $R1 - R7$.
Certain uncertainties exist in the estimation of the proportions of accretion reactions, as the
PAM_chem_v8 model only includes the first-generation reactions of precursors, whereas the
subsequential fragmentation and re-initiation of stabilized products can generate a series of new
$RO_2$ that will influence the proportions of accretion reactions. We are only certain that the
significance of accretion reactions in both Exp. 1-12 and Exp. 2-3 is larger than the ambient.
The much-expanded proportion of HOM dimers through accretion reactions makes it
inadequate to compare yields of HOM dimers and HOM monomers. However, this deviation
will not influence our conclusion on multi-generation OH oxidation and identification of HOM
dimers can help us identify the exact $RO_2$ in the OFR and confirm the conditions of secondary
OH oxidation according to the number of hydrogen atoms in the molecules.
In addition, certain compounds might have condensed onto pre-existing particles in the
real atmosphere before an appreciable fraction of such compounds undergoes the re-initiated
OH oxidation. Therefore, even if the same product can be generated both in the laboratory
experiments and the ambient atmosphere, the relative significance of this product is not
completely identical. Though OOMs might have the potential to undergo multi-generation OH
oxidation, the exact proportion of this reaction in the ambient strongly depends on their
volatility, in other words, condensation sink of these OOMs. The typical monomeric
termination products of 1,3,5-TMB-derived BPR, $C_9H_{12}O_4$, $C_9H_{14}O_4$, $C_9H_{14}O_5$, and $C_9H_{13}NO_6$,
are estimated to have saturation vapor concentrations ($C^*$) of 30.20, 30.20, 0.85, and 3.39 μg/m$^3$
at 300 K, respectively with the volatility parameterization developed in the CLOUD chamber
oxidation experiments of aromatics (Wang et al., 2020a). From the perspective of volatility,
they all belong to semi-volatile organic compounds (SVOC, $0.3 < C^* < 300$ μg/m$^3$) and are
expected to exist in both the condensed and the gas phases at equilibrium in the atmosphere
(Bianchi et al., 2019). Compared to ambient conditions, the proportion of their condensation in
the laboratory were biased to be lower due to the accelerated bimolecular reactions. However,
this will not prevent the high [OH] experiments from showing the potential and ability of these
compounds to go through re-initiated OH oxidation, as these compounds would exist in
significant fractions in the gas phase in the real atmosphere.
However, the conditions are completely different for other HOM monomer products and
HOM dimer products with much lower volatility. It is difficult for a HOM dimer, e.g., $C_{18}H_{26}O_{10}$
estimated with a $C^*$ of $7.24\times10^{-13}$ μg/m$^3$ at 300 K, to survive long enough to experience an
appreciable re-initiated photochemical ageing. The lifetime of HOMs that can be classified as
LVOCs ($3\times10^{-5} < C^* < 0.3$ μg/m$^3$) and ELVOCs ($C^* < 3\times10^{-5}$ μg/m$^3$) can be estimated
according to the condensation sink (CS) in the atmosphere, as they are lost irreversibly onto
surfaces. The median value of CS in urban Beijing was reported to be around 0.019 s$^{-1}$ and
0.057 s$^{-1}$ during NPF days and non-NPF days, respectively, whereas the values in Shanghai
were reported to be around 0.013 s$^{-1}$ and 0.017 s$^{-1}$. respectively (Deng et al., 2020; Yao et al.,
2018), which are all much higher than the physical loss in our PAM OFR, i.e., 0.0023 s$^{-1}$.
LVOCs and ELVOCs are believed to be lost irreversibly to the surface in both the laboratory
and ambient because of their low volatility. By assuming a similar diffusion coefficient of
LVOCs and ELVOCs to that of sulfuric acid, the lifetimes of LVOCs and ELVOCs in the
ambient still can still be as high as 77 s for the condensation loss, which is close to the residence
time of our PAM OFR. Therefore, if they were generated by oxidation of aromatics in the
ambient, these LVOCs and ELVOCs should at least have the potential to experience the same
OH exposures as those in our low [OH] experiments, i.e., at least $5.86\times10^8$ molecule cm$^{-3}$ s. On
the other hand, the detailed proportions of LVOCs and ELVOCs after a large OH exposure
should be lower than those in the lab due to their magnified physical loss in the ambient. This
means that if the multi-generation products of those compounds were observed in the ambient
air, they should have been generated via a reaction that happened very recently.
**3.2 Oxidation products in high [OH] experiments**

A total of 33 HOM monomers with formulae of $C_{7-9}H_{8-16}O_{6-11}$ and 22 HOM dimers with

formulae of $C_{17-18}H_{24-30}O_{8-14}$ were observed in the $1^{st}$-round experiments of gas phase OH-
initiated oxidation of 1,3,5-TMB in the OFR, i.e., high [OH] experiments, as listed in Table S5.
The relative signal contributions of HOMs to the total signals of all HOMs at an OH exposure
of $2.38\times10^{10}$ molecules cm$^{-3}$ s are listed as an example in Table S5. The most abundant HOM
products were also shown in stack in Figure 2, whose relationships with OH exposures are
superimposed by a gamma function ($f(x) = ax^m e^{-x}$) simulation line to guide the eyes. The
sum of normalized HOM monomers' abundance increased monotonically up to the highest OH
exposure of $5\times10^{10}$ molecule cm$^{-3}$ s, whereas those of HOM dimers showed a non-monotonic
dependence on OH exposure. The observed faster increase of accretion products than that of
HOM monomers can be explained jointly by the fast second-order kinetics for accretion
reactions of $RO_2$ (Berndt et al., 2018b) and the high concentrations of relevant radicals in this
work. On the other hand, most of the first-generation HOM dimers formed from accretion
reactions contain at least one C=C bond and have more functionalities than HOM monomers,
and thus should be more reactive to OH radicals, which, together with a faster deposition loss
of dimers, results in a faster consumption of HOM dimers than monomers in the OFR. The
faster production and consumption of HOM dimers allowed their concentrations to summit at
middle levels of OH exposures. As stated in Section 3.1, because of the inherent disadvantage
of laboratory experiments, $[RO_2]$ is always too high in the OFR, which has been pointed out in
a previous study (Bianchi et al., 2019). The accretion reactions in the OFR are relatively more
significant than it should be in the ambient atmosphere. We do not mean to compare the
abundance of HOM monomer and HOM dimer crossly here, but to pay attention to the
molecular characterization.
**(a)**

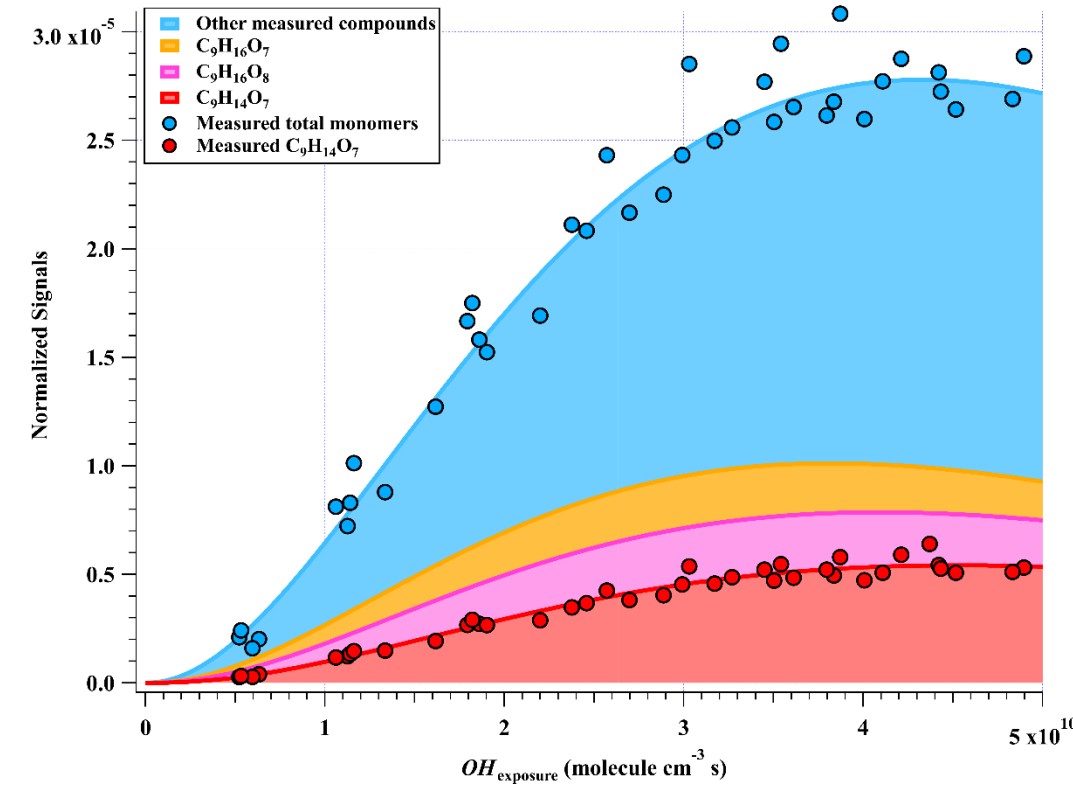


**(b)**

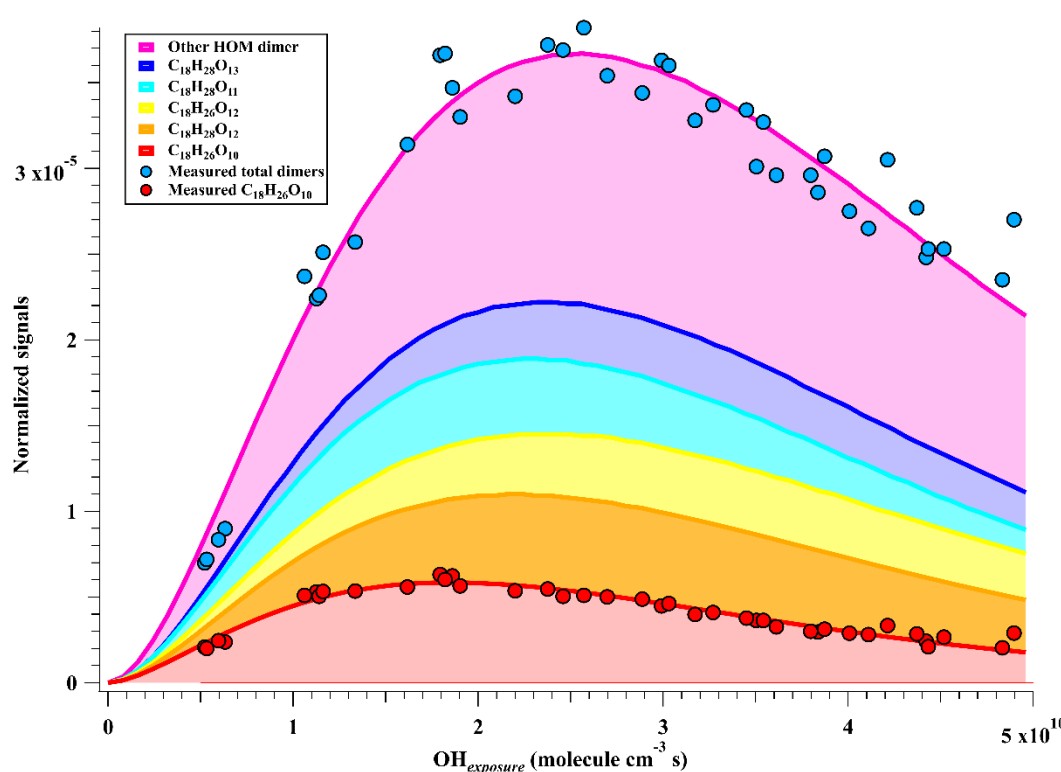


**Figure 2.** Normalized signals of (a) HOM monomers and (b) HOM dimers versus OH exposure
in the high [OH] experiments, which are fitted via a gamma function and shown in stack.

Theoretically, at a given RH and UV (i.e., a given [OH]), an increase in the initial TMB

would lead to formation of more $RO_2$, which corresponds to a larger $RO_2/OH$. However, under
our high [OH] experimental conditions, the $RO_2/OH/HO_2$ channels of $RO_2$ radicals are always
minor, and thus an increase in $RO_2/OH$ would not have a significant impact on the relative
distribution of products formed from these channels. We compared product MS for experiments
with a similar OH exposure but different initial concentrations of TMB (e.g., Exp. 1-3 v.s. Exp.
1-19, and Exp. 1-12 v.s. Exp. 1-22). The OH exposures of Exp. 1-3 and Exp. 1-19 were
estimated by the consumption of precursors to be $5.2 \times 10^9$ and $5.3 \times 10^9$ molecule $cm^{-3}$ s,
respectively, but the initial concentration of TMB of Exp. 1-3 was 25% more than that in Exp.
1-19. Meanwhile, the OH exposures of Exp. 1-12 and Exp. 1-22 were $4.5 \times 10^{10}$ and $4.4 \times 10^{10}$
molecule $cm^{-3}$ s, respectively, but the initial concentration of TMB of Exp. 1-12 was 48% more
than that in Exp. 1-22. Figure S2 shows comparisons between the product MS of Exp. 1-3 and
Exp. 1-19, as well as of Exp. 1-12 and Exp. 1-22, indicating that increase in the initial
concentration of precursors generally resulted in a minor increment in the absolute signals of
HOMs. Clearly, the relative distributions of products in these experiments are quite similar,
indicating a minor difference in the relative distributions of products caused by fluctuations of
initial concentrations of TMB.
3.2.1 HOM monomers

Previous studies indicate that oxidation products derived from the peroxide-bicyclic

pathway represent a main fraction of HOMs (Wang et al., 2017; Zaytsev et al., 2019). For 1,3,5-
TMB, this pathway, as recommended by MCM, starts from a BPR, $C_9H_{13}O_5\cdot$ (MCM name:
TM135BPRO2) (Molteni et al., 2018). According to MCM and Molteni et al. (2018), Scheme
1 has been proposed to provide a good understanding of this reaction system and the structures
of oxidation products. Molteni et al. (2018) suggested that $C_9H_{13}O_7\cdot$, i.e., peroxy radical formed
from autooxidation of $C_9H_{13}O_5\cdot$ has two isomers. A second-step of endo-cyclization is required
in the formation of one of the isomer, which is extremely slow and not competitive as shown
in several previous studies using both experimental and theoretical approaches (Wang et al.,
2017; Xu et al., 2020). Even if such a second $O_2$ bridging to a double bond is assumed to be
possible, the abundance of this isomer should be significantly smaller than the other one,
because of the much faster reaction rate of H-shift reaction. Therefore, we do not take the
$C_9H_{13}O_7\cdot$ isomer containing a double endo-cyclization into consideration in this work. The
majority of HOM monomers is generated from subsequent reactions of $C_9H_{13}O_5\cdot$ and newly
formed $C_9H_{13}O_7\cdot$, both of which contain one C=C bond in the carbon backbone and thus have
a feasible site for OH addition. Meanwhile, the autoxidation reaction rate for newly formed
$C_9H_{13}O_7\cdot$ should be significantly smaller than $C_9H_{13}O_5\cdot$, as there is no hydrogen atom in
$C_9H_{13}O_7\cdot$ that is able to undergo a hydrogen atom shift at an appreciable rate based on our
current understanding. Therefore, the subsequent autoxidation reaction should not be able to
generate large amounts of more oxidized $RO_2$.
Monomeric termination products of BPR, as shown in Scheme 1, were not detected by
nitrate CIMS in this round of experiments, which might be due to the fast sub-sequential OH
oxidation of these products under high [OH] environment since they were observed under low
[OH] environments as shown in Section 3.3. Monomeric termination products of $C_9H_{13}O_7\cdot$ were
all observed clearly, including $C_9H_{12}O_6$, $C_9H_{14}O_6$, and $C_9H_{14}O_7$. Especially, $C_9H_{14}O_7$ was the
most abundant one among all of the HOM monomer products (Figure 2a). As proved by a
previous study, these three species should be typical first-generation stabilized products derived
from autoxidation (Wang et al., 2020b). These HOM monomers should consist of several
isomers bearing the same formula, because products from the secondary reactions cannot share
the same structure as that of the one from the first-generation reaction. However, limited by the
inherent disadvantages of mass spectrometers, we could not distinguish isomers here and
further illustrate their different chemical behaviors.

**Scheme 1.** Oxidation pathways of the bicyclic peroxy radical $C_9H_{13}O_5\cdot$ (MCM name: TM135BPRO2) in the OH-initiated oxidation of 1,3,5-TMB. Green, blue, and black formulae denote alkyl peroxy radicals, alkoxy radicals and stabilized products, respectively. Black arrows denote the autoxidation pathway. MCM names for $HO_2$- and $RO_2$-termination products of TM135BPRO2 are present.

In addition to these three ones, the next most prominent products to $C_9H_{14}O_7$ were $C_9H_{16}O_7$ and $C_9H_{16}O_8$ (Figure 3a), which are produced from multi-generation oxidation according to their DBE. Based on the formulae of these three HOM monomers, they ($C_9H_{14}O_7$, $C_9H_{16}O_7$, and $C_9H_{16}O_8$) could be formed from the bimolecular termination reactions of $C_9H_{15}O_8\cdot$, which can be generated by an OH attack to $C_9H_{14}O_5$ (Scheme 2), the hydroperoxyl termination product of the BPR, $C_9H_{13}O_5\cdot$. The other HOM monomers characterized with high signals were $C_9H_{14}O_8$ and $C_9H_{16}O_9$ (Figure 3b). These two HOM monomers ($C_9H_{14}O_8$ and $C_9H_{16}O_9$), together with $C_9H_{16}O_8$, correspond to the monomeric termination products of $C_9H_{15}O_9\cdot$, which is highly likely

the peroxy radical generated by an OH attack to $C_9H_{14}O_6$ (Scheme 3), i.e., the hydroxyl
termination product of $C_9H_{13}O_7\cdot$. As discussed earlier, $C_9H_{13}O_7\cdot$ is a typical autoxidation
reaction product of the BPR of $C_9H_{13}O_5\cdot$. Therefore, detected signals of $C_9H_{16}O_8$ should be the
sum of two isomers' signals at least. Other HOM monomers were generally observed at much
lower signals and thus were not plotted individually.
**(a)**

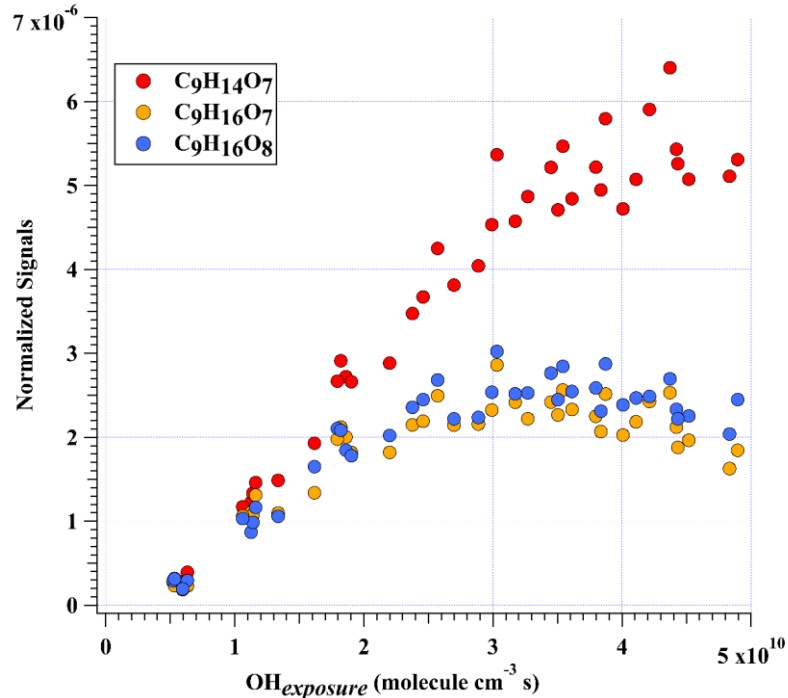


**(b)**

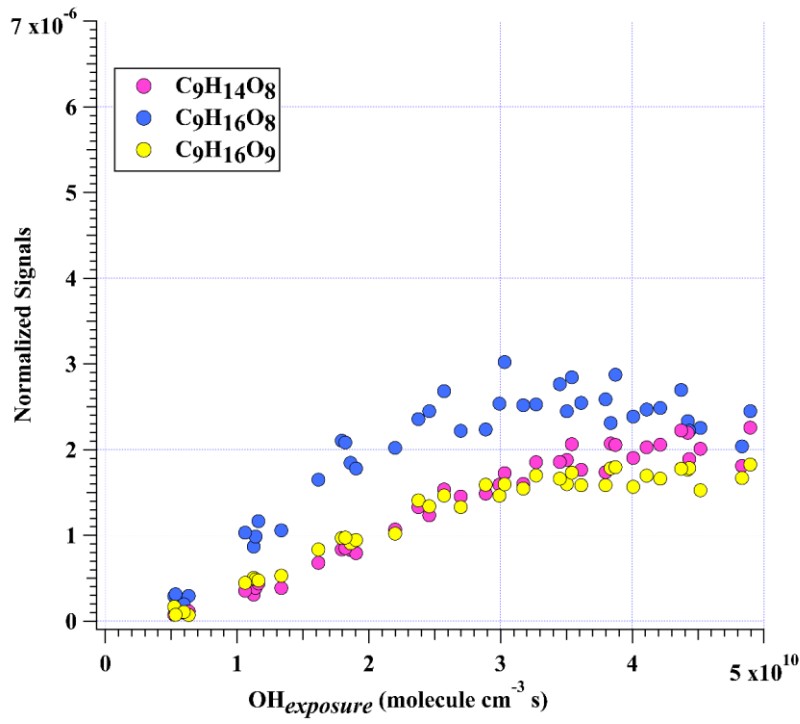


**Figure 3.** Normalized signals of (a) $C_9H_{14}O_7$, $C_9H_{16}O_7$, and $C_9H_{16}O_8$ and (b) $C_9H_{14}O_8$, $C_9H_{16}O_8$, and $C_9H_{16}O_9$ measured at the exit of OFR in our high [OH] experiments without $NO_x$ as a function of OH exposure. $C_9H_{16}O_8$ are shown in both plots to better illustrate the chemical profiles of different compound groups.


**Scheme 2.** Proposed formation pathways of $C_9H_{14}O_7$, $C_9H_{16}O_7$, and $C_9H_{16}O_8$ via the secondary OH oxidation of TM135BPOOH.


**Scheme 3.** Proposed formation pathways of $C_9H_{14}O_8$, $C_9H_{16}O_8$, and $C_9H_{16}O_9$ via the secondary
OH oxidation of TM135BPOOH.
It is worth noting that HOM monomers with 18 hydrogen atoms, i.e., a DBE of 1, were
never observed in our experiments, including a potential stabilized hydroperoxyl products
formed from $C_9H_{17}O_m\cdot$. This is expected, since $C_9H_{17}O_m\cdot$ should be in really low concentrations,
if ever existed. As indicated by its hydrogen number, a $C_9H_{17}O_m\cdot$ was formed by at least two
OH additions to the C=C bond of a $C_9H_{13}O_m\cdot$, but the main BPR, $C_9H_{13}O_5\cdot$, and its autoxidation
product ($C_9H_{13}O_7\cdot$), are characterized with one C=C bond on the ring, which makes this
formation pathway impossible. Other ring-breakage pathways should not contribute to the
formation of this radical ($C_9H_{17}O_m\cdot$) because of their low branching ratio as determined by
recent studies (Zaytsev et al., 2019; Xu et al., 2020).
Proposed according to MCM and Molteni et al. (2018), scheme 4 shows the NO
termination pathways of the main BPR $C_9H_{13}O_5\cdot$ and its autoxidation product, $C_9H_{13}O_7\cdot$. After
introducing $N_2O$ into PAM OFR, quantities of organonitrates were generated, including both
C9 and C18 organonitrates. The averaged mass spectrometry of nitrate CIMS in the $4.41\times10^{10}$
molecule cm$^{-3}$ NO experiment and $1.18\times10^{11}$ molecule cm$^{-3}$ NO experiment is shown in Figure
S3. Organonitrates were formed via the NO + $RO_2$ reaction, called as NO termination reactions.
The distribution of oxidation products under these two NO settings were similar.

**Scheme 4**. NO termination reactions of the bicyclic peroxy radical $C_9H_{13}O_5\cdot$ (MCM name: TM135BPRO2) and its autoxidation reaction products. Green, blue, and black formulae denote alkyl peroxy radicals, alkoxy radicals and stabilized products, respectively. Black arrows denote the autoxidation pathway. MCM names of NO-termination products of TM135BPRO2 are present.

As discussed above, most of the first-generation HOMs should contain a C=C bond in the carbon backbone. The ubiquitous existence of organonitrates that contain two nitrogen atoms exactly confirms the extensive secondary OH oxidation in the systems, because the NO termination reaction of $RO_2$ is the only pathway that can generate organonitrates in our experiments and this pathway can only introduce one nitrogen atom at a time, as indicated in Scheme 4. $RO_2$ can react with $NO_2$ to form peroxynitrates ($ROONO_2$) but these species are

thermally unstable except at very low temperatures or when the $RO_2$ is an acylperoxy radical (Orlando and Tyndall, 2012), neither of which were not met in our experiments. The concentrations of $NO_3$ were estimated to be lower than $2.45 \times 10^7$ molecule $cm^{-3}$ by our modified PAM_chem_v8 because of the existence of decent concentrations of NO, which would consume $NO_3$ at a rapid reaction rate, i.e., $2.7 \times 10^{-11}$ molecule$^{-1}$ $cm^3$ $s^{-1}$ (IUPAC dataset , https://iupac-aeris.ipsl.fr, last access: 26 October 2023). Therefore, $NO_2$ and $NO_3$ were not likely to react with $RO_2$ to form large amounts of organonitrates in our experiments. Taking the most abundant organonitrate, $C_9H_{14}N_2O_{10}$, as an example, it was exactly the NO termination product of $C_9H_{14}NO_9\cdot$, which was generated from an OH attack and a subsequent $O_2$ addition to $C_9H_{13}NO_6$, the NO termination product of $C_9H_{13}O_5\cdot$. For other organonitrates, $C_9H_{13}NO_8$, the second most abundant organonitrate, could be either a NO termination product of $C_9H_{13}O_7\cdot$ or, together with other most abundant organonitrates, $C_9H_{15}NO_7$ and $C_9H_{15}NO_8$, classical termination products of $C_9H_{14}NO_9\cdot$. $C_9H_{14}N_2O_{10}$, $C_9H_{15}NO_7$, and $C_9H_{15}NO_8$ all have a DBE of 2 lower than the precursor and thus are the typical multi-generation OH oxidation products.

The NO:$RO_2$ ratio in the PAM OFR in this series of experiments is lower than typical values in the ambient atmosphere, which is due to the existence of $O_3$ that was utilized to generate $O(^1D)$ in the OFR and its rapid reaction rate with NO. However, due to rapid reaction rate constants between NO and $RO_2$, i.e., around $8.5 \times 10^{-12}$ molecule$^{-1}$ $cm^3$ $s^{-1}$, the reaction rate for the NO termination channel of $RO_2$ was as fast as around $0.3 - 1.0$ $s^{-1}$. Large amounts of organonitrates would still be formed, as discussed in Section 3.1. Our conclusion is also valid because of detection of compounds with multiple nitrogen atoms.

### 3.2.2 HOM dimers

Accretion reaction $RO_2 + RO'_2 \rightarrow ROOR' + O_2$ is a source of gas-phase dimer compounds from highly oxidized, functional $RO_2$ radicals (Ehn et al., 2014; Berndt et al., 2018b; Zhao et al., 2018; Berndt et al., 2018a). $C_{18}H_{26}O_8$ and $C_{18}H_{26}O_{10}$ are two typical accretion reaction products in the 1,3,5-TMB + OH system, whose formation pathways have been elucidated (Berndt et al., 2018b). $C_{18}H_{26}O_8$ can only be formed via the accretion reaction of two $C_9H_{13}O_5\cdot$. $C_9H_{13}O_3\cdot$ is not likely to react with $C_9H_{13}O_7\cdot$ to form large amounts of $C_{18}H_{26}O_8$. $C_9H_{13}O_3\cdot$ can only be formed after addition of a hydroxyl radical to the aromatic ring of 1,3,5-TMB and a subsequent $O_2$ addition to the newly formed hydroxyl-substituted cyclohexadienyl radical (Vereecken, 2019). However, the lifetime of this radical is extremely short, as $C_9H_{13}O_3\cdot$ will undertake a ring-closure reaction and get attached by a $O_2$ very rapidly, forming BPR, $C_9H_{13}O_5\cdot$. Its short lifetime and low concentration, as indicated by Berndt et al. (2018), lead to its insignificant role in the accretion reactions. In contrast, $C_{18}H_{26}O_{10}$ can be formed either by the accretion reaction between $C_9H_{13}O_5\cdot$ and $C_9H_{13}O_7\cdot$ or via a second OH attack to $C_{18}H_{26}O_8$.

These two HOM dimers are so far the only ones that are confirmed to be formed via the
accretion reactions (Berndt et al., 2018b; Bianchi et al., 2019).
$C_{18}H_{26}O_{10}$ was characterized with the highest dimer signals for experiments with OH
exposures under $3.5 \times 10^{10}$ molecule cm$^{-3}$ s. Nevertheless, $C_{18}H_{26}O_{10}$, together with $C_{18}H_{28}O_{12}$,
$C_{18}H_{26}O_{12}$, $C_{18}H_{28}O_{11}$, $C_{18}H_{28}O_{13}$, and $C_{18}H_{28}O_{10}$ contributed more than 50% of total HOM
dimer signals at any OH exposure levels (Figure 2b). These six most abundant HOM dimers
correspond exactly to the hydroperoxyl, hydroxyl, and carbonyl termination products of
$C_{18}H_{27}O_{11}\cdot$ and $C_{18}H_{27}O_{13}\cdot$, respectively. These two RO$_2$ ($C_{18}H_{27}O_{11}\cdot$ and $C_{18}H_{27}O_{13}\cdot$), on the
other hand, could be generated by OH attacks to $C_{18}H_{26}O_8$ and $C_{18}H_{26}O_{10}$, respectively, which
strongly suggests the significant role of secondary OH chemistry in the formation of HOMs in
our experiments. In addition, $C_{18}H_{28}O_x$ can also be formed through accretion of a $C_9H_{13}O_m\cdot$
radical and a $C_9H_{15}O_m\cdot$ radical, as suggested by previous studies (Molteni et al., 2018;
Tsiligiannis et al., 2019). However, since a $C_9H_{15}O_m\cdot$ radical, as suggested by its hydrogen atom
number, can only be formed via an OH addition to the stabilized $C_9H_{14}O_m$ products through
multi-generation OH reactions, our conclusion that $C_{18}H_{28}O_x$ are multi-generation OH
oxidation products still holds. Figure 4 shows the normalized signals of these abundant HOM
dimers at different OH exposures.
**(a)**

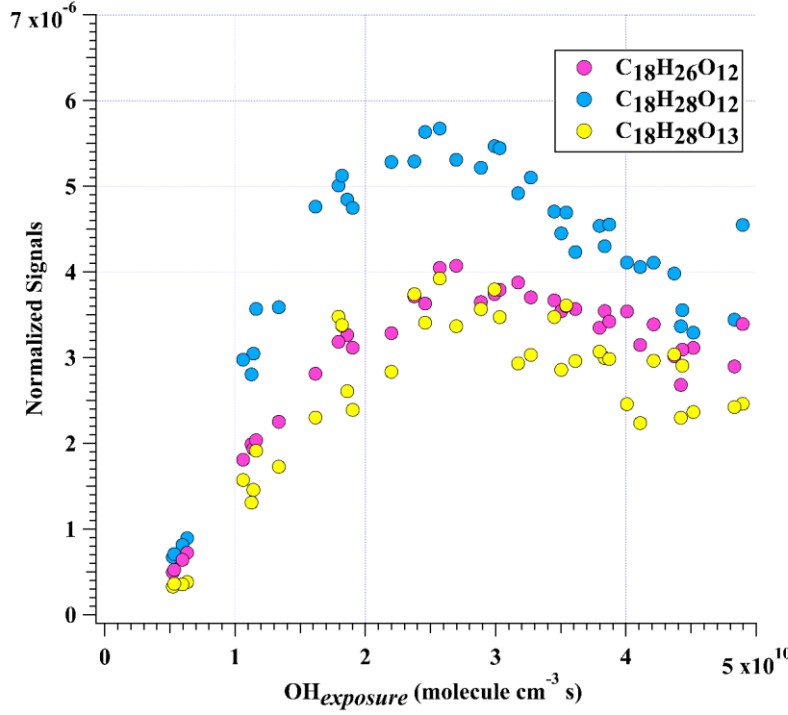


**(b)**

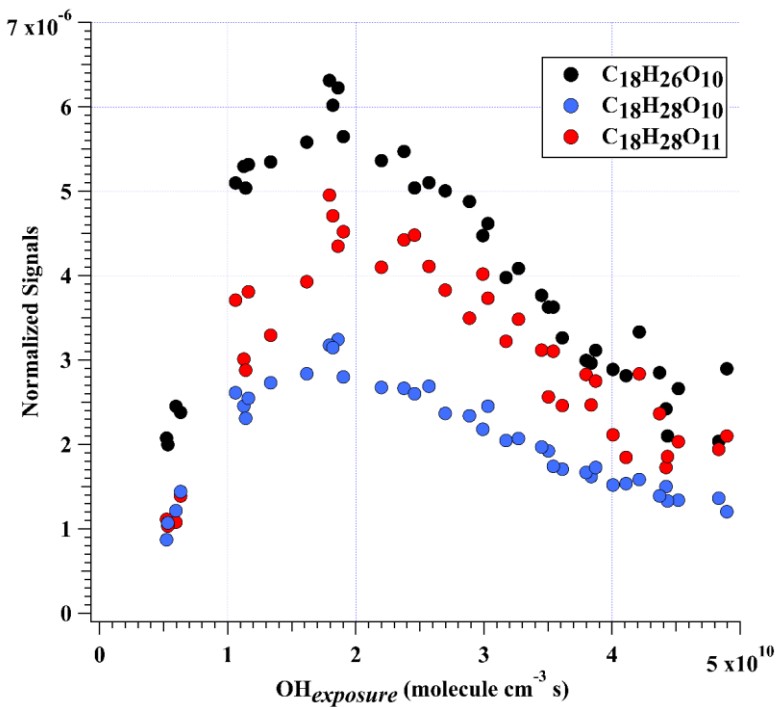


**Figure 4.** Normalized signals of (a) $C_{18}H_{26}O_{12}$, $C_{18}H_{28}O_{12}$, and $C_{18}H_{28}O_{13}$, and (b) $C_{18}H_{26}O_{10}$, $C_{18}H_{28}O_{10}$, and $C_{18}H_{28}O_{11}$ measured at the exit of OFR in our high [OH] experiments without $NO_x$ as a function of OH exposure.

This decrease of dimer at relatively high OH exposures are likely due to the accelerated
accretion reactions in the OFR, resulted by the high $RO_2$ concentrations. The HOM dimers are
formed earlier compared to under ambient conditions and then can go through the further
oxidation reactions. Note that this does not mean the maximum concentrations of HOM dimers
will also accurately occur at the same OH exposures in the atmosphere, because the detailed
appearance time of the maximum concentrations of HOM dimers is dependent on their
formation rate and loss rate. In our experiments, the formation rate and loss rate were not
accelerated equally. On the other hand, the loss pathways of HOM dimers were not exactly the
same as the ambient. This series of experiments are not meant to specifically find out the
detailed OH exposures when the maximum concentrations of HOM dimers will occur, but try
to indicate how HOM dimers evolve with the increase of OH exposures. This work can be
regarded as an indicator for the potential chemical fates of HOM dimers in the atmosphere if
their survival time permitted. It should be noted that the gas-phase chemistry in the PAM OFR
cannot be exactly the same as that in the ambient. Reactions of OH with OVOCs often lead to
$HO_2$ formation, resulting in a $HO_2:RO_2$ ratio larger than 1 in the real atmosphere (Bianchi et al.,
2019). A recent campaign conducted at a rural site in the Yangtze River Delta estimated that
the local ratio of $HO_2:RO_2$, the latter of which was presumably derived from longer chain
alkanes (> $C_3$), alkenes, and aromatic compounds, was around 1.66 (Ma et al., 2022). Such a
high $HO_2:RO_2$ ratio condition is typically difficult to be simulated in the laboratory experiments,
as the precursors are usually hydrocarbons without any OVOCs (Peng and Jimenez, 2020). This
is exactly the case for our experiments, but its influences on our conclusion were tiny, as have
been discussed in the Section 3.1. Therefore, the difference in the distribution of products will
not change our conclusion.
Such an active secondary OH chemistry is consistent with the fast OH reaction rates of
HOMs. We take $C_{18}H_{26}O_8$ whose plausible structure is shown in Figure S4 as an example,
which is the accretion product of two $C_9H_{13}O_5\cdot$. Its OH reaction rate constant is estimated to be
around $2.07 \times 10^{-10}$ $cm^3$ molecule$^{-1}$ s$^{-1}$ according to the structure-activity relationship (Jenkin et
al., 2018b, a), whose details are provided in Supplementary Text S1. This rate is several times
larger than that of 1,3,5-TMB, which enables a very active secondary OH chemistry in the
system. MCM recommended an OH reaction rate of $1.28 \times 10^{-10}$ $cm^3$ molecule$^{-1}$ s$^{-1}$ for
TM135BPOOH ($C_9H_{14}O_5$) and $1.00 \times 10^{-10}$ $cm^3$ molecule$^{-1}$ s$^{-1}$ for TM135OBPOH ($C_9H_{12}O_4$)
(Jenkin et al., 2003). The OH reaction rate for $C_{18}H_{26}O_8$ should also be fast due to the C=C
bonds in its structure, which is activated by the adjacent functionalities. Our calculation result
is consistent with this estimation.
The distributions of C18 organonitrates also verified the extensive secondary reactions.
The most abundant C18 organonitrate, $C_{18}H_{27}NO_{12}$ was a NO termination product of radical
$C_{18}H_{27}O_{11}\cdot$, which, as mentioned above, was the radical generated from the OH reaction with
$C_{18}H_{26}O_8$. $C_{18}H_{27}NO_{12}$ can also be formed either by accretion between a $C_9H_{15}O_m\cdot$ radical and
a $C_9H_{12}NO_m\cdot$ radical or accretion between a $C_9H_{13}O_m\cdot$ radical and a $C_9H_{14}NO_m\cdot$ radical. Both
$C_9H_{15}O_m\cdot$ and $C_9H_{14}NO_m\cdot$ radicals are a typical multi-generation $RO_2$ and thus prove
$C_{18}H_{27}NO_{12}$ is a multi-generation OH oxidation product. Other C18 organonitrates are believed
to be formed in a similar pathway. Hence, plenty of organonitrates have been formed via the
multi-generation OH reactions of first-generation stabilized products.
**3.3 Oxidation products in low [OH] experiments**
Given the larger sampling port, lower initial ozone concentrations, lower UV light
intensities, and a better performance of mass spectrometer in this series of low [OH]
experiments, a number of new species were detected in the 2$^{nd}$-round experiments, including
three typical termination reaction products of BPR, i.e., $C_9H_{14}O_4$, $C_9H_{14}O_5$, and $C_9H_{13}NO_6$, and
a number of low volatile compounds, e.g., $C_9H_xO_{11}$ ($x = 12 - 15$). The distributions of oxidation
products detected by nitrate CI-TOF in Exp. 2-3, 2-4, and 2-7, representative low [OH]
experiments, are displayed in Figure 5. The detailed molecular formula and their contributions
to total HOMs signals are provided in Tables S6 and S7.
In addition, certain C9 and C18 HOMs with lower DBE than typical first-generation
products predicted by MCM (Saunders et al., 2003) or reported by previous studies (Berndt et
al., 2018b), were detected in Exp. 2-3, 2-4, and 2-7, although [OH] in these experiments are
much lower than those in the $1^{st}$-round experiments.
Observation of compounds with lower DBE in Exp. 2-3, 2-4, and 2-7 including HOM
monomers with DBE lower than 3 and HOM dimers with DBE lower than 6, as well as
monomer radicals with DBE lower than 3 including $C_9H_{15}O_m\cdot$ ($m = 7 - 11$) and $C_9H_{14}NO_9\cdot$,
proves the re-initiation of OH oxidation of the stabilized products in experiments with
atmospheric relevant [OH]. All the stabilized products and radicals depicted in the proposed
mechanisms (Scheme 2 and Scheme 3) were detected in both Exp. 2-3 and Exp. 2-4, except for
$C_9H_{15}O_9\cdot$ that was only detected in Exp. 2-3. This means that the proposed reaction pathways
have already happened under atmospheric [OH] conditions with limited OH exposures.
However, as we do not know the exact structures of these OOMs and radicals, the proposed
reaction pathways are merely based on the chemical formulae detected by nitrate CIMS and
nitrate CI-TOF and proposed according to the general mechanisms of OH addition reactions to
the C=C bond. Other reaction pathways to generate these compounds or other isomers
generated in these pathways are undoubtedly feasible.
A lot of compounds detected in the experiments without $NO_x$ were not observed in the
counterpart experiments with $NO_x$. We also did not detect decent signals of HOM dimers in the
$NO_x$-present experiments in the $2^{nd}$-round experiments. Such a dramatic decrease in the
abundance of HOM dimers after the introduction of $NO_x$ into the aromatic oxidation system
has been reported in several previous studies (Garmash et al., 2020; Wang et al., 2020b;
Tsiligiannis et al., 2019). This might come from the dominant significance of NO + $RO_2$
reactions ($R8 - R9$) after the introduction of $NO_x$ into system, making signals of certain HOMs
from other channels lower than the detection limit of the instrument. The proportions of other
reaction channels decreased, and were reassigned to the NO channel, as evidenced by the fact
that most of observed oxidation products were organonitrates, which is in an excellent
agreement with the modeled channel proportions in Section 3.1.
Many organonitrates were observed in both series of experiments. In the low [OH]
experiments, the most significant compound was $C_9H_{13}NO_8$, whose formula matches the NO
termination product of $C_9H_{13}O_7\cdot$, i.e., autoxidation product of BPR. The second most important
compound, $C_9H_{14}N_2O_{10}$ in our low [OH] experiments, was the most significant product in the
high [OH] experiments in presence of $NO_x$, whose formula matches the NO termination product
of $C_9H_{14}NO_9\cdot$, i.e., the $RO_2$ formed via an OH addition to $C_9H_{13}NO_6$, the NO termination
product of BPR. All of the products and radicals mentioned above were observed in Exp. 2-7,
as shown in Figure 5c. From the perspective of molecular formula, $C_9H_{14}N_2O_{10}$ is also one of
the most frequently observed multi-nitrogen-containing compound in polluted atmospheres,
whose seasonal variations show a good correlation with [OH] (Guo et al., 2022; Yang et al.,

2023).

A comparison of relative abundances of C9 and C18 products under different [OH] levels
is helpful for the elucidation of their formation pathways. The difference in product
distributions between Exp. 2-3 ([OH] = ~$1.69\times10^7$ molecule cm$^{-3}$) and Exp. 2-1 ([OH] =
~$1.03\times10^8$ molecule cm$^{-3}$), as well as between Exp. 2-3 and Exp. 1-12 ([OH] = ~$8.47\times10^8$
molecule cm$^{-3}$) is shown in Figure 6. The normalized abundance was obtained by normalizing
all the products to the most abundant one in each experiment, i.e., $C_{18}H_{26}O_{10}$ in Exp. 2-1 and
Exp. 2-3, and $C_9H_{14}O_7$ in Exp. 1-12. The changes in the normalized abundance were obtained
by subtracting the normalized abundance in Exp. 2-1 from that in Exp. 2-3, and Exp. 1-12 from
Exp. 2-3. As the [OH] and OH exposure increased, there was a noticeable rise in the relative
abundance of more oxygenated compounds, which can be attributed to the larger proportion of
multi-generation OH oxidation in high OH exposure experiments. This comparison
demonstrates the capacity and potential of multi-generation OH oxidation to reduce DBE and
elevate the oxygenated levels of oxidation products.
In conclusion, observation of the same low DBE compounds, i.e., DBE = 2, in both low
[OH] and high [OH] experiments confirms the feasibility of the generation of HOMs under
atmospheric relevant conditions. The detection of $C_9H_{14}O_5$, $C_9H_{15}O_8\cdot$, $C_9H_{14}O_7$, $C_9H_{14}O_8$,
$C_9H_{15}O_7\cdot$, and $C_9H_{16}O_8$, and $C_9H_{14}O_6$, $C_9H_{15}O_9\cdot$, $C_9H_{14}O_8$, $C_9H_{14}O_9$, $C_9H_{15}O_8\cdot$, and $C_9H_{16}O_9$, in
low [OH] experiments also confirms the potential existence of the proposed mechanisms, i.e.,
Scheme 2 and Scheme 3, respectively. Certainly, other potential formation pathways for these
products are possible.
**(a)**

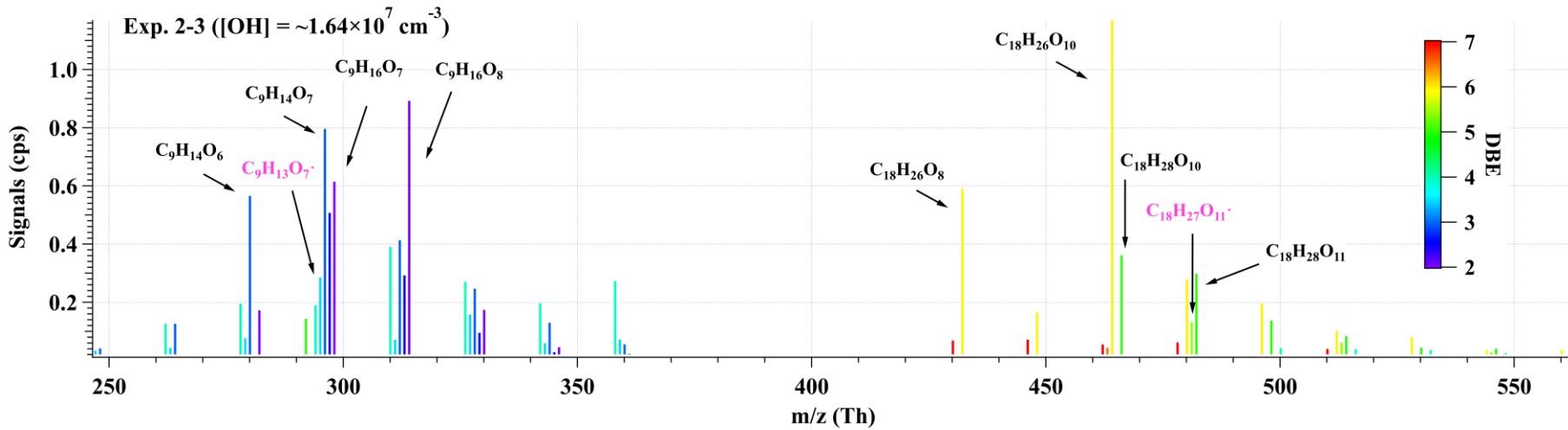


**(b)**

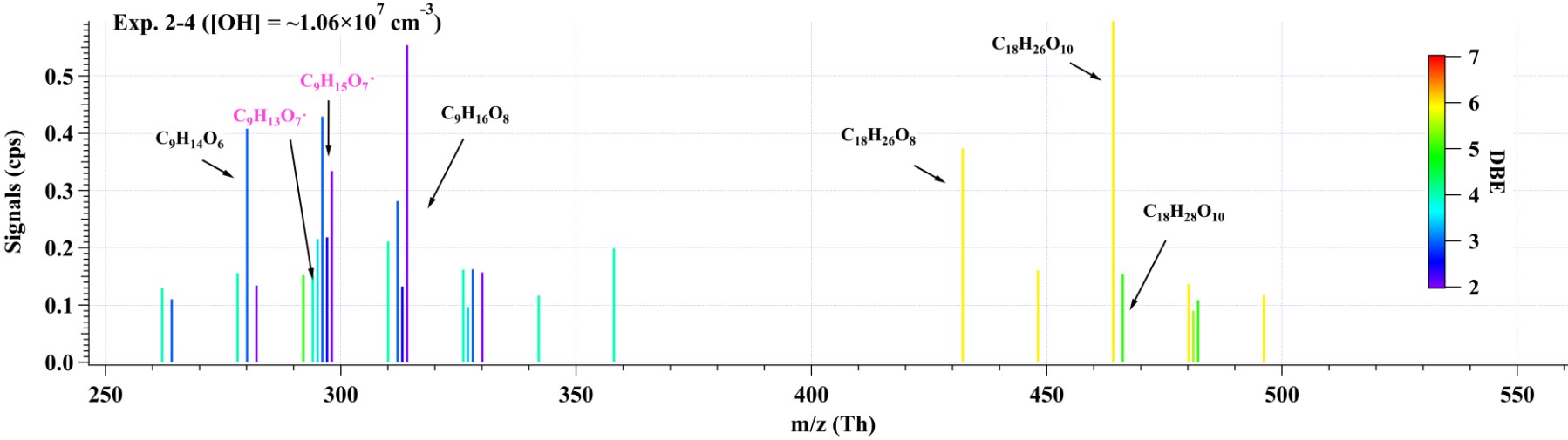

**(c)**

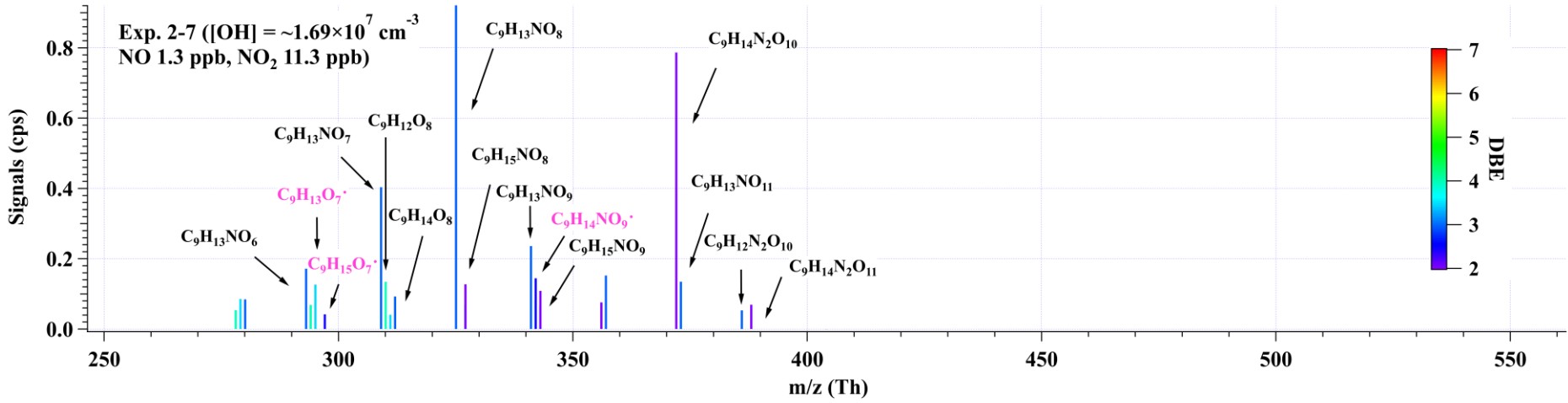



**Figure 5.** Distributions of C9 and C18 products detected by nitrate CI-TOF in (a) Exp. 2-3, (b) Exp. 2-4, and (c) Exp. 2-7. The reagent ion, $NO_3^-$, is omitted in
the label for the molecular formula. Important radicals were labelled in pink. Note that no convinced signals of HOM dimers were observed in the 2nd-round
experiments with $NO_x$.


**(a)**

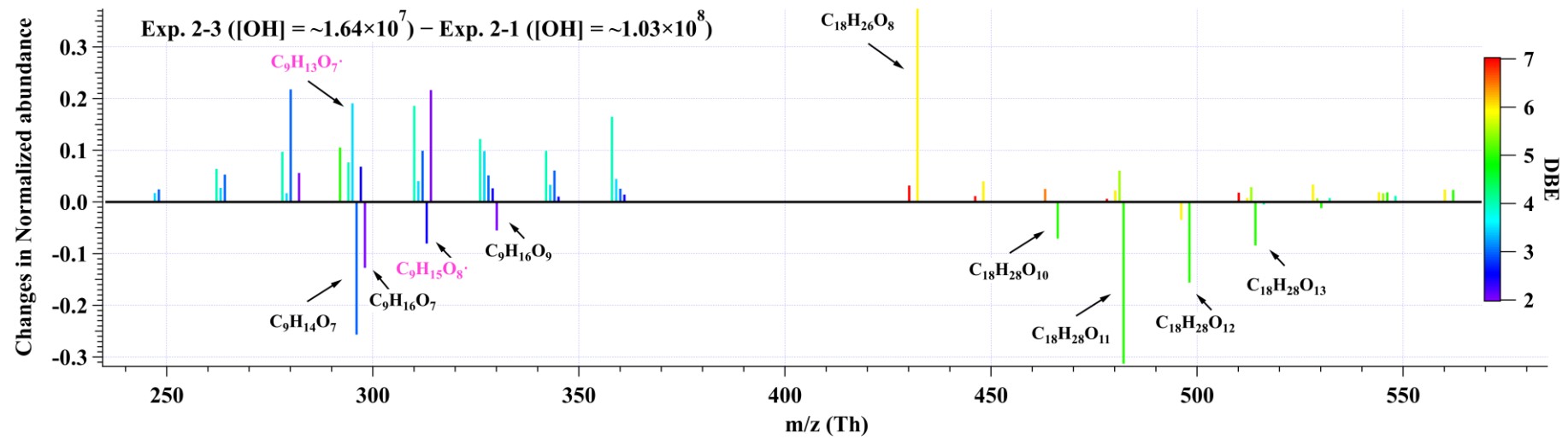


**(b)**

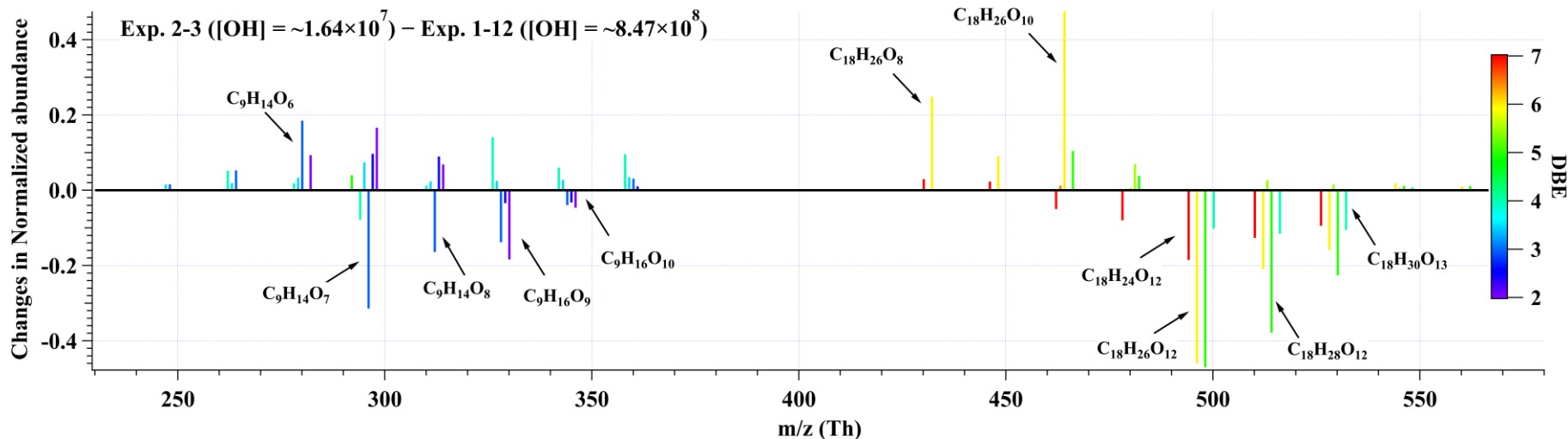


**Figure 6**. The changes in normalized abundance of C9 and C18 products observed by nitrate CI-TOF in (a) Exp.2-3 relative to Exp.2-1, and (b) Exp.2-3 relative
to Exp.1-12. The reagent ion, $NO_3^-$, is omitted in the label. The normalized abundance was obtained by normalizing all the products to the most abundant one
in each experiment, i.e., $C_{18}H_{26}O_{10}$ in Exp.2-1 and Exp.2-3, and $C_9H_{14}O_7$ in Exp.1-12.

**4 Atmospheric Implications**

This study highlights the influences of OH exposure on the distribution and evolution of 1,3,5-TMB-derived HOMs. Secondary OH reactions can influence HOMs' composition by directly reacting with the stabilized first-generation oxidation products, leading to enhanced formation of HOMs, if the stabilized, first-generation oxidation products could survive from condensation loss onto pre-existing particles. Observation of organonitrates generated in the NO experiments further confirmed the secondary OH oxidation. Due to the elevated abundance and the reduced volatility of HOMs, growth rates of newly formed nanoparticles in the presence of HOMs could be raised, especially in high-OH environments, which prevails in the summer noon. Substantially high concentrations of OH have been frequently observed in polluted environments during summer, e.g., megacities in China (Tan et al., 2019), and thus more active secondary OH reactions are expected compared to wintertime. As a plausible consequence, seasonal differences of HOMs and new particle formation (NPF) are resulted (Qiao et al., 2021; Yao et al., 2018; Guo et al., 2022). Furthermore, previous studies suggest that high concentrations of NO can suppress the formation of HOMs via the suppression of autoxidation (Pye et al., 2019), but the influences of such a suppression could have been overestimated, since secondary OH reactions can continue to oxidize the stabilized organonitrates. Our conclusions help to explain the existing gap between model prediction and ambient measurement on the HOMs concentrations (Qi et al., 2018), and to build a global HOMs simulation model.

*Data availability.* Data used in this work are available upon request from the corresponding authors.

*Supplement.* The supplement related to this article is available online.

*Author contributions.* LW and Yuwei Wang designed the experiments. Yuwei Wang and Chuang Li conducted the laboratory experiments. Yuwei Wang analyzed the data. Yuwei Wang and LW wrote the paper. All co-authors discussed the results and commented on the manuscript.

*Competing interests.* The authors declare that they have no conflict of interest.

*Acknowledgments.* This work was financially supported by the National Natural Science Foundation of China (21925601, 22127811) and National Key R&D Program of China (No. 2022YFB2602001). The authors declare no competing interests. Yuwei Wang would like to

thank Andrew T. Lambe, Peng Zhe, and Jose Jimenez for helpful discussions on PAM

experiments.

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
