# Peer review of "organic molecules lead to plentiful highly oxygenated organic"

_EGUsphere, 2023_

## Referee Comment (RC1)

Review of Egusphere submission: Secondary reactions of aromatics-derived oxygenated organic molecules lead to plentiful highly oxygenated organic molecules within an intraday OH exposure, *by Wang, Y. et al.*

**Significance**

Aromatic compound oxidation is responsible for a sizeable fraction of urban air pollution. Aromatics contribute significantly to the condensable product pool, and consequently are an important source, often even the dominant, of anthropogenic secondary organic aerosol (SOA). The autoxidation pathways to highly oxygenated organic compounds (HOM) from aromatics have puzzled the atmospheric community for around a decade and the major achievements on the topic have been published in several previous works (e.g., Wang et al., 2017, Molteni et al., 2018, Garmash et al., 2020 etc.). The current work aims to add to this by studying 1,3,5-trimethylbenzene (mesitylene) in an oxidation flow reactor setup using far above ambient OH concentrations, attempting to simulate long atmospheric oxidation timescales.

While the research performed is certainly timely, it has been performed with methodology that unrealistically biases the oxidation conditions, and thus prevents gaining the sought after mechanistic insight. While the used oxidation flow reactor approach has its merits in developing emission regulations, it is not a platform for studying detailed molecular level oxidation chemistry of atmospheric relevant condensable product formation. As such, I can only recommend rejecting the work with its current analysis and conclusions.

Below I detail why I feel the work is performed with inadequate research methodology, and I'll point out several issues I hope the authors will pay attention in preparing the next draft for submission.

**Major comments**

Unfortunately, the PAM OFR setup used in the current work with very high [OH] is not suitable for studying mechanistic details of atmospheric oxidation, perhaps even less of aromatic compound oxidation where the sequential OH oxidation and photo-oxidation of intermediates and products is important. The PAM methodology has been constructed to allow estimating the potential aerosol mass from a given emission, and it is really a method aiming for emission regulations, rather than molecular level mechanistic details. The high OH concentrations lead to unrealistically high primary radical concentrations and skew the reaction system towards very rapid RO2 + OH pathways. The design also necessarily leads into higher RO2 + RO2 rates favoring accretion product formation, but also radical propagation channels by RO formation. Additionally, the formation rate of closed-shell species is accelerated allowing for more efficient sequential OH oxidation. According to the presented results, even oxidation of the accretion products is possible that commonly would be expected to contribute to the growing aerosol, and not be lost in chemical degradation by reactions with oxidants. In the atmosphere it really matters what is the correct reaction timescale, and thus the order of the sequential reactions. Hence, it's difficult to see how a PAM type setup could be used to study mechanistic details of atmospheric oxidation chains.

So, once again, PAM was constructed to enable making emission regulations, and not for studying details of atmospheric chemistry, though several groups have seemed to adopt it for

such a purpose recently. PAM is by design non-linear in oxidation chemistry regime and is thus not capable for detailing the molecular mechanisms. As the Authors also confess, the autoxidation pathways are the most important at low loadings, when processes like RO2 + RO2, and RO2 + OH, are suppressed. The timing and order of reactions happening in a sequential oxidation do make a big impact.

**Further comments I hope will help in sketching the next draft.**

What is the influence of aromatic photochemistry in your PAM setup? Aromatics are known to strongly absorb light at relatively long wavelengths, and the oxygenated aromatics even more (see e.g., https://www.uv-vis-spectral-atlas-mainz.org/uvvis/), so I'm wondering how was the relevance of the used light sources tested in this work? This is not irrelevant for aromatic oxidation.

You used a relatively long ¼ inch Teflon sampling tube for the CIMS. This is the smallest tube diameter I've ever come across with nitrate CIMS sampling. One would expect the HOM losses, especially the most oxygenated ones, to be very significant in this tube. Nevertheless, HOM with high O-content seems to be detected with this setup too!

Note that ELVOC would rarely be expected to nucleate by itself, and LVOC basically never.

Jenkin 2003 reference does not have autoxidation.

The autoxidation reaction of BPR by H-abstraction has been found relatively slow by Wang et al 2017, not rapid.

Several of the products detected seem to have worryingly many H-atoms in the structures. Especially the C9H17Om radicals.

How well does the relatively low NO with the high RO2 simulate atmospheric NOx chemistry?

"On the other hand, the structure of resulting CxH2x-6O7·is strongly different from that of BPR,". → Do you mean the rings are broken?

" Such a slow autoxidation reaction rate cannot explain the extensive existence of HOM monomers with more than 7 oxygen atoms and HOM dimers with more than 10 oxygen atoms, which are the maximum numbers of oxygen atoms in stabilized monomer and dimer products, respectively, formed from CxH2x-6O7·(Mentel et al., 2015; Molteni et al., 2018; Wang et al., 2020)."
→ There's a recent paper from my group that could provide an explanation what is observed here: https://www.nature.com/articles/s41467-023-40675-2.

I find it confusing to draw the "double-peroxide-ring" pathways in Schemes 1 and 2, if you even explicitly mention that they are unlikely. I advise to remove them, and the text " Another possibility is the formation of a second oxygen bridge after the hydrogen shift of BPR (Molteni et al., 2018)," altogether.

"with an OH exposure equivalent to 2.4 – 19.4 days of atmospheric photochemical ageing. Certainly, such extremely high OH exposures favor secondary OH chemistry and help to facilitate our understanding on product distributions"

→ I would argue it doesn't, except for PAM conditions. As explained above, it does matter at what order and rate different oxidation steps happen in the atmosphere, and using such a high OH doses seem to necessarily skew up the chemistry. Figure 1 seems to be a good indication of this, as the "dimers" are generated faster than the monomers, and at the higher OH dose even the sum of "dimers" decrease.

"Indeed, laboratory experiments show that RO2 formed during the second-generation OH oxidation of the first-generation stabilized oxidation products can also undergo autoxidation reactions,"

→ This is extremely natural, as autoxidation is 'auto-catalytic oxidation' and mainly enabled by the loosening of the adjacent H-atoms next to the gained functional groups. Autoxidation inherently accelerates in many, if not all, chemical systems.

"High atmospheric concentrations of OH"

→ What is high atmospheric concentration to you? In the atmosphere [OH] is mostly buffered by [CO] and [CH4].

Figure S4 has a good idea but is difficult to read with such a small scale.

Was the aromatic sample illuminated with the same light source that was used for N2O photolysis? If so, then the influence of photochemistry is likely important for the results obtained.

You make a point that estimating HOM penetration through the system to the detector is difficult to quantify, yet it seems your calculations assume that 1,3,5-TMB and HOMs have similar losses in the system. This does not seem reasonable. How does this then influence the determined "nominal relative molar yields of HOMs"?

What do you mean by increase being monotonic or non-monotonic?

Almost all the monomeric termination products in Scheme 1 have two strong H-bonding functional groups (i.e., -OH and -OOH), and thus would be expected to be seen with nitrate ion charging (see, e.g., https://pubs.acs.org/doi/10.1021/acs.jpca.7b10015). Perhaps the proposed scheme is not correct?

"because products from the secondary reactions cannot share the same structure as that of the one from the first-generation reaction."

→ Except perhaps in recycling or regeneration reactions. However, the important bit here is that you can make isomeric products, and the mass spectrometric detection utilized here would not separate them.

"C18H26O8 can only be formed via the accretion reaction of two C9H13O5·"

→ Nope. Could be, for example, through O3 and O7 radicals as well.

I don't understand what the point of the next sentence is: "There are currently no evidences supporting that C9H15Om· radicals can participate in the formation of HOM dimers with 28 hydrogens." Why would you expect the H15 radicals behave in a unique way? But also, supposedly none of the previous studies used as high OH dose, which would explain why such products were not observed. The general observation of dimers with H28 dominating seems worrying.

It seems worrying that the dimer products decrease already at such a short reaction times. This seems to amply indicate how skewed the chemical system is and that either further chemical processing, or aerosol formation, reduced the dimer yield.

A OH:HO2 ratio is given two times although it should presumably be RO2:HO2.

Consider the part: "In addition, high concentrations of radicals might also terminate the RO2 chain earlier, which inhibits the autoxidation reactions in the PAM OFR." This is true. The RO2 lifetime is critically shortened likely inhibiting normally competitive H-shift isomerization reactions. Then consider: "However, these could only influence the distribution of oxidation products at most, and would not affect the chemical behaviors of HOMs under different OH exposures." This is not true. Both conditions favor oxidation of the aromatic parent molecule, but the same HOMs are unlikely to form under so different oxidation conditions.

"The OH reaction rate for C18H26O8 should be around twice of these values, as there are two C=C bonds in its structure. Our calculation result is consistent with this estimation."
→ This seems extremely unlikely as the indicated rate is already basically at the collision limit and the big dimer compound is sterically hindered, which would imply a lower reaction rate.

"because the NO termination reaction of RO2 is the only pathway that can generate organonitrates"
→ Why would NO3 or NO2 chemistry not form organonitrates?

A strange comment considering previous literature: "since no evidence supports that a nitrogen-containing monomeric RO2 can go through accretion reactions."

---

## Author Comment (AC1)

RE: A point-to-point response to reviewers' comments

"Secondary OH reactions of aromatics-derived oxygenated organic molecules lead to plentiful highly oxygenated organic molecules within an intraday OH exposure" by Yuwei Wang, Yueyang Li, Gan Yang, Xueyan Yang, Yizhen Wu, Chuang Li, Lei Yao, Hefeng, Zhang, Lin Wang (egusphere-2023-1702)

Dear Dr. Liggio,

We are very grateful to the helpful comments from the reviewers, and have carefully revised our manuscript accordingly. A point-to-point response to the comments, which are repeated in *italic*, is given below.

We are looking forward to your decision at your earliest convenience.

Best regards,

Lin Wang
Fudan University
lin_wang@fudan.edu.cn

**Reviewer #1**

**Significance**

*Aromatic compound oxidation is responsible for a sizeable fraction of urban air pollution. Aromatics contribute significantly to the condensable product pool, and consequently are an important source, often even the dominant, of anthropogenic secondary organic aerosol (SOA). The autoxidation pathways to highly oxygenated organic compounds (HOM) from aromatics have puzzled the atmospheric community for around a decade and the major achievements on the topic have been published in several previous works (e.g., Wang et al., 2017, Molteni et al., 2018, Garmash et al., 2020 etc.). The current work aims to add to this by studying 1,3,5-trimethylbenzene (mesitylene) in an oxidation flow reactor setup using far above ambient OH concentrations, attempting to simulate long atmospheric oxidation timescales.*

*While the research performed is certainly timely, it has been performed with methodology that unrealistically biases the oxidation conditions, and thus prevents gaining the sought after mechanistic insight. While the used oxidation flow reactor approach has its merits in developing emission regulations, it is not a platform for studying detailed molecular level oxidation chemistry of atmospheric relevant condensable product formation. As such, I can only recommend rejecting the work with its current analysis and conclusions. Below I detail why I feel the work is performed with inadequate research methodology, and I'll point out several issues I hope the authors will pay attention in preparing the next draft for submission.*

**Response:**

We are very sorry that Reviewer #1 was concerned about the experimental setup in this study, which in fact has been practiced and evaluated, in addition to by three studies that have been cited by Reviewer #1 himself several lines above (*Wang et al., 2017, Molteni et al., 2018, Garmash et al., 2020*), in a number of previous studies (e.g., Tsiligiannis et al., 2019; Cheng et al., 2021). Below we will address his concerns in details, and try to convince Reviewer #1 that results in the current methodology are applicable to oxidation mechanisms.

**Major comments**

*Q 1.1 Unfortunately, the PAM OFR setup used in the current work with very high [OH] is not suitable for studying mechanistic details of atmospheric oxidation, perhaps even less of aromatic compound oxidation where the sequential OH oxidation and photo-oxidation of intermediates and products is important. The PAM methodology has been constructed to allow estimating the potential aerosol mass from a given emission, and it is really a method aiming for emission regulations, rather than molecular level mechanistic details. The high OH concentrations lead to unrealistically high primary radical concentrations and skew the reaction system towards very rapid $RO_2$ + OH pathways. The design also necessarily leads into higher $RO_2$ + $RO_2$ rates favoring accretion product formation, but also radical propagation channels by RO formation. Additionally, the formation rate of closed-shell species is accelerated allowing for more efficient sequential OH oxidation. According to the presented results, even oxidation of the accretion products is possible that commonly would be expected to contribute to the growing aerosol, and not be lost in chemical degradation by reactions with oxidants. In the atmosphere it really matters what is the correct reaction timescale, and thus the order of the sequential reactions. Hence, it's difficult to see how a PAM type setup could be used to study mechanistic details of atmospheric oxidation chains .*

*So, once again, PAM was constructed to enable making emission regulations, and not for studying*

*details of atmospheric chemistry, though several groups have seemed to adopt it for such a purpose recently. PAM is by design non-linear in oxidation chemistry regime and is thus not capable for detailing the molecular mechanisms. As the Authors also confess, the autoxidation pathways are the most important at low loadings, when processes like $RO_2 + RO_2$, and $RO_2 + OH$, are suppressed. The timing and order of reactions happening in a sequential oxidation do make a big impact.*

**Response 1.1:**

We are very grateful for comments on our manuscript by Reviewer #1.

As stated in the manuscript, the current OH does settings were deliberately selected. Following the definition by Garmash et al. (2020), OH concentrations integrated over the corresponding residence time would define an OH dose, which is also referred as OH exposure and can be used to compare results between different systems or to those in the ambient atmosphere. We set our OH concentration as used in our experiments in order to obtain a desired OH dose, i.e., an intraday OH exposure, which fills the current gap in terms of the extent of oxidation of aromatics in previous studies that focused on HOMs. Garmash et al. (2020) and Cheng et al. (2021) both used an extremely high OH exposure, which is equivalent to atmospheric oxidation times of 6.7 hours - 10 days and 2.4 - 19.4 days at OH concentrations of $1.5 \times 10^6$ molecule cm$^{-3}$, respectively. OH exposures in our experiments, on the other hand, are roughly equivalent to atmospheric oxidation times of $0.9 – 9.2$ hours at OH concentrations of $1.5 \times 10^6$ molecule cm$^{-3}$.

Also, to avoid potential misunderstanding, we would like to note that though we used a PAM OFR to conduct our experiments, our settings were much different from the traditional settings of PAM utilized in previous investigations (e.g., Kang et al., 2007). In the traditional settings of PAM, a large OH exposure equivalent to ~ 10 days was utilized to generate amounts of aerosols to investigate potential aerosols formed by given precursors. In this study, we merely used the hardware of PAM and actually used PAM as an OFR with relatively low OH exposures.

Indeed, the main concerns raised by Reviewer #1 that important processes in OFR, i.e., photolysis, $RO_2$ isomerization, and condensation, may do not scale with OH equally, are important issues. To validate our settings, a PAM chemistry model (PAM_chem_v8), utilized widely in previous studies, is chosen with the latest updates to calculate radical profiles in our OFR (Li et al., 2015; Cheng et al., 2021; Wang et al., 2020; Mehra et al., 2020; Lambe et al., 2015, 2018; Peng and Jimenez, 2020; Lambe et al., 2017). This model is based on a photochemical box model that includes chemistry of photolysis of oxygen, water vapor, and other trace gases by the primary wavelengths of mercury lamps, and simplified VOC and $RO_2$ chemistry, but further reactions of the first-generation stabilized products and the second-generation organic radicals are not considered (Table R1, also as Table S2 in the revised manuscript). Kinetic data for this modified PAM chemistry model are obtained from the IUPAC (International Union of Pure and Applied Chemistry) dataset (https://iupac-aeris.ipsl.fr, last access: 26 October 2023) and the MCM dataset (MCM v3.3.1, https://mcm.york.ac.uk/MCM/, last access: 9 October 2023), except for those that are specifically discussed in details below. Note that the total $RO_2$ concentration is simplified to be the sum of concentrations of BPR and $C_9H_{13}O_7\cdot$ in our study.

**Table R1.** Reactions included in the modified PAM_chem_v8 under the settings with only 254 nm UV lights on. For experiments in the absence of $NO_x$, the input value of $N_2O$ is 0 and all the $NO_x$-related reactions proceed with a zero rate. $RO_2$ is the sum of BPR and $C_9H_{13}O_7\cdot$ for simplification.

| No | Reactions | Reaction rate constants/photolysis rate |
|----|-----------|------------------------------------------|

|  |  | (molecule$^{-1}$ cm$^3$ s$^{-1}$/ s$^{-1}$) |
|---|---|---|
| 1 | HO$_2$ + $h\nu$ ($\lambda$ = 254 nm) = OH + O($^1$D) | $2.63\times10^{-19}\times$flux$_{254}$ |
| 2 | O$_3$ + O($^1$D) = 2O$_2$ | $1.20\times10^{-10}$ |
| 3 | O$_3$ + O($^1$D) = O + O + O$_2$ | $1.20\times10^{-10}$ |
| 4 | O + OH = H + O$_2$ | $2.20\times10^{-11}\times e^{120/T}$ |
| 5 | O($^1$D) + H$_2$ = OH + H | $1.20\times10^{-10}$ |
| 6 | HO$_2$ + H = 2OH | $7.20\times10^{-11}$ |
| 7 | HO$_2$ + H = O + H$_2$O | $1.60\times10^{-12}$ |
| 8 | HO$_2$ + H = H$_2$ + O$_2$ | $6.90\times10^{-12}$ |
| 9 | O$_3$ + H = OH + O$_2$ | $1.40\times10^{-11}\times e^{-470/T}$ |
| 10 | N$_2$O + O($^1$D) = 2NO | $6.70\times10^{-11}\times e^{20/T}$ |
| 11 | N$_2$O + O($^1$D) = N$_2$ + O$_2$ | $4.70\times10^{-11}\times e^{20/T}$ |
| 12 | O + HO$_2$ = OH + O$_2$ | $3.02\times10^{-11}\times e^{200/T}$ |
| 13 | O + H$_2$O$_2$ = OH + HO$_2$ | $1.40\times10^{-12}\times e^{-2000/T}$ |
| 14 | O + O$_3$ = 2O$_2$ | $8.00\times10^{-12}\times e^{-2060/T}$ |
| 15 | O + NO$_3$ = NO$_2$ + O$_2$ | $1.00\times10^{-11}$ |
| 16 | O + NO$_2$ = NO + O$_2$ | $5.12\times10^{-12}\times e^{210/T}$ |
| 17 | OH + O$_3$ = HO$_2$ + O$_2$ | $1.70\times10^{-12}\times e^{-940/T}$ |
| 18 | OH + HO$_2$ = H$_2$O + O$_2$ | $4.80\times10^{-11}\times e^{250/T}$ |
| 19 | OH + HONO = H$_2$O + NO$_2$ | $1.80\times10^{-11}\times e^{-390/T}$ |
| 20 | OH + H$_2$O$_2$ = H$_2$O + HO$_2$ | $2.90\times10^{-12}\times e^{-160/T}$ |
| 21 | OH + H$_2$ = H$_2$O + H | $2.80\times10^{-12}\times e^{-1800/T}$ |
| 22 | OH + OH = H$_2$O + O | $1.80\times10^{-12}$ |
| 23 | HO$_2$ + O$_3$ = OH + O$_2$ | $1.00\times10^{-14}\times e^{-490/T}$ |
| 24 | HO$_2$ + NO = OH + NO$_2$ | $3.50\times10^{-12}\times e^{270/T}$ |
| 25 | NO + O$_3$ = NO$_2$ + O$_2$ | $2.00\times10^{-12}\times e^{-1400/T}$ |
| 26 | NO$_2$ + O$_3$ = NO$_3$ + O$_2$ | $1.20\times10^{-13}\times e^{-2450/T}$ |
| 27 | NO + NO$_3$ = 2NO + O$_2$ | $1.50\times10^{-11}\times e^{170/T}$ |
| 28 | NO$_3$ + NO$_3$ = 2NO$_2$ + O$_2$ | $8.50\times10^{-13}\times e^{-2450/T}$ |
| 29 | N$_2$O$_5$ + H$_2$O = 2HNO$_3$ | $2.00\times10^{-21}$ |
| 30 | O + O$_2$ + M = O$_3$ + M | $6.00\times10^{-34}\times$M$\times$(300/T)$^{2.4}$ |
| 31 | H + O$_2$ + M = HO$_2$ + M | $k_o = 4.40\times10^{-32}\times$M$\times$(300/T)$^{1.3}$
 $k_h = 7.50\times10^{-11}\times$(300/T)$^{0.2}$

 $k = k_o/(1+(k_o/k_h))\times0.6^{(1+(\log10(k_o/k_h))^{-2}}$ |
| 32 | OH + OH + M = H$_2$O$_2$ + M | $k_o = 6.90\times10^{-31}\times$M$\times$(300/T)
 $k_h = 2.60\times10^{-11}$

 $k = k_o/(1+(k_o/k_h))\times0.6^{(1+(\log10(k_o/k_h))^{-2}}$ |
| 33 | OH + NO + M = HONO + M | $k_o = 7.00\times10^{-31}\times$M$\times$(300/T)$^{2.6}$
 $k_h = 3.60\times10^{-11}\times$(300/T)$^{0.1}$

 $k = k_o/(1+(k_o/k_h))\times0.6^{(1+(\log10(k_o/k_h))^{-2}}$ |
| 34 | OH + NO$_2$ + M = HNO$_3$ + M | $k_o = 1.80\times10^{-30}\times$M$\times$(300/T)$^{2.6}$
 $k_h = 2.80\times10^{-11}$

 $k = k_o/(1+(k_o/k_h))\times0.6^{(1+(\log10(k_o/k_h))^{-2}}$ |
| 35 | OH + HNO$_3$ = H$_2$O + NO$_3$ | $k_{00} = 2.40\times10^{-14}\times e^{460/T}$
 $k_{01} = 6.50\times10^{-34}\times e^{2199/T}$
 $k_{02} = 2.80\times10^{-11}\times e^{-2450/T}$

 $k = k_{00} + (k_{01}\times$M$)/(1+(k_{01}\times$M$)/k_{02})$ |
| 36 | HO$_2$ + NO$_2$ + M = HO$_2$NO$_2$ + M | $k_o = 1.80\times10^{-31}\times$M$\times$(300/T)$^{3.2}$
 $k_h = 4.70\times10^{-12}\times$(300/T)$^{1.4}$

 $k = k_o/(1+(k_o/k_h))\times0.6^{(1+(\log10(k_o/k_h))^{-2}}$
 $k_{reverse} = k/(2.10\times10^{-27}\times e^{10900/T})$ |
| 37 | NO$_2$ + NO$_3$ + M = N$_2$O$_5$ + M | $k_o = 2.00\times10^{-30}\times$M$\times$(300/T)$^{4.4}$
 $k_h = 1.40\times10^{-12}\times$(300/T)$^{0.7}$

 $k = k_o/(1+(k_o/k_h))\times0.6^{(1+(\log10(k_o/k_h))^{-2}}$
 $k_{reverse} = k/(2.70\times10^{-27}\times e^{11000/T})$ |

| | | |
|---|---|---|
| 38 | OH + HNO$_4$ = products | $1.30\times10^{-12}\times e^{250/T}$ |
| 39 | Sci + H$_2$O = products | $4.00\times10^{-15}$ |
| 40 | 1,3,5-TMB + OH = BPR | $0.8\times5.67\times10^{-11}$ |
| 41 | 1,3,5-TMB + OH = Products | $0.2\times5.67\times10^{-11}$ |
| 42 | BPR = C9H13O7 | 7 |
| 43 | BPR + RO$_2$ = ROOR' | $1.70\times10^{-10}$ |
| 44 | BPR + RO$_2$ = R=O/ROH + O$_2$ | $0.4\times8.8\times10^{-13}$ |
| 45 | BPR + RO$_2$ = 2RO + O$_2$ | $0.6\times8.8\times10^{-13}$ |
| 46 | BPR + OH = RPO$_2$ + H$_2$O | $1.00\times10^{-10}$ |
| 47 | BPR + HO$_2$ = ROOH + O$_2$ | $1.20\times10^{-11}$ |
| 48 | BPR = wall loss | 0.0023 |
| 49 | BPR + NO = RO + NO$_2$ | $0.843\times8.50\times10^{-12}$ |
| 50 | BPR + NO + M = RONO$_2$+ M | $0.157\times8.50\times10^{-12}$ |
| 51 | C9H13O7 + RO$_2$ = ROOR' | $2.60\times10^{-10}$ |
| 52 | C9H13O7 + RO$_2$ = R=O/ROH + O$_2$ | $0.4\times8.8\times10^{-13}$ |
| 53 | C9H13O7 + RO$_2$ = 2RO + O$_2$ | $0.6\times8.8\times10^{-13}$ |
| 54 | C9H13O7 + OH = RPO$_2$ + H$_2$O | $1.00\times10^{-10}$ |
| 55 | C9H13O7 + HO$_2$ = ROOH + O$_2$ | $1.20\times10^{-11}$ |
| 56 | C9H13O7 = wall loss | 0.0023 |
| 57 | C9H13O7+ NO = RO + NO$_2$ | $0.843\times8.50\times10^{-12}$ |
| 58 | C9H13O7 + NO + M = RONO$_2$+ M | $0.157\times8.50\times10^{-12}$ |
| 59 | ROOH + OH = RO$_2$ + H$_2$O | $5.30\times10^{-12}\times e^{190/T}\times0.6$ |
| 60 | ROOH + OH = RPHO + OH + H$_2$O | $5.30\times10^{-12}\times e^{190/T}\times0.4$ |
| 61 | RO + O$_2$ = RPO + HO$_2$ | $6.00\times10^{-15}$ |
| 62 | H$_2$O$_2$ + $hv$ ($\lambda$ = 254 nm) = 2OH | $6.70\times10^{-20}\times$flux$_{254}$ |
| 63 | NO$_2$ + $hv$ ($\lambda$ = 254 nm) = O + NO | $1.00\times10^{-20}\times$flux$_{254}$ |
| 64 | HONO + $hv$ ($\lambda$ = 254 nm) = OH + NO | $1.40\times10^{-19}\times$flux$_{254}$ |
| 65 | HNO$_3$ + $hv$ ($\lambda$ = 254 nm) = OH + NO$_2$ | $1.95\times10^{-20}\times$flux$_{254}$ |
| 66 | HNO$_4$ + $hv$ ($\lambda$ = 254 nm) = HO$_2$ + NO$_2$ | $3.60\times10^{-19}\times$flux$_{254}$ |
| 67 | N$_2$O$_5$ + $hv$ ($\lambda$ = 254 nm) = NO$_2$ + NO$_3$ | $3.20\times10^{-19}\times$flux$_{254}$ |

In this work, autoxidation and accretion of 1,3,5-TMB-derived BPR, as well as the subsequent reactions of the autoxidation product of BPR, i.e., C$_9$H$_{13}$O$_7$·, are newly implemented or modified in this model (Reaction No. 42 – 58 in Table R1). The pathways of peroxy radicals and their kinetics are discussed below. NO$_x$-related reactions are also included in the model. When we simulate experiments without NO$_x$, these reactions do not contribute to the simulation results.

RO$_2$ can react with a number of radicals, generating termination products or other radicals.

$$RO_2 + R'O_2 \rightarrow RO + R'O + O_2 \qquad (R1)$$
$$RO_2 + R'O_2 \rightarrow R=O + R'OH + O_2 \qquad (R2)$$
$$RO_2 + R'O_2 \rightarrow ROH + R'=O + O_2 \qquad (R3)$$
$$RO_2 + R'O_2 \rightarrow ROOR' + O_2 \qquad (R4)$$
$$RO_2 + HO_2 \rightarrow ROOH + O_2 \qquad (R5)$$

$R1$, $R2$, and $R3$ are reactions of RO$_2$ + RO$_2$, forming alkoxy radicals, carbonyl termination products, and hydroxyl termination products, respectively. $R4$ is an accretion reaction, forming dimers via combination of two monomeric RO$_2$. $R5$ is the reaction between RO$_2$ and HO$_2$, forming hydroperoxyl radicals. The reaction rate constants for RO$_2$ in $R1 – R5$ are obtained by MCM or previous investigations (e.g., Jenkin et al., 2003; Berndt et al., 2018; Peng and Jimenez, 2020). We treat $R1 – R3$ as a total reaction with a reaction rate constant of $8.8\times10^{-13}$ molecule$^{-1}$ cm$^3$ s$^{-1}$, and branching ratios of $R1 - R3$ of 0.6, 0.2, and 0.2, respectively, as suggested by MCM (Jenkin et al., 2003). The reaction rate constants of BPR and C$_9$H$_{13}$O$_7$· for $R4$ are $1.7\times10^{-10}$ and $2.6\times10^{-10}$ molecule$^{-1}$ cm$^3$ s$^{-1}$, respectively (Berndt et al., 2018). The reaction rate constants for $R5$ is $1.5\times10^{-}$

[11] molecule$^{-1}$ cm$^3$ s$^{-1}$ (Jenkin et al., 2003).

$$RO_2 + OH \rightarrow Products \qquad\qquad (R6)$$

$R6$ is the reaction between OH and $RO_2$. The reaction rate constant for $R6$ is $1 \times 10^{-10}$ molecule$^{-1}$ cm$^3$ s$^{-1}$ according to previous studies (Bossolasco et al., 2014; Yan et al., 2016; Assaf et al., 2016, 2017; Peng and Jimenez, 2020). Current knowledge on the reaction products for the reaction of $CH_3O_2\cdot + OH$, the most studied $RO_2 + OH$ reaction, is summarized in Table R2 (also as Table S3). The products of this reaction are suggested to include a Criegee intermediate ($CH_2O_2\cdot$), a stabilized methylhydrotrioxide ($CH_3OOOH$), an alkoxy radical ($CH_3O\cdot$), and methanol ($CH_3OH$) (Yan et al., 2016; Fittschen, 2019; Caravan et al., 2018; Müller et al., 2016). Müller et al. (2016) and Caravan et al. (2018) suggest that the formation of $CH_2O_2\cdot$ is actually infeasible, and Yan et al. (2016) estimated an upper limit branching ratio of 5% for this pathway. The branching ratios of stabilized products $CH_3OH$ and $CH_3OOOH$ are 6 - 7% (Caravan et al., 2018; Müller et al., 2016) and 7% (Müller et al., 2016), respectively. The most significant product of this reaction is the alkoxy radical ($CH_3O\cdot$), with a branching ratio of more than 86% (Müller et al., 2016). In the absence of $NO_x$, $CH_3OH$ and $CH_3O\cdot$ can also be formed via the traditional unimolecular reaction between $CH_3O_2\cdot$ and $RO_2$, i.e., $R1$ and $R3$. The possible role of this reaction of large $RO_2$, i.e., BPR and other C9-$RO_2$, with OH has not yet been investigated. However, according to the branching ratios for the reaction of $CH_3O_2\cdot + OH$, this reaction is likely to form RO instead of stabilized C9 products. Hence, we assume that the branching ratios of hydrotrioxide (ROOOH), RO, and ROH are 0.07, 0.86, and 0.07, respectively, for BPR + OH and C9-$RO_2$ + OH.

Table R2. The branching ratios of different pathways for $CH_3O_2\cdot + OH$.

| Reactions | Branching ratio | References |
| --- | --- | --- |
| $CH_3O_2\cdot + OH \rightarrow CH_2O_2\cdot + H_2O$ | < 5% | (Yan et al., 2016) |
| | 0 | (Caravan et al., 2018; Müller et al., 2016) |
| $CH_3O_2\cdot + OH \rightarrow CH_3O\cdot + HO_2$ | 86% | (Müller et al., 2016) |
| $CH_3O_2\cdot + OH \rightarrow CH_3OH + HO_2$ | 6 ± 2% | (Caravan et al., 2018) |
| | 7% | (Müller et al., 2016) |
| $CH_3O_2\cdot + OH \rightarrow CH_3OOOH$ | 7% | (Müller et al., 2016) |

$$RO_2 \xrightarrow{\text{isomerization}} Products \qquad\qquad (R7)$$

Unimolecular reactions can also contribute to consumption of $RO_2$ in the PAM OFR. $RO_2$ isomerization rate coefficients are highly dependent on their structures, spanning from $10^{-3}$ - $10^6$ s$^{-1}$ (Bianchi et al., 2019; Crounse et al., 2013; Knap and Jørgensen, 2017; Praske et al., 2018). However, only some substituted acyl $RO_2$ can undergo rapid isomerization at a reaction rate of $10^6$ s$^{-1}$ (Knap and Jørgensen, 2017). 1,3,5-TMB-derived BPR and its autoxidation product, $C_9H_{13}O_7\cdot$, do not belong to the group of substituted acyl $RO_2$ (Molteni et al., 2018; Tsiligiannis et al., 2019). The most important unimolecular reactions for 1,3,5-TMB-derived BPR is likely autoxidation while the precise autoxidation reaction rates of 1,3,5-TMB-derived BPR and other $RO_2$ in this system are currently unclear (Bianchi et al., 2019; Molteni et al., 2018). Previous theoretical investigations suggest that more than 90% BPR generated by the oxidation of 1,3,5-TMB possess a structure favoring autoxidation and thus their overall autoxidation reaction rate is relatively fast (Wang et al., 2017). Laboratory experiments also indicate a higher HOM molar yield for 1,3,5-TMB than

ethylbenzene and xylenes (Molteni et al., 2018). We arbitrarily set the autoxidation reaction rate of 1,3,5-TMB-derived BPR the same as that of ethylbenzene-derived BPR, i.e., 7.0 s$^{-1}$, as a lower limit to estimate the fate of 1,3,5-TMB-derived $RO_2$ (Wang et al., 2017). Indeed, this value is not necessarily appropriate for all the $RO_2$ in this system and this estimation is a simplified result mainly based on the most important $RO_2$ in the oxidation of 1,3,5-TMB, i.e., BPR. Meanwhile, this value will not influence the total concentration of $RO_2$ but the concentration of BPR, as the total $RO_2$ concentration is simplified to be the sum of concentrations of BPR and $C_9H_{13}O_7\cdot$.

The reactions between NO and $RO_2$ can generate alkoxy radicals similar to $R1$ and organonitrates, which are regarded as $R8$ and $R9$.

$$RO_2 + NO \rightarrow RO + NO_2 \qquad\qquad\qquad (R8)$$
$$RO_2 + NO \rightarrow RONO_2 \qquad\qquad\qquad (R9)$$

The reaction rate for the sum of these two reactions is taken as $8.5\times10^{-12}$ molecule$^{-1}$ cm$^3$ s$^{-1}$. The branching ratios of these two reactions are 0.843 and 0.157, respectively, according to MCM (Jenkin et al., 2003).

Alkoxy radicals, RO, will be generated in $R1$, $R6$, and $R8$. The widely used near-explicit mechanism, MCM, assumes that RO formed via the alkoxy channel of BPR ($R1$) will decompose into small molecules. Recently, Xu et al. (2020) probed the chemical fates of BPR-derived RO, hereafter referred to as bicyclic alkoxy radical (BCP-oxy), in the oxidation of benzene by laboratory experiments and model calculations, which can be taken as a reference to induce the mechanism of 135-TMB-derived BCP-oxy. BCP-oxy can undergo two reactions, i.e., ring-breakage and ring-closure, and a new calculation result suggests that the branching ratio of ring-breakage reaction is larger than 98% (Wang et al., 2013). 56% of ring-breakage reactions will break benzene-derived BCP-oxy into butenedial and glyoxal, and the rest 44% will generate a C6 alkyl radical by the 1,5-aldehydic H-shift. The latter C6 alkyl radical will further undergo other reactions, including a 93% branching ratio for decomposition reactions that results in a reduction of carbon atom number (Xu et al., 2020). Therefore, most of benzene-derived BCP-oxy will likely decompose into compounds with fewer carbon atoms. We assume that 1,3,5-TMB-derived BCP-oxy will undertake these decomposition reactions with a similar branching ratio, which means that these radicals cannot form a large number of stabilized products that can influence the distributions of stabilized C9 products in nitrate CIMS.

The physical loss of $RO_2$ in the PAM OFR consists of the condensation loss to the aerosol particles and the diffusion loss to the OFR walls, which can be regarded as $R10$.

$$RO_2 \rightarrow physical\ loss \qquad\qquad\qquad (R10)$$

In our experiments, measurement results by a long-SMPS show that the aerosol particles presented in the PAM OFR were few and thus the condensation loss of HOMs to the aerosol particles was minor and not further considered. The first-order loss rate of HOMs to the OFR walls, $k_{wall}$, is limited by eddy diffusion and can be calculated with the following function (Cheng et al., 2021; Palm et al., 2016; McMurry and Grosjean, 1985):

$$k_{wall} = \frac{A}{V} \cdot \frac{2}{\pi} \cdot \sqrt{k_e D_g} \qquad\qquad\qquad (Eq1)$$

where the OFR surface-area-volume ratio ($A/V$) is 25 m$^{-1}$ and the coefficient of eddy diffusion ($k_e$) is 0.0042 s$^{-1}$, as estimated by the method utilized in a previous study (Brune, 2019) and given in $Eq2$.

$$k_e = 0.004 + 10^{-2.25}V^{0.74} \qquad (Eq2)$$

where $V$ is the enclosure volume (m$^3$). The molecular diffusion coefficient, $D_g$, is estimated with the method as described by Fuller et al. (1966) and is around $5\times10^{-6}$ m$^2$ s$^{-1}$ with 1,3,5-TMB derived BPR as an example. Hence, $k_{wall}$ is around 0.0023 s$^{-1}$ in the PAM OFR.

The input parameters of temperature, mean residence time, water vapor concentration, O$_3$ concentration, and the initial 1,3,5-TMB concentration are 25 ℃, 53 s, 0.8%, 500 ppbv, and 50 ppbv, respectively, as measured directly in the experiments. The actinic flux at 254 nm, $I_{254}$, is constrained by comparing OH exposures by model output and OH exposures estimated by the consumption of 1,3,5-TMB as measured by a Vocus PTR. Consumption of O$_3$ estimated by the model agrees well with the measured results, with discrepancies being always within 10% at different OH exposures.

The above calculation allows us to evaluate radical concentrations and fates of RO$_2$ in our OFR, and to compare results between our experiments and those under ambient conditions.

Concentration profiles of OH, RO$_2$, and HO$_2$ as a function of OH exposures in our experiments without NO$_x$ are illustrated in Figure R1a (also as Figure S1a). According to the modified PAM_chem_v8, when OH increased from $1.09\times10^8$ to $1.57\times10^9$ molecule cm$^{-3}$, HO$_2$ concentrations increased from $7.72\times10^8$ to $3.18\times10^9$ molecule cm$^{-3}$, whereas RO$_2$ concentrations increased from $4.83\times10^9$ to $8.48\times10^9$ molecule cm$^{-3}$. The radical concentrations in our experiments with NO$_x$ (Figure R1b, also as Figure S1b) varied in a similar range, with RO$_2$ ranging from $3.89\times10^9$ to $9.34\times10^9$ molecule cm$^{-3}$, HO$_2$ ranging from $3.66\times10^9$ to $6.82\times10^9$ molecule cm$^{-3}$, and OH ranging from $4.83\times10^8$ to $9.05\times10^8$ molecule cm$^{-3}$, respectively. The ratios between HO$_2$/OH and RO$_2$/OH in our experiments are displayed in Figure R1c (also as Figure S1c). The HO$_2$/OH ratio ranged between 1.9 and 7.1 in our PAM OFR experiments without NO$_x$, and the RO$_2$/OH ratio ranged between 4.9 and 47.9. In experiments with NO$_x$, the HO$_2$/OH ratio ranged between 3.7 and 17.9, whilst the RO$_2$/OH ratio ranged between 4.0 and 13.2. A recent comprehensive ambient campaign conducted in the wintertime central Beijing reported mean daytime peak concentrations of $8.8\times10^7$, $3.9\times10^7$, and $2.7\times10^6$ molecule cm$^{-3}$ for total RO$_2$, HO$_2$, and OH, respectively (Slater et al., 2020), which corresponds to ambient RO$_2$/OH and HO$_2$/OH ratios of 32.6 and 14.4 (Figure R1c), respectively. Therefore, radical ratios in our flow tube were generally in the same order of magnitude with the ambient conditions.

(a)

[Figure]

(b)

[Figure]

(c)

[Figure]

**Figure R1.** (a) Concentration profiles of OH, HO$_2$, BPR, and total RO$_2$ in the PAM OFR experiments without NO$_x$, as a function of OH exposures. The average total concentrations of RO$_2$ were scaled with a factor of 0.1 for a better visualization. (b) Concentration profiles of OH, HO$_2$, BPR, and total RO$_2$ in the PAM OFR experiments with NO$_x$, as a function of OH exposures. The average total concentrations of RO$_2$ were scaled with a factor of 0.1 for a better visualization. (c) HO$_2$/OH, RO$_2$×0.1/OH, and their ambient values. The ambient values were calculated according to Slater et al. (2020).

[revised manuscript text omitted]

Overall, we argue that we focused on the detailed formulae of stabilized products and confirmed the extensive existence of secondary OH oxidation through the OFR experiments. Our experimental results can be taken as reference to characterize chemical behaviors of HOMs in the atmosphere. Yields of Organonitrates and HOM dimers have been altered in our experiments, whilst their formulae clearly confirm their generation pathways and the significance of secondary OH oxidation. We acknowledge that our previous discussion on yields of HOMs in the original manuscript could be misleading, and thus we have removed those contents in the revised manuscript.

In addition, our settings are much closer to the true ambient compared to the three studies listed by Reviewer #1.

The Wang et al. (2017) study did not provide detailed concentrations of aromatic precursors and generated extremely low concentrations of OH ($(2.4 - 53) \times 10^4$ molecule cm$^{-3}$) by ozonolysis of tetramethylethylene in their study. Almost no $HO_2$ were formed in the flow tube, which made the termination of $RO_2$ very slow when comparing to the unimolecular reactions.

The OH concentrations in our OFR experiments, i.e., $1.09 \times 10^8 - 1.57 \times 10^9$ molecule cm$^{-3}$, are close to those in the Garmash et al. (2020) chamber experiments, which were in the range of $1.2 \times 10^7 - 4.5 \times 10^8$ molecule cm$^{-3}$. The much higher residence time in their experiments (48 min) than ours (53 s) makes the OH dose in our experiments much lower than theirs. Secondary OH reactions of stabilized first-generation products in their system is likely more favorable than ours. The Garmash et al. (2020) study did not provide a detailed estimation on concentrations of $RO_2$ and $HO_2$ in their experiments. However, according to their results, the termination products were dominated by -OOH, indicating the existence of a high $HO_2$ concentration. Meanwhile, the ratio between toluene-derived monomers and dimers detected by their nitrate-CIMS was 0.66, indicating a high $RO_2$ concentration that favors accretion reactions in their experiments. The high concentration of precursors (~ 400 ppm benzene/ ~ 25 ppm toluene/ ~ 0.4 ppm naphthalene) in their 'University of Helsinki flow reactor' also likely resulted in an extremely high $RO_2$ condition.

Compared to the Molteni et al. (2018) study, our experiments are generally much closer to the true ambient. Their OH concentrations in the 1,3,5-TMB oxidation experiments are around $7 \times 10^5$ molecule cm$^{-3}$. On the other hand, their extremely high $HO_2$ concentrations, i.e., $8 \times 10^9$ molecule cm$^{-3}$, resulted in a $HO_2$/OH of 20000 and led to a much earlier $RO_2$ termination.

All the three studies utilized oxidation products observed in the OFR or chambers as evidence to derive reaction mechanisms (Garmash et al., 2020; Wang et al., 2017; Molteni et al., 2018), though Wang et al. (2017) used extra supports from quantum chemical calculations. We believe that our results are relevant and provide further insights into the oxidation mechanisms of aromatics. On the other hand, our experiments fill in the gap of oxidation of aromatics under an intraday OH exposure.

We have revised our manuscript to include in the above argument, which reads,

**2. Methods. (Line 252-295)**

To validate our settings, a PAM chemistry model (PAM_chem_v8), utilized widely in previous studies, were chosen with the latest updates to calculate radical profiles in our OFR (Li et al., 2015; Cheng et al., 2021; Wang et al., 2020; Mehra et al., 2020; Lambe et al., 2015, 2018; Peng and Jimenez, 2020; Lambe et al., 2017). This model is based on a photochemical box model that includes chemistry of photolysis of oxygen, water vapor, and other trace gases by the primary wavelengths of mercury lamps, and simplified VOC and $RO_2$ chemistry (Table S2), but further reactions of the first-generation stabilized products and the second-generation organic radicals are not considered. The detailed reactions involved with $RO_2$ include:

$$RO_2 + R'O_2 \rightarrow RO + R'O + O_2 \qquad (R1)$$
$$RO_2 + R'O_2 \rightarrow R = O + R'OH + O_2 \qquad (R2)$$
$$RO_2 + R'O_2 \rightarrow ROH + R' = O + O_2 \qquad (R3)$$
$$RO_2 + R'O_2 \rightarrow ROOR' + O_2 \qquad (R4)$$
$$RO_2 + HO_2 \rightarrow ROOH + O_2 \qquad (R5)$$
$$RO_2 + OH \rightarrow Products \qquad (R6)$$
$$RO_2 \xrightarrow{isomerization} Products \qquad (R7)$$
$$RO_2 + NO \rightarrow RO + NO_2 \qquad (R8)$$
$$RO_2 + NO \rightarrow RONO_2 \qquad (R9)$$
$$RO_2 \rightarrow physical\ loss \qquad (R10)$$

$R1$, $R2$, and $R3$ are reactions of $RO_2 + RO_2$, forming alkoxy radicals, carbonyl termination products, and hydroxyl termination products, respectively. $R4$ is the accretion reaction, forming dimers via combination of two monomeric $RO_2$. $R5$ is the reaction between $RO_2$ and $HO_2$, forming hydroperoxyl radicals. $R6$ is the reaction between OH and $RO_2$, whose reaction products are proposed with a reference from the previous studies concluded in Table S3. $R7$ is the unimolecular reactions of $RO_2$ in the PAM OFR, among which the autoxidation reaction rate is the most significant. $R8$ and $R9$ are the reactions between NO and $RO_2$, generating alkoxy radicals and organonitrates, respectively. $R10$ is the physical loss of $RO_2$.

Reactions in the modified PAM_chem_v8 and their detailed kinetics are provided in Table S2. Kinetic data 
[revised manuscript text omitted]

**Supplement:**
**Text S1. Introduction of the newly implemented and modified reactions in PAM model.**

To better illustrate and evaluate the chemistry in the PAM OFR in our experiments, a PAM chemistry model (PAM_chem_v8), utilized widely in previous studies, is chosen with the latest updates to calculate radical profiles in our OFR (Li et al., 2015; Cheng et al., 2021; Wang et al.,

2020; Mehra et al., 2020; Lambe et al., 2015, 2018; Peng and Jimenez, 2020; Lambe et al., 2017). In this work, autoxidation and accretion of 1,3,5-TMB-derived BPR, as well as the subsequent reactions of the autoxidation product of BPR, i.e., $C_9H_{13}O_7\cdot$, are newly implemented or modified in this model (Reaction No. 42 – 58 in Table S2). The pathways of the peroxy radicals and their kinetics are discussed below. $NO_x$-related reactions are also included in the model. When we simulate experiments without $NO_x$, these reactions do not contribute to the simulation results.

RO₂ can react with a number of radicals, generating termination products or other radicals.

$$RO_2 + R'O_2 \rightarrow RO + R'O + O_2 \qquad\qquad (R1)$$
$$RO_2 + R'O_2 \rightarrow R = O + R'OH + O_2 \qquad\qquad (R2)$$
$$RO_2 + R'O_2 \rightarrow ROH + R' = O + O_2 \qquad\qquad (R3)$$
$$RO_2 + R'O_2 \rightarrow ROOR' + O_2 \qquad\qquad (R4)$$
$$RO_2 + HO_2 \rightarrow ROOH + O_2 \qquad\qquad (R5)$$

$R1$, $R2$, and $R3$ are reactions of $RO_2 + RO_2$, forming alkoxy radicals, carbonyl termination products, and hydroxyl termination products, respectively. $R4$ is an accretion reaction, forming dimers via combination of two monomeric $RO_2$. $R5$ is the reaction between $RO_2$ and $HO_2$, forming hydroperoxyl radicals. The reaction rate constants for $RO_2$ in $R1 - R5$ were obtained by MCM or previous investigations (e.g., Jenkin et al., 2003; Berndt et al., 2018; Peng and Jimenez, 2020). We treat $R1 - R3$ as a total reaction with a reaction rate constant of $8.8\times10^{-13}$ molecule$^{-1}$ cm$^3$ s$^{-1}$, and branching ratios of $R1 - R3$ of 0.6, 0.2, and 0.2, respectively, as suggested by MCM (Jenkin et al., 2003). The reaction rate constants of BPR and $C_9H_{13}O_7\cdot$ for $R4$ are $1.7\times10^{-10}$ and $2.6\times10^{-10}$ molecule$^{-1}$ cm$^3$ s$^{-1}$, respectively (Berndt et al., 2018). The reaction rate constants for $R5$ is $1.5\times10^{-11}$ molecule$^{-1}$ cm$^3$ s$^{-1}$ (Jenkin et al., 2003).

$$RO_2 + OH \rightarrow Products \qquad\qquad (R6)$$

$R6$ is the reaction between OH and $RO_2$. The reaction rate constant for $R6$ is $1\times10^{-10}$ molecule$^{-1}$ cm$^3$ s$^{-1}$ according to previous studies (Bossolasco et al., 2014; Yan et al., 2016; Assaf et al., 2016, 2017; Peng and Jimenez, 2020). Current knowledge on the reaction products for the reaction of $CH_3O_2\cdot + OH$, the most studied $RO_2 + OH$ reaction, is summarized in Table S3. The products of this reaction are suggested to include a Criegee intermediate ($CH_2O_2\cdot$), a stabilized methylhydrotrioxide ($CH_3OOOH$), an alkoxy radical ($CH_3O\cdot$), and methanol ($CH_3OH$) (Yan et al., 2016; Fittschen, 2019; Caravan et al., 2018; Müller et al., 2016). Müller et al. (2016) and Caravan et al. (2018) suggest that the formation of $CH_2O_2\cdot$ is actually infeasible, and Yan et al. (2016) estimated an upper limit branching ratio of 5% for this pathway. The branching ratios of stabilized products $CH_3OH$ and $CH_3OOOH$ are 6 - 7% (Caravan et al., 2018; Müller et al., 2016) and 7% (Müller et al., 2016), respectively. The most significant product of this reaction is the alkoxy radical ($CH_3O\cdot$), with a branching ratio of more than 86% (Müller et al., 2016). In the absence of $NO_x$, $CH_3OH$ and $CH_3O\cdot$ can also be formed via the traditional unimolecular reaction between $CH_3O_2\cdot$ and $RO_2$, i.e., $R1$ and $R3$. The possible role of this reaction of large $RO_2$, i.e., BPR and other $C9$-$RO_2$, with OH has not yet been investigated. However, according to the branching ratios for the reaction of $CH_3O_2\cdot + OH$, this reaction is likely to form RO instead of stabilized C9 products. Hence, we assume that the branching ratios of hydrotrioxide (ROOOH), RO, and ROH are 0.07, 0.86, and 0.07, respectively, for BPR + OH and $C9$-$RO_2$ + OH.

$$RO_2 \xrightarrow{\text{isomerization}} Products \qquad\qquad (R7)$$

Unimolecular reactions can also contribute to consumption of $RO_2$ in the PAM OFR. $RO_2$

isomerization rate coefficients are highly dependent on their structures, spanning from $10^{-3}$ - $10^6$ s$^{-1}$ (Bianchi et al., 2019; Crounse et al., 2013; Knap and Jørgensen, 2017; Praske et al., 2018). However, only some substituted acyl RO$_2$ can undergo rapid isomerization at a reaction rate of $10^6$ s$^{-1}$ (Knap and Jørgensen, 2017). 1,3,5-TMB-derived BPR and its autoxidation product, C$_9$H$_{13}$O$_7$·, do not belong to the group of substituted acyl RO$_2$ (Molteni et al., 2018; Tsiligiannis et al., 2019). The most important unimolecular reactions for 1,3,5-TMB-derived BPR is likely autoxidation while the precise autoxidation reaction rates of 1,3,5-TMB-derived BPR and other RO$_2$ in this system are currently unclear (Bianchi et al., 2019; Molteni et al., 2018). Previous theoretical investigations suggest that more than 90% BPR generated by the oxidation of 1,3,5-TMB possess a structure favoring autoxidation and thus their overall autoxidation reaction rate is relatively fast (Wang et al., 2017). Laboratory experiments also indicate a higher HOM molar yield for 1,3,5-TMB than ethylbenzene and xylenes (Molteni et al., 2018). We arbitrarily set the autoxidation reaction rate of 1,3,5-TMB-derived BPR the same as that of ethylbenzene-derived BPR, i.e., 7.0 s$^{-1}$, as a lower limit to estimate the fate of 1,3,5-TMB-derived RO$_2$ (Wang et al., 2017). Indeed, this value is not necessarily appropriate for all the RO$_2$ in this system and this estimation is a simplified result mainly based on the most important RO$_2$ in the oxidation of 1,3,5-TMB, i.e., BPR. Meanwhile, this value will not influence the total concentration of RO$_2$ but the concentration of BPR, as the total RO$_2$ concentration is simplified to be the sum of concentrations of BPR and C$_9$H$_{13}$O$_7$·.

Especially, because RO$_2$ in our experiments mainly consists of RO$_2$ with multiple oxygenated functionalities and high carbon contents, i.e., BPR and its isomerization products, which can undergo accretion reactions rapidly as fast as $10^{-10}$ molecule$^{-1}$ cm$^3$ s$^{-1}$. However, only around 50% RO$_2$ in the atmosphere are typically derived from aromatics and long-chain-alkanes containing carbon atoms larger than 4 that can undertake accretion reactions at a considerable reaction rate coefficient (Berndt et al., 2018; Bianchi et al., 2019), as observed in Beijing (Slater et al., 2020; Tan et al., 2018). Therefore, the proportion of accretion reaction in the ambient was calculated with half of the reaction rate coefficient, i.e., $8.5\times10^{-11}$ molecule$^{-1}$ cm$^3$ s$^{-1}$. The reactions between NO and RO$_2$ can generate alkoxy radicals similar to $R1$ and organonitrates, which are regarded as $R9$ and $R10$.

$$RO_2 + NO \rightarrow RO + NO_2 \qquad\qquad (R8)$$
$$RO_2 + NO \rightarrow RONO_2 \qquad\qquad (R9)$$

The reaction rate for the sum of these two reactions is $8.5\times10^{-12}$ molecule$^{-1}$ cm$^3$ s$^{-1}$. The branching ratios of these two reactions are 0.843 and 0.157, respectively, according to MCM (Jenkin et al., 2003).

Alkoxy radicals, RO, will be generated in $R1$ , $R6$, and $R8$. The widely used near-explicit mechanism, MCM, assumes that RO formed via the alkoxy channel of BPR ($R1$) will decompose into small molecules. Recently, Xu et al. (2020) probed the chemical fates of BPR-derived RO, hereafter referred to as bicyclic alkoxy radical (BCP-oxy), in the oxidation of benzene by laboratory experiments and model calculations, which can be taken as a reference to induce the mechanism of 135-TMB-derived BCP-oxy. BCP-oxy can undergo two reactions, i.e., ring-breakage and ring-closure, and a new calculation result suggests that the branching ratio of ring-breakage reaction is larger than 98% (Wang et al., 2013). 56% of ring-breakage reactions will break benzene-derived BCP-oxy into butenedial and glyoxal, and the rest 44% will generate a C6 alkyl radical by the 1,5-aldehydic H-shift. The latter C6 alkyl radical will further undergo other reactions, including a 93% branching ratio for decomposition reactions that results in a reduction of carbon atom number (Xu

et al., 2020). Therefore, most of benzene-derived BCP-oxy will likely decompose into compounds with fewer carbon atoms. We assume that 1,3,5-TMB-derived BCP-oxy will undertake these decomposition reactions with a similar branching ratio, which means that these radicals cannot form a large number of stabilized products that can influence the distributions of stabilized C9 products in nitrate CIMS.

The physical loss of $RO_2$ in the PAM OFR consists of the condensation loss to the aerosol particles and the diffusion loss to the OFR walls, which can be regarded as $R10$.

$$RO_2 \rightarrow physical\ loss \qquad\qquad (R10)$$

In our experiments, measurement results by a long-SMPS show that the aerosol particles presented in the PAM OFR were few and thus the condensation loss of HOMs to the aerosol particles was minor and not further considered. The first-order loss rate of HOMs to the OFR walls, $k_{wall}$, is limited by eddy diffusion and can be calculated with the following function (Cheng et al., 2021; Palm et al., 2016; McMurry and Grosjean, 1985):

$$k_{wall} = \frac{A}{V} \cdot \frac{2}{\pi} \cdot \sqrt{k_e D_g} \qquad\qquad (Eq1)$$

where the OFR surface-area-volume ratio ($A/V$) is 25 m$^{-1}$ and the coefficient of eddy diffusion ($k_e$) is 0.0042 s$^{-1}$, as estimated by the method utilized in a previous study (Brune, 2019) and given in $Eq2$.

$$k_e = 0.004 + 10^{-2.25} V^{0.74} \qquad\qquad (Eq2)$$

$V$ is the enclosure volume (m$^3$). The molecular diffusion coefficient, $D_g$, is estimated with the method as described by Fuller et al. (1966) and is around 5×10$^{-6}$ m$^2$ s$^{-1}$ with 1,3,5-TMB derived BPR as an example. Hence, $k_{wall}$ is around 0.0023 s$^{-1}$ in the PAM OFR.

(a)

[Figure]

(b)

[Figure]

(c)

[Figure]

**Figure S1.** (a) Concentration profiles of OH, $HO_2$, BPR, and total $RO_2$ in the PAM OFR as a function of OH exposures. The average total concentrations of $RO_2$ were scaled with a factor of 0.1 for a better visualization. (b) $HO_2$/OH, $RO_2 \times 0.1$/OH, and their ambient values. The ambient values were calculated according to Slater et al. (2020).

**Table S2.** Reactions included in the modified PAM_chem_v8 under the settings with only 254 nm UV lights on. For experiments in the absence of $NO_x$, the input value of $N_2O$ is 0 and all the $NO_x$-related reactions actually proceed with a zero rate. $RO_2$ is the sum of BPR and $C_9H_{13}O_7 \cdot$ for simplification.

| No | Reactions | Reaction rate constants/photolysis rate (molecule$^{-1}$ cm$^3$ s$^{-1}$/ s$^{-1}$) |
|----|-----------|-------------------------------------------------------------------------------------|
| 1 | HO$_2$ + $hv$ ($\lambda$ = 254 nm) = OH + O($^1$D) | $2.63\times10^{-19}\times$flux$_{254}$ |
| 2 | O$_3$ + O($^1$D) = 2O$_2$ | $1.20\times10^{-10}$ |
| 3 | O$_3$ + O($^1$D) = O + O + O$_2$ | $1.20\times10^{-10}$ |
| 4 | O + OH = H + O$_2$ | $2.20\times10^{-11}\times e^{120/T}$ |
| 5 | O($^1$D) + H$_2$ = OH + H | $1.20\times10^{-10}$ |
| 6 | HO$_2$ + H = 2OH | $7.20\times10^{-11}$ |
| 7 | HO$_2$ + H = O + H$_2$O | $1.60\times10^{-12}$ |
| 8 | HO$_2$ + H = H$_2$ + O$_2$ | $6.90\times10^{-12}$ |
| 9 | O$_3$ + H = OH + O$_2$ | $1.40\times10^{-11}\times e^{-470/T}$ |
| 10 | N$_2$O + O($^1$D) = 2NO | $6.70\times10^{-11}\times e^{20/T}$ |
| 11 | N$_2$O + O($^1$D) = N$_2$ + O$_2$ | $4.70\times10^{-11}\times e^{20/T}$ |
| 12 | O + HO$_2$ = OH + O$_2$ | $3.02\times10^{-11}\times e^{200/T}$ |
| 13 | O + H$_2$O$_2$ = OH + HO$_2$ | $1.40\times10^{-12}\times e^{-2000/T}$ |
| 14 | O + O$_3$ = 2O$_2$ | $8.00\times10^{-12}\times e^{-2060/T}$ |
| 15 | O + NO$_3$ = NO$_2$ + O$_2$ | $1.00\times10^{-11}$ |
| 16 | O + NO$_2$ = NO + O$_2$ | $5.12\times10^{-12}\times e^{210/T}$ |
| 17 | OH + O$_3$ = HO$_2$ + O$_2$ | $1.70\times10^{-12}\times e^{-940/T}$ |
| 18 | OH + HO$_2$ = H$_2$O + O$_2$ | $4.80\times10^{-11}\times e^{250/T}$ |
| 19 | OH + HONO = H$_2$O + NO$_2$ | $1.80\times10^{-11}\times e^{-390/T}$ |
| 20 | OH + H$_2$O$_2$ = H$_2$O + HO$_2$ | $2.90\times10^{-12}\times e^{-160/T}$ |
| 21 | OH + H$_2$ = H$_2$O + H | $2.80\times10^{-12}\times e^{-1800/T}$ |
| 22 | OH + OH = H$_2$O + O | $1.80\times10^{-12}$ |
| 23 | HO$_2$ + O$_3$ = OH + O$_2$ | $1.00\times10^{-14}\times e^{-490/T}$ |
| 24 | HO$_2$ + NO = OH + NO$_2$ | $3.50\times10^{-12}\times e^{270/T}$ |
| 25 | NO + O$_3$ = NO$_2$ + O$_2$ | $2.00\times10^{-12}\times e^{-1400/T}$ |
| 26 | NO$_2$ + O$_3$ = NO$_3$ + O$_2$ | $1.20\times10^{-13}\times e^{-2450/T}$ |
| 27 | NO + NO$_3$ = 2NO + O$_2$ | $1.50\times10^{-11}\times e^{170/T}$ |
| 28 | NO$_3$ + NO$_3$ = 2NO$_2$ + O$_2$ | $8.50\times10^{-13}\times e^{-2450/T}$ |
| 29 | N$_2$O$_5$ + H$_2$O = 2HNO$_3$ | $2.00\times10^{-21}$ |
| 30 | O + O$_2$ + M = O$_3$ + M | $6.00\times10^{-34}\times$M$\times(300/T)^{2.4}$ |
| 31 | H + O$_2$ + M = HO$_2$ + M | $k_o = 4.40\times10^{-32}\times$M$\times(300/T)^{1.3}$
 $k_h = 7.50\times10^{-11}\times(300/T)^{0.2}$

 $k = k_o/(1+(k_o/k_h))\times0.6^{(1+(\log10(k_o/k_h))^{-2})}$ |
| 32 | OH + OH + M = H$_2$O$_2$ + M | $k_o = 6.90\times10^{-31}\times$M$\times(300/T)$
 $k_h = 2.60\times10^{-11}$

 $k = k_o/(1+(k_o/k_h))\times0.6^{(1+(\log10(k_o/k_h))^{-2})}$ |
| 33 | OH + NO + M = HONO + M | $k_o = 7.00\times10^{-31}\times$M$\times(300/T)^{2.6}$
 $k_h = 3.60\times10^{-11}\times(300/T)^{0.1}$

 $k = k_o/(1+(k_o/k_h))\times0.6^{(1+(\log10(k_o/k_h))^{-2})}$ |
| 34 | OH + NO$_2$ + M = HNO$_3$ + M | $k_o = 1.80\times10^{-30}\times$M$\times(300/T)^{2.6}$
 $k_h = 2.80\times10^{-11}$

 $k = k_o/(1+(k_o/k_h))\times0.6^{(1+(\log10(k_o/k_h))^{-2})}$ |
| 35 | OH + HNO$_3$ = H$_2$O + NO$_3$ | $k_{00} = 2.40\times10^{-14}\times e^{460/T}$
 $k_{01} = 6.50\times10^{-34}\times e^{2199/T}$
 $k_{02} = 2.80\times10^{-11}\times e^{-2450/T}$
 $k = k_{00} + (k_{01}\times M)/(1+(k_{01}\times M)/k_{02})$ |
| 36 | HO$_2$ + NO$_2$ + M = HO$_2$NO$_2$ + M | $k_o = 1.80\times10^{-31}\times$M$\times(300/T)^{3.2}$
 $k_h = 4.70\times10^{-12}\times(300/T)^{1.4}$

 $k = k_o/(1+(k_o/k_h))\times0.6^{(1+(\log10(k_o/k_h))^{-2})}$
 $k_{reverse} = k /(2.10\times10^{-27}\times e^{10900/T})$ |
| 37 | NO$_2$ + NO$_3$ + M = N$_2$O$_5$ + M | $k_o = 2.00\times10^{-30}\times$M$\times(300/T)^{4.4}$
 $k_h = 1.40\times10^{-12}\times(300/T)^{0.7}$

 $k = k_o/(1+(k_o/k_h))\times0.6^{(1+(\log10(k_o/k_h))^{-2})}$ |

$$k_{reverse} = k /(2.70 \times 10^{-27} \times e^{11000/T})$$

| | | |
|---|---|---|
| 38 | OH + HNO$_4$ = products | $1.30 \times 10^{-12} \times e^{250/T}$ |
| 39 | Sci + H$_2$O = products | $4.00 \times 10^{-15}$ |
| 40 | 1,3,5-TMB + OH = BPR | $0.8 \times 5.67 \times 10^{-11}$ |
| 41 | 1,3,5-TMB + OH = Products | $0.2 \times 5.67 \times 10^{-11}$ |
| 42 | BPR = C9H13O7 | 7 |
| 43 | BPR + RO$_2$ = ROOR' | $1.70 \times 10^{-10}$ |
| 44 | BPR + RO$_2$ = R=O/ROH + O$_2$ | $0.4 \times 8.8 \times 10^{-13}$ |
| 45 | BPR + RO$_2$ = 2RO + O$_2$ | $0.6 \times 8.8 \times 10^{-13}$ |
| 46 | BPR + OH = RPO$_2$ + H$_2$O | $1.00 \times 10^{-10}$ |
| 47 | BPR + HO$_2$ = ROOH + O$_2$ | $1.20 \times 10^{-11}$ |
| 48 | BPR = wall loss | 0.0023 |
| 49 | BPR + NO = RO + NO$_2$ | $0.843 \times 8.50 \times 10^{-12}$ |
| 50 | BPR + NO + M = RONO$_2$+ M | $0.157 \times 8.50 \times 10^{-12}$ |
| 51 | C9H13O7 + RO$_2$ = ROOR' | $2.60 \times 10^{-10}$ |
| 52 | C9H13O7 + RO$_2$ = R=O/ROH + O$_2$ | $0.4 \times 8.8 \times 10^{-13}$ |
| 53 | C9H13O7 + RO$_2$ = 2RO + O$_2$ | $0.6 \times 8.8 \times 10^{-13}$ |
| 54 | C9H13O7 + OH = RPO$_2$ + H$_2$O | $1.00 \times 10^{-10}$ |
| 55 | C9H13O7 + HO$_2$ = ROOH + O$_2$ | $1.20 \times 10^{-11}$ |
| 56 | C9H13O7 = wall loss | 0.0023 |
| 57 | C9H13O7+ NO = RO + NO$_2$ | $0.843 \times 8.50 \times 10^{-12}$ |
| 58 | C9H13O7 + NO + M = RONO$_2$+ M | $0.157 \times 8.50 \times 10^{-12}$ |
| 59 | ROOH + OH = RO$_2$ + H$_2$O | $5.30 \times 10^{-12} \times e^{190/T} \times 0.6$ |
| 60 | ROOH + OH = RPHO + OH + H$_2$O | $5.30 \times 10^{-12} \times e^{190/T} \times 0.4$ |
| 61 | RO + O$_2$ = RPO + HO$_2$ | $6.00 \times 10^{-15}$ |
| 62 | H$_2$O$_2$ + $hv$ ($\lambda$ = 254 nm) = 2OH | $6.70 \times 10^{-20} \times \text{flux}_{254}$ |
| 63 | NO$_2$ + $hv$ ($\lambda$ = 254 nm) = O + NO | $1.00 \times 10^{-20} \times \text{flux}_{254}$ |
| 64 | HONO + $hv$ ($\lambda$ = 254 nm) = OH + NO | $1.40 \times 10^{-19} \times \text{flux}_{254}$ |
| 65 | HNO$_3$ + $hv$ ($\lambda$ = 254 nm) = OH + NO$_2$ | $1.95 \times 10^{-20} \times \text{flux}_{254}$ |
| 66 | HNO$_4$ + $hv$ ($\lambda$ = 254 nm) = HO$_2$ + NO$_2$ | $3.60 \times 10^{-19} \times \text{flux}_{254}$ |
| 67 | N$_2$O$_5$ + $hv$ ($\lambda$ = 254 nm) = NO$_2$ + NO$_3$ | $3.20 \times 10^{-19} \times \text{flux}_{254}$ |

**Table S3.** The branching ratios of different pathways for CH$_3$O$_2\cdot$ + OH.

| Reactions | Branching ratio | References |
|---|---|---|
| CH$_3$O$_2\cdot$ + OH → CH$_2$O$_2\cdot$ + H$_2$O | < 5% | (Yan et al., 2016) |
| | 0 | (Caravan et al., 2018; Müller et al., 2016) |
| CH$_3$O$_2\cdot$ + OH → CH$_3$O$\cdot$ + HO$_2$ | 86% | (Müller et al., 2016) |
| CH$_3$O$_2\cdot$ + OH → CH$_3$OH + HO$_2$ | 6 ± 2% | (Caravan et al., 2018) |
| | 7% | (Müller et al., 2016) |
| CH$_3$O$_2\cdot$ + OH → CH$_3$OOOH | 7% | (Müller et al., 2016) |

*Q 1.2 What is the influence of aromatic photochemistry in your PAM setup? Aromatics are known to strongly absorb light at relatively long wavelengths, and the oxygenated aromatics even more (see e.g., https://www.uv-vis-spectral-atlas-mainz.org/uvvis/), so I'm wondering how was the relevance of the used light sources tested in this work? This is not irrelevant for aromatic oxidation.*

**Response 1.2:**

Peng et al. (2016) evaluated the relative significance of photolysis of 1,3,5-TMB in the OFR.

The absorption cross-sections of 1,3,5-TMB at 254 nm is $3.68\times10^{-19}$ cm$^2$ (Keller-Rudek et al., 2013). For our experiments in the absence of NO$_x$, UV photon fluxes at 254 nm are estimated to range from $5.6\times10^{14}$ to $2.5\times10^{15}$ photons cm$^{-2}$ s$^{-1}$ based on the modified PAM_chem_v8. The photolysis rate of 1,3,5-TMB is estimated to range between $2.1\times10^{-4}$ and $9.2\times10^{-4}$ s$^{-1}$. Since the OH reaction rate constant of 1,3,5-TMB is $5.67\times10^{-11}$ molecule$^{-1}$ cm$^3$ s$^{-1}$ (Jenkin et al., 2003) and the OH concentration in the OFR was in the range of $1.09\times10^8$ – $1.57\times10^9$ molecule cm$^{-3}$, the OH reaction rate for 1,3,5-TMB is estimated to be $6.2\times10^{-3}$ – $8.9\times10^{-2}$ s$^{-1}$. Therefore, the ratio of photolysis-to-OH reaction was merely 0.010 – 0.033. Hence, photolysis of 1,3,5-TMB was insignificant in the OFR.

For stabilized products such as HOMs, the relative significance of photolysis can be estimated based on their photolysis rates. The cross sections of organic molecules are usually ~ $3.9\times10^{-18}$ - $3.9\times10^{-17}$ cm$^2$ (Peng et al., 2016). The photolysis quantum yields of multifunctional species are unlikely to be larger than those of species with only one carbonyl and one hydroxyl, as discussed in previous studies (Peng et al., 2016; Peng and Jimenez, 2020), which are around 0.1. The UV photon fluxes at 254 nm were in the range of $5.6\times10^{14}$ – $2.5\times10^{15}$ photons cm$^{-2}$ s$^{-1}$ as stated in the above. Therefore, the photolysis rates of HOMs are estimated to range between $2.18\times10^{-4}$ and $9.75\times10^{-3}$ s$^{-1}$. The reaction rate between OH and the stabilized first-generation products are estimated to be around $1.28\times10^{-10}$ molecule$^{-1}$ cm$^3$ s$^{-1}$, as suggested by MCM. Hence, the ratio of photolysis rates of HOMs to their secondary OH oxidation rates is estimated to be around 0.020 – 0.056. Meanwhile, photolysis of HOMs can lead to decomposition, decreasing detected signals of HOMs, but unlikely to generate new HOMs.

We have revised our manuscript (Line 177 – 189), which reads:

"Non-tropospheric VOC photolysis is a typical issue that should be taken into account when evaluating the settings of OFR laboratory experiments. Photolysis of the precursor and HOMs were evaluated, showing that photolysis was not a contributor to our observation. The photolysis rate of 1,3,5-TMB can be estimated based on the absorption cross-sections of 1,3,5-TMB at 254 nm (Keller-Rudek et al., 2013) and UV photon fluxes estimated by a chemistry model discussed in the following sections. The ratio of photolysis-to-OH reaction in our experiments was merely 0.010 – 0.033. Hence, photolysis of 1,3,5-TMB was insignificant in the OFR.

For stabilized products such as HOMs, the cross sections of organic molecules are usually ~ $3.9\times10^{-18}$ - $3.9\times10^{-17}$ cm$^2$ (Peng et al., 2016), while the reaction rate between OH and the stabilized first-generation products are estimated to be around $1.28\times10^{-10}$ molecule$^{-1}$ cm$^3$ s$^{-1}$, as suggested by MCM (Jenkin et al., 2003). Hence, the ratio of photolysis rates of HOMs to their secondary OH oxidation rates is estimated to be merely around 0.020 – 0.056."

*Q 1.3. You used a relatively long ¼ inch Teflon sampling tube for the CIMS. This is the smallest tube diameter I've ever come across with nitrate CIMS sampling. One would expect the HOM losses, especially the most oxygenated ones, to be very significant in this tube. Nevertheless, HOM with high O-content seems to be detected with this setup too!*

**Response 1.3:**
In our experiments, the sample flow is sampled from the center of the reactor. This PAM design

is identical to those utilized in the Jimenez group and other groups (Li et al., 2015; Lambe et al., 2017, 2015), including the position and type of lamps, volume, and the sampling method. Transmission efficiencies for typical gases, bis(2-ethylhexyl)sebacate particles, and $H_2SO_4$ particles in the PAM OFR, and sampling efficiencies into sampling tubes have been well characterized in a previous study, which shows a better transmission efficiency compared to other types of flow tubes (Lambe et al., 2011). This setting also validates usage of the PAM_chem_v8 model to estimate concentrations of radicals in the OFR. We acknowledge that this is not a perfect sampling setting for nitrate CIMS. However, the reduction in the sampling efficiencies of various HOMs is like to be close, if not identical, which keeps the distribution of HOMs.

We've revised our manuscript (Line 212 - 217), which reads:

"We followed the same sampling method of PAM OFR as those in previous studies, in order to obtain a similar flow tube residence time distributions (RTDs) and thus validate usage of the modified PAM_chem_v8 model to estimate concentrations of radicals in the OFR. We acknowledge that this is not a perfect sampling setting for nitrate CIMS. However, the reduction in the sampling efficiencies of various HOMs is likely to be close, if not identical, which keeps the distributions of HOMs."

***Q 1.4.*** *Jenkin 2003 reference does not have autoxidation.*

**Response 1.4:**

We use this reference to show that BPR is the main product of OH-initiated oxidation of aromatics. To avoid misunderstanding, we move this reference to a position closer to the BPR statement and add a new citation on autoxidation.

We have revised the manuscript (Line 70 - 74), which reads,

"Take alkylbenzenes as an example, previous studies suggest that the main products of OH-initiated oxidation of alkylbenzenes ($C_xH_{2x-6}$, x=7, 8, or 9), i.e., bicyclic peroxy radicals (BPR, $C_xH_{2x-6}O_5\cdot$, x=7, 8, or 9) (Jenkin et al., 2003), can undergo an autoxidation reaction and form a new peroxy radical, $C_xH_{2x-6}O_7\cdot$ (x=7, 8, or 9) (Wang et al., 2017)."

***Q 1.5.*** *The autoxidation reaction of BPR by H-abstraction has been found relatively slow by Wang et al 2017, not rapid.*

**Response 1.5:**

We assume that Reviewer #1 was referring to the following sentence, "The autoxidation reaction of BPR could be very fast because an allylic radical will be formed after the hydrogen shift (Wang et al., 2017)." Here, We intend to discuss the intramolecular H-shift or so-called H-migration (Bianchi et al., 2019) instead of the H-abstraction by the OH.

As stated in the Wang et al. (2017) study, "The routes via R4-BPRs are particularly important because of the relatively fast H-migration". Therefore, the exact autoxidation reaction rate should depend on the detailed structures of $RO_2$, some of which can be fast if their structures favor autoxidation.

We have revised our manuscript (Line 78 - 79), which reads, "The autoxidation of BPR could be fast if it has a favorable structure, as found in a previous study (Wang et al., 2017)."

***Q 1.6.*** *Several of the products detected seem to have worryingly many H-atoms in the structures. Especially the $C_9H_{17}O_m$ radicals.*

**Response 1.6:**

We are very sorry that we do not identify the exact molecule the referee is referring to.

In our experiments, HOM monomers typically contained 12 – 16 hydrogen atoms and HOM dimers typically contained 26 – 30 hydrogen atoms. The hydrogen atom numbers were the same as those reported in previous 1,3,5-TMB oxidation experiments (Molteni et al., 2018; Tsiligiannis et al., 2019).

As for $C_9H_{17}O_m$, we have not observed any compounds that support the existence of $C_9H_{17}O_m$ radicals (Line 296 – 304 in the original manuscript). we suggest that all the detected monomers did not possess hydrogen atoms more than 16.

***Q 1.7.*** *How well does the relatively low NO with the high $RO_2$ simulate atmospheric $NO_x$ chemistry?*

**Response 1.7:**

Please refer to our response 1.1

We acknowledge that the $NO:RO_2$ ratio in the PAM OFR is lower than typical values in the ambient atmosphere. There were two $NO_x$ settings in our experiments, which used 1.8 ppbv NO + 70 ppbv $NO_2$ and 4.8 ppbv NO + 120 ppbv $NO_2$, respectively. Because of the existence of $O_3$ that was utilized to generate $O(^1D)$ in the OFR and its rapid reaction rate with NO, the NO concentration in our system was unlikely to be increased by a large content. These two sets of $NO_x$ experiments are meant to validate the existence of multigeneration OH oxidation in the system, as proved by the existence of compounds with multiple nitrogen atoms in such a low $NO:RO_2$ ratio condition.

We have revised our manuscript (Line 551 - 556), which reads,

"The $NO:RO_2$ ratio in the PAM OFR is lower than typical values in the ambient atmosphere, which is due to the existence of $O_3$ that was utilized to generate $O(^1D)$ in the OFR and its rapid reaction rate with NO. However, due to rapid reaction rate constants between NO and $RO_2$, i.e., around $8.5 \times 10^{-12}$ molecule$^{-1}$ cm$^3$ s$^{-1}$, the reaction rate for the NO termination channel of $RO_2$ was as fast as around $0.3 – 1.0$ s$^{-1}$. Large amounts of organonitrates would still be formed. Our conclusion is also valid because of detection of compounds with multiple nitrogen atoms."

***Q 1.8.*** *"Such a slow autoxidation reaction rate cannot explain the extensive existence of HOM monomers with more than 7 oxygen atoms and HOM dimers with more than 10 oxygen atoms, which are the maximum numbers of oxygen atoms in stabilized monomer and dimer products, respectively, formed from $C_xH_{2x-6}O_7$.(Mentel et al., 2015; Molteni et al., 2018; Wang et al., 2020)."* → *There's a recent paper from my group that could provide an explanation what is observed here: https://www.nature.com/articles/s41467-023-40675-2*

**Response 1.8:**

Thanks for pointing out the latest reference.

We have revised our manuscript (Line 90 – 95), which reads,

"A very recent investigation offers new insights into the formation mechanism of these products, indicating the molecular rearrangement of BPR can initiate a series of autoxidation (Iyer

et al., 2023). However, the formation mechanism of HOMs with a large hydrogen atom number is still vague, e.g., monomer products with 16 hydrogen atoms in the OH-initiated oxidation of TMB and with 14 hydrogen atoms in the OH-initiated oxidation of xylene."

***Q 1.9.*** *I find it confusing to draw the "double-peroxide-ring" pathways in Schemes 1 and 2, if you even explicitly mention that they are unlikely. I advise to remove them, and the text " Another possibility is the formation of a second oxygen bridge after the hydrogen shift of BPR (Molteni et al., 2018)," altogether.*

**Response 1.9:**

Thanks for the suggestion.

We have revised our manuscript (Line 420 - 427), which reads,

"… has two isomers. A second-step of endo-cyclization is required in the formation of one of the isomer, which is extremely slow and not competitive as shown in several previous studies using both experimental and theoretical approaches (Wang et al., 2017; Xu et al., 2020). Even if such a second $O_2$ bridging to a double bond is assumed to be possible, the abundance of this isomer should be significantly smaller than the other one, because of the much faster reaction rate of H-shift reaction. Therefore, we do not take the $C_9H_{13}O_7\cdot$ isomer containing a double endo-cyclization into consideration in this work. The majority of HOM monomers is generated from subsequent reactions of $C_9H_{13}O_5\cdot$ and newly formed $C_9H_{13}O_7\cdot$, both of which contain one C=C bond in the carbon backbone and thus have a feasible site for OH addition. Meanwhile, the autoxidation reaction rate for newly formed $C_9H_{13}O_7\cdot$ should be …".

We have revised Scheme 1 as below:

**Scheme 1.** Oxidation pathways of the bicyclic peroxy radical $C_9H_{13}O_5\cdot$ (MCM name: TM135BPRO2) in the OH-initiated oxidation of 1,3,5-TMB. Green, blue, and black formulae denote alkyl peroxy radicals, alkoxy radicals and stabilized products, respectively. Black arrows denote the autoxidation pathway. MCM names for $HO_2$- and $RO_2$-termination products of TM135BPRO2 are present.

We have also revised Scheme 2 (Scheme 4 in the revised manuscript) as below:

**Scheme 4.** NO termination reactions of the bicyclic peroxy radical $C_9H_{13}O_5\cdot$ (MCM name: TM135BPRO2) and its autoxidation reaction products. Green, blue, and black formulae denote alkyl peroxy radicals, alkoxy radicals and stabilized products, respectively. Black arrows denote the autoxidation pathway. MCM names of NO-termination products of TM135BPRO2 are present.

*Q 1.10. "with an OH exposure equivalent to 2.4 – 19.4 days of atmospheric photochemical ageing. Certainly, such extremely high OH exposures favor secondary OH chemistry and help to facilitate our understanding on product distributions"*
*→ I would argue it doesn't, except for PAM conditions. As explained above, it does matter at what order and rate different oxidation steps happen in the atmosphere, and using such a high OH doses seem to necessarily skew up the chemistry. Figure 1 seems to be a good indication of this, as the "dimers" are generated faster than the monomers, and at the higher OH dose even the sum of "dimers" decrease.*

**Response 1.10:**

Validation of our experiments has been discussed in our Response 1.1.

The accretion reaction rate constant of 1,3,5-TMB-derived BPR has been well measured by

Berndt et al. (2018), which is around $1.7 \times 10^{-10}$ molecule$^{-1}$ cm$^3$ s$^{-1}$. They also calculated the reaction rates between RO$_2$ and RO$_2$, as well as between HO$_2$ and RO$_2$ under NO < 40 pptv, which shows that accretion reactions of BPR dominate if total concentrations of aromatics are within the range 4 – 40 ppbv. This estimation on the fates of RO$_2$ is similar to our experiments in the absence of NO$_x$. Hence, the faster formation of HOM dimers than monomers is expected, which made the maximum concentrations of HOM dimers appear earlier than HOM monomers. However, this study is not meant to compare HOM monomer and HOM dimer signals crossly here, but to pay attention to their formulae.

In order to avoid misunderstanding, we have revised our manuscript (Line 370 - 373), which reads,

"… The most abundant HOM products are also shown in stack in Figure 2, whose relationships with OH exposures are superimposed by …"

(Line 384 - 389), which reads,

"Because of the inherent disadvantage of laboratory experiments, RO$_2$ concentrations are always too high in the OFR, which has been pointed out in a previous study (Bianchi et al., 2019). The accretion reactions in the OFR are relatively more significant than it should be in the ambient atmosphere. We do not mean to compare HOM monomer and HOM dimer signals crossly here, but to pay attention to their formulae."

(Line 453), which reads,
"of the HOM monomer products (Figure 2a)"

and (Line 577 – 578), which reads,
"and C$_{18}$H$_{28}$O$_{10}$ contributed more than 50% of total HOM dimer signals at any OH exposure levels (Figure 2b)."

In addition, we have revised Figure 1 with stacked plots of the distributions of HOM products, and moved the original Figure S2 into the main text as Figure 2b.

(a)

[Figure]

(b)

[Figure]

**Figure 2.** Normalized signals of (a) HOM monomers and (b) HOM dimers versus OH exposure, which are fitted via a gamma function and shown in stacked.

***Q 1.11.*** *"Indeed, laboratory experiments show that RO₂ formed during the second-generation OH oxidation of the first-generation stabilized oxidation products can also undergo autoxidation reactions,"*
*→ This is extremely natural, as autoxidation is 'auto-catalytic oxidation' and mainly enabled by the loosening of the adjacent H-atoms next to the gained functional groups. Autoxidation inherently accelerates in many, if not all, chemical systems.*

**Response 1.11:**
Since Reviewer #1 agrees with our argument, nothing has been changed here.

***Q 1.12.*** *"High atmospheric concentrations of OH"*
*→ What is high atmospheric concentration to you? In the atmosphere [OH] is mostly buffered by [CO] and [CH4].*

**Response 1.12:**
Thanks for pointing out this vague expression. We have revised our manuscript (Line 121 – 124), which reads,

"OH with an atmospheric concentration up to $6\times10^6 – 2.6\times10^7$ molecule cm$^{-3}$, which is several times higher than the typical average atmospheric OH concentration, i.e., $1.5\times10^6$ molecule cm$^{-3}$ (Jacob, 1999), has been frequently observed in both urban and suburban environments in China (Tan et al., 2019; Lu et al., 2012)."

***Q 1.13.*** *Figure S4 has a good idea but is difficult to read with such a small scale.*

**Response 1.13:**
We have replotted this figure (Figure S3 in the revised supplement), as shown below,

[Figure]

**Figure S3.** Average mass spectrometry of HOMs detected by nitrate CIMS in the NOx experiments, presented with the averaged normalized signals in 1.8 ppb NO + 70 ppb NO2 and 4.8 ppb NO + 120 ppb NO2 experiments. For comparison, the mass spectrometry under the low NOx experiments is shown in opposite values.

***Q 1.14.*** *Was the aromatic sample illuminated with the same light source that was used for $N_2O$ photolysis? If so, then the influence of photochemistry is likely important for the results obtained.*

**Response 1.14:**

In fact, in our experiment settings, $N_2O$ did not photolyze, but reacted with $O(^1D)$ generated via photolysis of $O_3$ at 254 nm UV light, as stated in the original manuscript (Line 160 – 161). Our setting is different from those in previous studies where $N_2O$ was photolyzed with 185 nm UV light to form $O(^1D)$ and then generate NO (Lambe et al., 2017).

Our settings have been validated in a number of previous investigations (Lambe et al., 2017; Peng and Jimenez, 2020; Lambe et al., 2018). In fact, irradiance of 254 nm in the $NO_x$ experiments was no more than $1.8\times10^{15}$ photons $cm^{-2}$ s. The influence of such a low level of irradiance, i.e., photolysis of the parent aromatics and the first-generation products was not significant, as discussed in Response 1.2.

***Q 1.15.*** *You make a point that estimating HOM penetration through the system to the detector is difficult to quantify, yet it seems your calculations assume that 1,3,5-TMB and HOMs have similar losses in the system. This does not seem reasonable. How does this then influence the determined "nominal relative molar yields of HOMs"?*

**Response 1.15:**

As shown in Section S1 of the original manuscript, when we estimated the "nominal relative molar yields", the penetration efficiency of HOMs was set as $k_{loss}$, since the measured signals of HOMs were regarded not as true values at the exit of the OFR, but as values after diffusion loss.

On the other hand, we measured the concentration of 1,3,5-TMB at the exit of OFR by the Vocus-PTR through a sampling line, and regard the measured value as that at the exit of the OFR. Therefore, the loss coefficient of TMB through the system to the detector was assumed to be 0 $s^{-1}$. We did not assume HOMs and their parent compound have similar losses.

Meanwhile, the concept of "nominal relative molar yields of HOMs" has been removed in the revised manuscript as suggested by Reviewer #2.

***Q 1.16.*** *What do you mean by increase being monotonic or non-monotonic?*

**Response 1.16:**

A monotonic function is a function that is either entirely nonincreasing or nondecreasing. In other words, a function is monotonic if its first derivative does not change positive/negative signs and does not need be continuous (Royden and Fitzpatrick, 2018). Then, non-monotonic function is a function whose derivative changes positive/negative signs.

According to the gamma function fitting for HOM monomers and dimers, the derivatives of the fitting function of HOM monomers did not change sign during the intraday OH exposure, but the derivatives of HOM dimers did. Thus, we used the terms "increase monotonically or non-monotonically" to describe their behavior.

***Q 1.17.*** *Almost all the monomeric termination products in Scheme 1 have two strong H-bonding*

*functional groups (i.e., -OH and -OOH), and thus would be expected to be seen with nitrate ion charging (see, e.g., https://pubs.acs.org/doi/10.1021/acs.jpca.7b10015). Perhaps the proposed scheme is not correct?*

**Response 1.17:**

The proposed scheme is exactly MCM, except for compounds generated after autoxidation.

We double checked the mass spectrometry. Indeed, $C_9H_{12}O_4$, $C_9H_{14}O_4$, and $C_9H_{14}O_5$ were not detected by our nitrate CIMS. Among them, $C_9H_{12}O_4$ and $C_9H_{14}O_4$ were not reported in previous nitrate CIMS measurements, either (Molteni et al., 2018; Tsiligiannis et al., 2019). $C_9H_{14}O_5$ was reported to be detected by nitrate CIMS in previous studies (Molteni et al., 2018; Tsiligiannis et al., 2019), but not shown in our experiments. This phenomena is likely due to the relatively low detection efficiency of compounds with fewer than 5 oxygen atoms by nitrate CIMS, which has been illustrated in previous chamber experiments (Riva et al., 2019).

We have revised our manuscript (Line 448 - 451), which reads,

"The monomeric termination products of BPR, as shown in Scheme 1, were not detected by nitrate CIMS due to their low oxygen contents and thus relative low detection efficiencies in nitrate CIMS, which has been investigated in a previous study (Riva et al., 2019). Those …"

*Q 1.18. "because products from the secondary reactions cannot share the same structure as that of the one from the first-generation reaction."*
*→ Except perhaps in recycling or regeneration reactions. However, the important bit here is that you can make isomeric products, and the mass spectrometric detection utilized here would not separate them.*

**Response 1.18:**

Thanks for this excellent point.

We have revised our manuscript (Line 460 - 462), which reads,

"…the one from the first-generation reaction. However, limited by the inherent disadvantages of mass spectrometers, we could not distinguish isomers here and further illustrate their different chemical behaviors."

*Q 1.19. "$C_{18}H_{26}O_8$ can only be formed via the accretion reaction of two $C_9H_{13}O_5$·"*
*→ Nope. Could be, for example, through O3 and O7 radicals as well.*

**Response 1.19:**

Only one investigation studied accretion reactions of 1,3,5-TMB-derived BPR, $C_9H_{13}O_5$·, and its autoxidation product, $C_9H_{13}O_7$·, and reported formation mechanisms of $C_{18}H_{26}O_8$ and $C_{18}H_{26}O_{10}$ (Berndt et al., 2018). We propose that $C_{18}H_{26}O_8$ can only be formed via the accretion reaction between two $C_9H_{13}O_5$· in the original manuscript., while Reviewer #1 argued that it might be formed via the accretion reaction between $C_9H_{13}O_3$· and $C_9H_{13}O_7$· based on their formulae.

To our knowledge, $C_9H_{13}O_3$· can only be formed after addition of a hydroxyl radical to the aromatic ring of 1,3,5-TMB and a subsequent $O_2$ addition to the newly formed hydroxyl-substituted cyclohexadienyl radical (Vereecken, 2019). However, the lifetime of this radical is extremely short, as $C_9H_{13}O_3$· will undertake a ring-closure reaction and get attached by a $O_2$ very rapidly, forming

the BPR, $C_9H_{13}O_5$· (Vereecken, 2019; Wang et al., 2013, 2017; Li and Wang, 2014). Therefore, $C_9H_{13}O_3$· is not listed as a product of the OH oxidation of 1,3,5-TMB in the MCM (Jenkin et al., 2003; Vereecken, 2019). Indeed, the concentrations of this radical is very low as evidenced by the non-detection of $C_9H_{13}O_3$· in $NH_4^+$-CI3-TOF measurements in the experiments of Berndt et al. (2018), and are not likely to play an important role in the accretion reactions.

We've revised our manuscript (Line 561 - 567), which reads,

"$C_9H_{13}O_3$· is not likely to react with $C_9H_{13}O_7$· to form large amounts of $C_{18}H_{26}O_8$. $C_9H_{13}O_3$· can only be formed after addition of a hydroxyl radical to the aromatic ring of 1,3,5-TMB and a subsequent $O_2$ addition to the newly formed hydroxyl-substituted cyclohexadienyl radical (Vereecken, 2019). However, the lifetime of this radical is extremely short, as $C_9H_{13}O_3$· will undertake a ring-closure reaction and get attached by a $O_2$ very rapidly, forming BPR, $C_9H_{13}O_5$·. Its short lifetime and low concentration, as indicated by Berndt et al. (2018), lead to its insignificant role in the accretion reactions,."

***Q 1.20.*** *I don't understand what the point of the next sentence is: "There are currently no evidences supporting that C9H15Om· radicals can participate in the formation of HOM dimers with 28 hydrogens." Why would you expect the H15 radicals behave in a unique way? But also, supposedly none of the previous studies used as high OH dose, which would explain why such products were not observed. The general observation of dimers with H28 dominating seems worrying.*

**Response 1.20:**

We acknowledge that there are neither theoretical nor experimental evidences to support a unique behavior of $C_9H_{15}O_m$· radicals. As a result, HOM dimers with 28 hydrogen atoms could also be formed via the accretion of a $C_9H_{13}O_m$· radical and a $C_9H_{15}O_m$· radical. However, since a $C_9H_{15}O_m$· radical, as suggested by its hydrogen atom number, can only be formed via an OH addition to the stabilized $C_9H_{14}O_m$ products through multi-generation OH reactions, our conclusion that $C_{18}H_{28}O_x$ are multi-generation OH oxidation products still holds.

In fact, $C_{18}H_{28}O_x$ have been observed frequently in the OH-initiated oxidation of 1,3,5-TMB at low OH exposures (Molteni et al., 2018; Tsiligiannis et al., 2019), which should not be regarded as a sign of overoxidation.

Firstly, the OH dose in our experiments cannot be considered as high as those in previous laboratory experiments investigating HOMs' generation by OH-initiated oxidation of aromatics (Garmash et al., 2020), as discussed in our Response 1.1.Secondly, OH oxidation experiments of 30 ppb 1,3,5-TMB at a relatively low OH exposure, i.e., $3.5 \times 10^9$ molecule $cm^{-3}$ s, show that the total signals of $C_{18}H_{26}O_m$ was close to those of $C_{18}H_{28}O_m$ (Tsiligiannis et al., 2019), representing 8.5% and 7.1%, respectively, of total signals detected by the nitrate CIMS.

Among all the reported experiments that have investigated the distribution of HOMs generated from the OH-initiated oxidation of 1,3,5-TMB (Molteni et al., 2018; Wang et al., 2020; Tsiligiannis et al., 2019), the Molteni et al. (2018) study used the lowest OH dose and precursor concentrations. Nevertheless, $C_{18}H_{28}O_x$ and $C_{18}H_{30}O_x$, in addition to the multigeneration OH product of $C_9H_{16}O_x$, were also detected, which indicate that the oxidation of stabilized products can start at a very early stage.

We've revised our manuscript (Line 584 - 589), which reads,

"In addition, $C_{18}H_{28}O_x$ can also be formed through accretion of a $C_9H_{13}O_m$· radical and a

$C_9H_{15}O_m\cdot$ radical, as suggested by previous studies (Molteni et al., 2018; Wang et al., 2020; Tsiligiannis et al., 2019). However, since a $C_9H_{15}O_m\cdot$ radical, as suggested by its hydrogen atom number, can only be formed via an OH addition to the stabilized $C_9H_{14}O_m$ products through multi-generation OH reactions, our conclusion that $C_{18}H_{28}O_x$ are multi-generation OH oxidation products still holds. …".

*Q 1.21. It seems worrying that the dimer products decrease already at such a short reaction times. This seems to amply indicate how skewed the chemical system is and that either further chemical processing, or aerosol formation, reduced the dimer yield.*

**Response 1.21:**

Please refer to our Response 1.1 and 1.10.

The aerosol formation is unlikely to play a role, as particles generated in the PAM OFR were limited.

The exact appearance time of the maximum concentrations of HOM dimers is dependent on the formation rate and loss rate. The formation rate and loss rate were not accelerated equally. On the other hand, the loss pathways of HOM dimers were not exactly the same as the ambient due to the lack of aerosols in the OFR. With the decrease of particulate pollution and thus condensation sinks in the polluted areas, the physical loss of HOMs might be lower and the chemical process can be more important in the ambient.

This series of experiments are not meant to specifically find out the detailed OH exposures when the maximum concentrations of HOM dimers will occur , but try to indicate how HOM dimers evolve with the increase of OH exposures in chemistry. This work can be regarded an indicator for the potential chemical fates of HOM dimers in the atmosphere.

Meanwhile, the concept of "nominal relative molar yields of HOMs" has been removed in the revised manuscript.

We have revised our manuscript (Line 600 - 614) as:

"This decrease of dimer at relatively high OH exposures are likely due to the accelerated accretion reactions in the OFR, resulted by the high $RO_2$ concentrations. The HOM dimers are formed earlier compared to under ambient conditions and then can go through the further oxidation reactions. Note that this does not mean the maximum concentrations of HOM dimers will also accurately occur at the same OH exposures in the atmosphere, because the detailed appearance time of the maximum concentrations of HOM dimers is dependent on their formation rate and loss rate. In our experiments, the formation rate and loss rate were not accelerated equally. On the other hand, the loss pathways of HOM dimers were not exactly the same as the ambient due to the lack of aerosols in the OFR. With the decrease of particulate pollution and thus condensation sinks in the polluted areas, the physical loss of HOMs might be lower and the chemical process can be more important. This series of experiments are not meant to specifically find out the detailed OH exposures when the maximum concentrations of HOM dimers will occur, but try to indicate how HOM dimers evolve with the increase of OH exposures. This work can be regarded as an indicator for the potential chemical fates of HOM dimers in the atmosphere.
"

***Q 1.22.*** *A OH:HO$_2$ ratio is given two times although it should presumably be RO$_2$:HO$_2$*

**Response 1.22:**

Sorry for this mistake. We have revised our manuscript (Line 640), which reads,
"Such a high HO$_2$: RO$_2$ ratio condition is typically difficult to …"

and Line (642 – 644), which reads,
"This is exactly the case for our experiments, but its influences on our conclusion were tiny, as have been discussed in the Section 3.1."

***Q 1.23.*** *Consider the part: "In addition, high concentrations of radicals might also terminate the RO2 chain earlier, which inhibits the autoxidation reactions in the PAM OFR." This is true. The RO2 lifetime is critically shortened likely inhibiting normally competitive H-shift isomerization reactions. Then consider: "However, these could only influence the distribution of oxidation products at most, and would not affect the chemical behaviors of HOMs under different OH exposures." This is not true. Both conditions favor oxidation of the aromatic parent molecule, but the same HOMs are unlikely to form under so different oxidation conditions.*

**Response 1.23:**

As stated in our Response 1.1, autoxidation always dominates the fates of RO$_2$, in both laboratory experiments and ambient atmosphere, because of its rapid reaction rate constant. Meanwhile, because of the similar RO$_2$ lifetimes between in the laboratory and in the ambient, the RO$_2$ lifetime is not "critically shortened".

In our experiments either with or without NO$_x$, the ratios between different radicals were similar to those in the ambient, whilst NO:RO$_2$ in the laboratory experiments was lower compared to the ambient value, which means that the bimolecular reactions except for RO$_2$ + NO were accelerated similarly. Nevertheless, the monomeric organonitrates generated in our experiments should have the same formulae as those generated in the ambient though in a lower yield, because their formation pathways were not influenced. The existence of multi-generation OH oxidation can be confirmed via the detection of compounds with multiple nitrogen atoms. On the other hand, compounds generated via *R1 - R7* in the lab will also be generated in the ambient, though their proportions were smaller in the ambient because of the dominant reaction channel of RO$_2$ + NO (*R8 - R9*).

Therefore, differences between the laboratory experiments and the ambient exist, which leads to differences in the distribution of products. However, differences in the distribution of products will not change our conclusion that considerable HOMs can be generated by multi-generation within an intraday OH exposure.

We have revised our manuscript (Line 644 - 648), which reads:
" … in the Section 3.1. Therefore, the difference in the distribution of products will not change our conclusion."

***Q 1.24.*** *"The OH reaction rate for C18H26O8 should be around twice of these values, as there are two*

*C=C bonds in its structure. Our calculation result is consistent with this estimation."*
*→ This seems extremely unlikely as the indicated rate is already basically at the collision limit and the big dimer compound is sterically hindered, which would imply a lower reaction rate.*

**Response 1.24:**

The collision limit at the room temperature is around $9 \times 10^{-10}$ molecule$^{-1}$ cm$^3$ s$^{-1}$ (Molteni et al., 2019), which is still much larger than our estimated reaction rate constants. To be rigorous, we now only emphasize on the fast reaction rate constant of $C_{18}H_{26}O_8$ instead of an exact estimation value.

We have revised our manuscript (Line 659 – 660), which reads,

"The OH reaction rate for $C_{18}H_{26}O_8$ should also be fast due to the C=C bonds in its structure, which is activated by the adjacent functionalities."

*Q 1.25. "because the NO termination reaction of RO₂ is the only pathway that can generate organonitrates"*
*→ Why would NO₃ or NO₂ chemistry not form organonitrates?*

**Response 1.25:**

We meant to suggest that the NO termination reaction of $RO_2$ is the only pathway that can efficiently generate organonitrates in our experiments. Organonitrates formed via reactions between $NO_2$ and $RO_2$ are believed to be unstable. On the other hand, concentrations of $NO_3$ were quite low (< 1 pptv estimated by the modified PAM_chem_v8) in our system because of the existence of decent concentrations of NO, which would react with $NO_3$ at a rapid reaction rate, i.e., $2.7 \times 10^{-11}$ molecule$^{-1}$ cm$^3$ s$^{-1}$ (IUPAC dataset, https://iupac-aeris.ipsl.fr, last access: 26 October 2023). Therefore, reactions between $NO_3$ and $RO_2$ would not generate organonitrates notably, either.

We've revised our manuscript (Line 533 – 543), which reads,

"…because the NO termination reaction of $RO_2$ is the only pathway that can generate sufficient amounts of organonitrates in our experiments and …, as indicated in Scheme 2. $RO_2$ can react with $NO_2$ to form peroxynitrates ($ROONO_2$) but these species are thermally unstable except at very low temperatures or when the $RO_2$ is an acylperoxy radical (Orlando and Tyndall, 2012), neither of which were not met in our experiments. The concentrations of $NO_3$ are estimated to be lower than 1 pptv by our modified PAM_chem_v8 because of the existence of decent concentrations of NO, which would consume $NO_3$ with a rapid reaction rate constant, i.e., $2.7 \times 10^{-11}$ molecule$^{-1}$ cm$^3$ s$^{-1}$ (IUPAC dataset , https://iupac-aeris.ipsl.fr, last access: 26 October 2023). Therefore, $NO_2$ and $NO_3$ were unlikely to react with $RO_2$ to form large amounts of organonitrates in our experiments."

*Q 1.26. A strange comment considering previous literature: "since no evidence supports that a nitrogen-containing monomeric RO2 can go through accretion reactions.*

**Response 1.26:**

We acknowledge that we could not provide strong evidences for this point. Either a $C_9H_{15}O_m\cdot$ radical and a $C_9H_{12}NO_m\cdot$ radical, or a $C_9H_{13}O_m\cdot$ radical and a $C_9H_{14}NO_m\cdot$ radical can react to form a $C_{18}H_{27}NO_m$.

We have revised our manuscript (Line 664 – 668), which reads,

[revised manuscript text omitted]

---

## Author Comment (AC2)

**Reviewer #2**

*Wang et al presents laboratory results where the authors oxidized thrimethylbenzene (TMB) in an oxidation flow reactor (OFR) with OH in the presence and absence of NOx to investigate the role of autoxidation. Using a combination of a Vocus proton transfer reaction mass spectrometer and nitrate chemical ionization mass spectrometer, the authors investigated the products produced from the oxidation of TMB. They argue that the highly oxidized material (HOMs), which has been observed and discussed to be potentially important for new particle formation and particle growth in both clean (e.g., Boreal forests) and polluted environments, is produced upon the second OH oxidation and subsequent reactions of the material instead of the first oxidation. They argue that the NO$_x$ products observed in the NO experiments provides evidence of this.*

*This paper is of interest to the community and recommend publication after addressing the comments below and potentially restructuring/rephrasing some of the conclusions to address the concerns Matti provided (though disagree with the argument that the OFR is not a tool to be used to understand chemistry and its only role is regulatory).*

**Response:**

We are very grateful for the comments from Reviewer #2 and have now revised our manuscript accordingly.

**Major**

**Q 2.1.** *Methods--how long is one condition sampled to ensure things have reached steady state for the calculations (e.g., lifetime of HOMs) to be true? This is also important for the assumption that is used to normalize the signal measured by HOMs divided by the TMB signal with Vocus (which is currently unclear why authors may not have used a signal from the nitrate CIMS instead that was constant).*

**Response 2.1:**

Once the signals of certain HOM are more than 3 standard deviations of its background signals, we believe that it is positively generated in our system. If the fluctuations in the 1-min-averaged signals of both TMB in the Vocus PTR and typical HOMs (i.e., $C_9H_{14}O_7(NO_3)^-$) in the nitrate CIMS are within 2% during a 10-min period, we assume that a steady state has been reached. We typically sampled for around 20 minutes for each experiment after the adjustment of UV lights. It typically only took around no more than 2 minutes for the signals of HOMs to stabilize after the adjustment of UV lights. This observation is similar to PAM OFR results from other groups (Figure S1 of Cheng et al., 2023).

Our TMB source is a home-made cylinder containing certain concentrations of TMB. The initial concentrations of TMB utilized in the experiments fluctuated slightly as determined with a calibrated Vocus PTR, which resulted from the sample preparation processes, but generally were around 50 ppbv. We tried to minimize potential influences of the differences in the initial TMB concentration on the signals of HOMs by normalizing the HOMs signals with the initial TMB signals..

Meanwhile, the concept of "nominal relative molar yields of HOMs" has been removed in the revised manuscript.

We have revised our manuscript (Line 168 - 174), which reads:

"…$O(^1D) + H_2O \rightarrow 2OH$. After turning on of UV lights, a certain HOM compound is believed to be generated if its signal is more than 3 standard deviations of its background signal. If the

fluctuations in the 1-min-averaged signals of both TMB in the Vocus PTR and typical HOMs (i.e., $C_9H_{14}O_7(NO_3)^-$) in the nitrate CIMS are within 2% during a 10-min period, we assume that a steady state has been reached. It usually took around no more than 2 minutes for the signals of HOMs to stabilize after the adjustment of UV lights. We typically monitored the reaction products for around 20 minutes for each experiment. An ozone …"

and (Line 284 - 288), which reads:

"…of a significant uncertainty. The initial concentrations of TMB utilized in the experiments fluctuated slightly, which resulted from sample preparation processes, but generally were around 50 ppbv. We tried to minimize potential influences of the differences in the initial TMB concentrations on the signals of HOMs by normalizing the HOMs signals with the initial TMB signal. To precisely illustrate …"

**Q 2.2** *Were any experiments conducted at lower or higher TMB mixing ratios? E.g., this may help address some of the concerns of Matti as lower/higher TMB should change the $RO_2/HO_2$ ratios and provide insight into the chemistry that is occurring in the OFR.*

**Response 2.2:**

As discussed in our Response 1.1, the $RO_2/OH$ and $HO_2/OH$, as well as $RO_2/HO_2$ in our experiments were generally similar to those in the urban Beijing. Theoretically, at a given RH and UV (i.e., a given OH), an increase in the initial TMB would lead to formation of more $RO_2$, which corresponds to a larger $RO_2/OH$. However, under our experimental conditions, the $RO_2/OH/HO_2$ channels of $RO_2$ radicals are always minor, and thus an increase in $RO_2/OH$ would not have a significant impact on the relative distribution of products formed from these channels.

We compared product MS for experiments with a similar OH exposure but different initial concentrations of TMB (e.g., Exp. 3 v.s. Exp. 19, and Exp. 12 v.s. Exp. 22). Clearly, the relative distributions of products in these experiments are quite similar except for abundance differences for a few peaks, indicating a minor difference in the relative distributions of products caused by fluctuations of initial concentrations of TMB.

We have revised our manuscript (Line 396 - 411) , which reads

"Theoretically, at a given RH and UV (i.e., a given OH), an increase in the initial TMB would lead to formation of more $RO_2$, which corresponds to a larger $RO_2/OH$. However, under our experimental conditions, the $RO_2/OH/HO_2$ channels of $RO_2$ radicals are always minor, and thus an increase in $RO_2/OH$ would not have a significant impact on the relative distribution of products formed from these channels We compared product MS for experiments with a similar OH exposure but different initial concentrations of TMB (e.g., Exp. 3 v.s. Exp. 19, and Exp. 12 v.s. Exp. 22). The OH exposures of Exp. 3 and Exp. 19 were estimated by the modified PAM_chem_v8 model to be $5.2\times10^9$ and $5.3\times10^9$ molecule $cm^{-3}$ s, respectively, but the initial concentration of TMB of Exp. 3 was 25% more than that in Exp. 19. Meanwhile, the OH exposures of Exp. 12 and Exp. 22 were $4.5\times10^{10}$ and $4.4\times10^{10}$ molecule $cm^{-3}$ s, respectively, but the initial concentration of TMB of Exp. 12 was 48% more than that in Exp. 22. Comparisons between the product MS of Exp. 3 and Exp. 19 (Figure S2), as well as of Exp. 12 and Exp. 22, show that increase in the initial concentration of precursors generally resulted in a minor increment in the absolute signals of HOMs. Clearly, the relative distributions of products in these experiments are quite similar, indicating a minor difference in the relative distributions of products caused by fluctuations of initial concentrations of TMB."

(a)

[Figure]

(b)

[Figure]

(c)

[Figure]

(d)

[Figure]

Figure S2. Comparison of the (a) monomer product MS between Exp. 3 and Exp .19, (b) dimer product MS between Exp. 3 and Exp. 19, (c) monomer product MS between Exp .12 and Exp .22, and (d) dimer product MS between Exp .12 and Exp .22. The signals of HOMs were raw ones in the nitrate CIMS.

*Q 2.3. Discussion of the mechanism: The purpose of the paper is to elucidate the mechanism of the production of HOMs. However, the authors only present the scheme from MCM without expanding the mechanism/scheme they believe they have observed, which makes the narrative very hard to follow. I strongly recommend expanding the schemes presented in the paper with the chemistry and products observed to improve the narrative and better understand how the second generation HOMs are being formed. To address the concerns of Matti, this can address both the chemistry that may be occurring in the OFR vs the chemistry that may be more prominent in urban atmosphere and the importance/products between the two regimes. Further, I think interspersing the results from the NOx chemistry into the discussion of the production of HOMs and which pathways occur would be beneficial instead of the NOx chemistry being a separate section. Right now, the NOx chemistry seems like a leftover section that is addressed to quickly instead of being used as a tool to verify the hypothesis that it is potentially a second OH attack is necessary to form the HOM.*

**Response 2.3:**

We are very grateful to the helpful suggestions of Reviewer #2. In the evised manuscript, we have proposed formation pathways of HOM monomers with the highest signals observed in our experiments, which are generated with the involvement of multi-generation OH oxidation.

We have revised our manuscript (Line 467 - 469), which reads:
", which can be genreated by an OH attack to $C_9H_{14}O_5$ (Scheme 2), the hydroperoxyl termination product of the BPR $C_9H_{13}O_5$·.
"
and (Line 472), which reads:

[revised manuscript text omitted]

…

"

***Q 2.4.*** *Units in the normalized relative molar yields: Currently, all figures that show the nominal relative molar yields are not intuitive to interpret and understand. E.g., the values in the y-axis are between 10^-10 to 10^-9, which would suggest that the HOMs are not important fates. It is not clear if it is due to taking the signal from the nitrate CIMS and normalizing to the signal from the Vocus may be the cause of this. Further, it is surprising the yields are apparently higher from the accretion (RO2+RO2) reactions compared to the monomer reactions. Due to the general lack of clarity and the concerns from reviewer number one, it may be better to focus on the fate of RO2 during these experiments instead of the yields, and which fates are more atmospherically relevant vs potentially related to the OFR. I would recommend also, to address the reviewers concerns, to include the estimated fate of the RO2 products due to fragmentation, photolysis, and wall loss in OFR.*

**Response 2.4:**

We are very grateful to these helpful suggestions. We have revised our manuscript by including "Session 3.1 Validation of experimental settings" that focuses on the estimation of radical concentrations and fates of $RO_2$. The concept of "nominal molar yield of HOMs" has been removed in the revised manuscript to address both reviewers' concerns. Please also refer to our Response 1.1.

**Minor**

***Q 2.5.*** *Color scheme. Please avoid using red and green in the same plot, as that will be difficult to interpret for color blind people.*

**Response 2.5:**

We have revised the Figure 1, Figure 2, Figure 3, Scheme 1, and Scheme 2 to address the concerns of Reviewer #1. The revised Figures avoid potential color scheme problems as mentioned. Please also refer to our Response 1.10 for Figure 1, as well as Response 1.9 for Scheme 1 and Scheme 2,.

We have revised Figure 2 and Figure 3, which reads:

"

(a)

[Figure]

(b)

[Figure]

**Figure 2.** Normalized signals of (a) $C_9H_{14}O_7$, $C_9H_{16}O_7$, and $C_9H_{16}O_8$ and (b) $C_9H_{14}O_8$, $C_9H_{16}O_8$, and $C_9H_{16}O_9$ measured at the exit of OFR in experiments without $NO_x$ as a function of OH exposure. $C_9H_{16}O_8$ are shown in both plots to better illustrate the chemical profiles of different compound groups.

(a)

[Figure]

(b)

[Figure]

**Figure 3.** Normalized signals of (a) $C_{18}H_{26}O_{12}$, $C_{18}H_{28}O_{12}$, and $C_{18}H_{28}O_{13}$, and (b) $C_{18}H_{26}O_{10}$, $C_{18}H_{28}O_{10}$, and $C_{18}H_{28}O_{11}$ measured at the exit of OFR in experiments without $NO_x$ as a function of OH exposure.

"

**Q 2.6.** *Ensure that SI figures are presented in same order discussed in paper (e.g., Fig. S4 is discussed after Fig. S5).*

**Response 2.6:**

Sorry for this. We have double-checked our manuscript to make sure that figures are presented in the same order as discussed in paper.

**Q 2.7.** *Line 335: declining instead of declination*

**Response 2.7:**

Since we have removed the nominal molar yield part in the manuscript. This word has been removed.

**Q 2.8.** *Line 350 - 363: It is confusing which ratio is being discussed as it is switched from HO2:RO2 to OH:HO2. Please clarify (and may be addressed with the rephrasing of products/RO2 fates).*

**Response 2.8:**

Thanks for the suggestions. We have revised this. Please refer to our Response 1.22.

We also focus on fates of $RO_2$ and the detailed branching ratios of their termination reactions in Session 3.1 and Supplementary Text S1 of revised manuscript. Please refer to our Response 1.1.

---

## Referee Report (RR1)

**2nd round review of Wang et al.,**

I thank the Authors for considering my comments on the work. While I do feel that several of my concerns were not really answered, I nevertheless appreciate the invested effort.

I am still not convinced that PAM type OFR setups could be used to study atmospheric relevant chemical mechanisms due to their high oxidant load conditions, and the usage of high doses of UV light, especially in connection to studying aromatic compound oxidation reactions. PAM type setups are apparently useful for deriving emission regulations as it is possible to, at least to an extent, estimate the amount of SOA produced during a long atmospheric processing time. These systems, by design, deviate from ambient concentration ranges, and thus are generally unsuited to study ambient relevant reaction mechanisms.

Also, I would advise the authors to be careful what they consider as an "ORF setup". It seems they have considered OFR to be everything where oxidation is studied, notwithstanding that several other setups have been previously meticulously characterized and assigned as well-controlled flow reactors and environmental chambers that aim to investigate the processes under ambient relevant conditions. Especially the setups mentioned in the response letter, the Jülich JPAC chambers and the very carefully controlled free-jet flow reactor of Torsten Berndt in TROPOS, Leipzig. In JPAC (e.g., Garmash et al., 2020) the reaction time is much larger (ca. 50 minutes), surface-to-volume smaller and the oxidation mixture more dilute, leading to more ambient relevant conditions. In TROPOS (e.g., Wang et al., 2017) the reactions have been studied in short reaction times (at 7.9 s) in a practically wall-less setup with careful control over the reagent concentrations. Equating these setups and works with the PAM and the current work appears wrong.

The chosen autoxidation rate of 7 s^-1 is very high, and not supported by the referenced study. The Authors state they based this on the expected autoxidation rate of ethylbenzene, yet this higher rate in the referenced work is due to a secondary hydrogen on the ethyl substituent, which is absent from methyl groups in the studied TMB molecule. Instead, the autoxidation rate from a methyl group was found 270 times slower (7 s^-1 / 0.026 s^-1). If one takes into account the symmetry with three methyl groups, then on could expect the 1,3,5-TMB rate as around 3 times higher, yet still likely less than 0.1 s^-1. Thus, this choice necessarily leads to severely overestimated fraction of autoxidized products. In fact, the Authors even state that "We arbitrarily set the autoxidation reaction rate…", and this seems to be critical for the analysis, leading to conclusion that 81% to 89.7% of the reaction products are due to autoxidation.

Similarly, as the pool of simulated RO2 consists only of the two prominent RO2 radicals (i.e., BPR and C9H13O7) which both have been assigned with a very high accretion rate, and then the whole pool of ROOR is counted based on these same two radicals. That would also seem to indicate an overestimated importance for the accretion reaction, or do I read it wrong? Furthermore, I still find the decreasing ROOR signal problematic, as under such short timescales it would seem to indicate either particle formation, or secondary reactions of the formed ROOR, neither which would be expected under atmospheric conditions at such short times. It is stated that no particle formation was observed with long SMPS, but I suppose the

SMPS cut-off limit is so high that it would not see the fresh particles in any case. Or what was the smallest measurable particle size with the SMPS system?

The Authors seem to several times make the claim that the OH concentration in the work of Garmash et al., was higher, which is not the case. Even the highest [OH] = $4.5 \times 10^8$ cm^-3 was still considerably smaller than reported in the current study (i.e., $1.5 \ 10^9$ cm^-3). Moreover, in the work of Garmash et al., it was recognized that with such an unrealistically high OH, you can't really derive sound mechanistic conclusions, and thus no mechanisms were proposed. However, the fact that the rate coefficients increase together with the number of OH substituents was recognized during the study, which is why the potential for "multi-generation OH oxidation" in product formation was acknowledged as a complication for data analysis, and no other mechanistic description was provided due to the non-conventional reaction conditions.

This comparison seems in any case problematic, as the Authors talk about 0.7-6.9 hours of atmospheric aging at $2 \times 10^6$ cm^-3 [OH], yet the residence time in the setup is only 53 seconds. The exact [OH] used seemed to be unknown, yet with the response letter the Authors quote an OH concentration of $1.5 \times 10^9$ cm^-3. Now, in Garmash et al., the residence time is roughly 60 times longer and the maximum used [OH] is $4.5 \times 10^8$ cm^-3, naturally leading to longer aging timescale. What the Authors seem to miss here is that what is more important is the absolute concentrations of the reagents, as they push the chemistry to an unrealistic regime, and not the equivalent dose. And repeating from above, Garmash et al., did not derive mechanistic conclusion from the higher [OH] experiments, other than the sequential oxidation behavior.

So, my problem with the current documentation is that PAM, and the OFR approach, is really a methodology for reaching emission regulations, but it's ill-suited for detailing complex chemistry under atmospheric relevant concentration regimes. It would make an interesting comparison to see how the products change between atmospheric $10^6$ to $10^7$ cm^-3 to OFR $10^9$ cm^-3, but unfortunately such an analysis was not provided. (*To put it another way, if you would have written the story like "if HOMs would survive a day – what would happen", and not trying to insist this is strictly atmospherically relevant product distribution, then this would have made much more sense*). Also, the Authors probably slightly misunderstood my comment about the potential influence of light: It would be expected to be a problem for the aromatic oxidation products undergoing photo-oxidation, not that the precursor TMB would be photolyzed. Also, I don't understand the comment that says: "Meanwhile, photolysis of HOMs can lead to decomposition, decreasing detected signals of HOMs, but unlikely to generate new HOMs." If the photolysis yields radical intermediates, as it does, then these new intermediates can continue oxidation to HOM as well. It's quite likely that this was occurring in the experiments.

Thanks for educating me that in PAM a ¼'' Teflon tube is used. This is not adequate for measuring very condensable products. This is a problem I had overlooked in the common PAM methodology.

Unfortunately, with these shortcomings I can't support publishing the work.

---

## Author Response (AR2)

**RE: A point-to-point response to reviewers' comments**

"Secondary OH reactions of aromatics-derived oxygenated organic molecules lead to plentiful highly oxygenated organic molecules within an intraday OH exposure" by Yuwei Wang, Chuang Li, Ying Zhang, Yueyang Li, Gan Yang, Xueyan Yang, Yizhen Wu, Lei Yao, Hefeng Zhang, Lin Wang (egusphere-2023-1702)

Dear Dr. Liggio,

We are very grateful for the comments from both reviewers and the revision opportunity you offered to improve our work. To address the concerns raised by the reviewer, extra experiments under atmospheric relevant OH concentrations ([OH]) have been conducted to make comparisons between results from high and low [OH] experiments. Nevertheless, this manuscript does not suggest that the product distributions are identical between ambient conditions and laboratory experiments, but is meant to elucidate the potential or ability of OOMs to generate more diverse HOMs if they survive for a long period of time.

A point-to-point response to the comments, which are repeated in *italic*, is given below.

We are looking forward to your decision at your earliest convenience.

Best regards,

Lin Wang
Fudan University
lin_wang@fudan.edu.cn

*Reviewer #1: I thank the Authors for considering my comments on the work. While I do feel that several of my concerns were not really answered, I nevertheless appreciate the invested effort.*

*I am still not convinced that PAM type OFR setups could be used to study atmospheric relevant chemical mechanisms due to their high oxidant load conditions, and the usage of high doses of UV light, especially in connection to studying aromatic compound oxidation reactions. PAM type setups are apparently useful for deriving emission regulations as it is possible to, at least to an extent, estimate the amount of SOA produced during a long atmospheric processing time. These systems, by design, deviate from ambient concentration ranges, and thus are generally unsuited to study ambient relevant reaction mechanisms.*

**R1.** We thank the reviewer for his comments. To address concerns raised by the reviewer on the high OH concentrations ([OH]) used in our experiments, extra experiments under atmospheric relevant $OH/HO_2/RO_2$ radical concentrations were conducted. Results in the new experiments show that the multi-generation OH oxidation and the corresponding products are manifest at atmospheric relevant [OH] ($\sim 10^7$ molecule $cm^{-3}$), even when the OH exposure is as low as $5.86 \times 10^8$ molecule $cm^{-3}$ s. The comparison between oxidation products detected in the low [OH] and high [OH] laboratory experiments show a shift in the relative abundance toward compounds with more hydrogen atoms under high [OH] experiments, which confirms that these compounds are strongly related to the accelerated $RO_2$ termination and multi-generation OH oxidation.

We agree with the reviewer that the conclusion of our high [OH] experiments is limited to show the potential or ability of aromatics-derived, stabilized OH oxidation products to generate more diverse HOM products by the re-initiation of OH oxidation, i.e., the existence of the sub-sequential oxidation of stabilized oxidation products. If the survival time of a specific OOM product permits, the re-initiation of its OH oxidation, i.e., $OOM + OH \rightarrow new\text{-}RO_2$, can be deduced/confirmed with the detection of compounds with higher hydrogen contents or nitrogen contents, i.e., lower double bond equivalence (DBE, calculated as

$nC - \frac{nH+nN}{2} + 1$ where $nC$, $nH$, and nN stand for the number of carbon, hydrogen, and nitrogen atoms in a

molecule, respectively). While supported to some extent by our new experiments with atmospheric relevant [OH], the existence of multi-generation OH oxidation of aromatics-derived OOMs under ambient conditions still necessitates further investigation, because of the significant variations in the detailed radical ratios, survival times of OOMs in the real atmosphere, and the exact atmospheric OH exposures under the diverse ambient conditions that cannot be captured solely through the current high [OH] experiments.

Nevertheless, this manuscript is meant to elucidate the potential formation pathways of HOMs, which are now supported by both high [OH] and atmospheric relevant [OH] experiments, if these HOMs were indeed formed and observed in ambient. The product distribution under ambient OH exposure, is not expected to be identical to that from OFR experiments with high [OH].

**R 1.1. Methods of new experiments with atmospheric relevant [OH]**

Seven extra experiments were conducted, four without $NO_x$ and three with $NO_x$. Their experimental conditions are listed in Table R1. Hereafter, we refer to the previous series of experiments as 'the $1^{st}$-round experiments' and the new series as 'the $2^{nd}$-round experiments', respectively. The $i^{th}$ experiment in the $1^{st}$-round experiments is labelled as 1-*i* and the one in the $2^{nd}$-round experiments as 2-*i*, where *i* stands for its serial number.

**Table R1.** Summary of the 2nd-round experiments.

| | Initial concentration of 1,3,5-TMB (ppb) | $O_3$ concentration (ppb)[*] | NO concentration (ppb) | $NO_2$ concentration (ppb) | Estimated OH exposure based on the precursor consumption ( × $10^8$ molecule cm$^{-3}$ s) |
|---|---|---|---|---|---|
| 2-1 | 30.8 | 147 | 0 | 0 | 54.7 |
| 2-2 | 30.8 | 151 | 0 | 0 | 23.8 |
| 2-3 | 30.8 | 152 | 0 | 0 | 9.56 |
| 2-4 | 30.8 | 152 | 0 | 0 | 5.86 |
| 2-5 | 34.5 | 123 | 7.11 | 38 | 31.1 |
| 2-6 | 34.5 | 137 | 3.10 | 21 | 19.7 |
| 2-7 | 34.5 | 144 | 1.30 | 11 | 8.83 |

[*] $O_3$ concentrations were stable values measured after the lights were turned on.

The experimental methods for the 2nd-round experiments have been updated according to either the comments of the reviewer or the availability of instruments.

In the 2nd-round experiments, a Vocus CI-TOF (Towerk AG, Switzerland) equipped with a Vocus Aim inlet and the same nitrate-ion chemical ionization source as adopted in the 1st-round experiments was utilized to measure oxidation products, hereafter referred as nitrate CI-TOF. The nitrate CI-TOF is characterized with a flat transmission efficiency between m/z 60 Th and m/z 500 Th, as well as a mass resolution of 10000 at m/z 200 Th.

We kept the reactor a plug flow one, and kept the residence time the same as in 1st-round experiments. The sampling system of our PAM OFR has been updated with the help of Aerodyne Research Inc. We replaced the sampling port at the exit of PAM OFR, and the reaction products were sampled from the PAM OFR via a 30 cm-long Teflon tube with a 1/2 in. outer diameter (OD) to our nitrate CI-TOF in the 2nd-round experiments. The Vocus PTR and the ozone monitor were connected to the PAM OFR from a separate port via a 120 cm-long Teflon tube with a 1/4 in. OD.

The relative humidity was kept around 20%. The initial concentration of $O_3$, light intensities of 254 nm UV lamps, and the initial concentration of 1,3,5-TMB were all tuned down to make the conditions as close to the ambient as possible. An experiment with a lamp intensity and a concentration of $O_3$ lower than those in Exp. 2-4 was tested but very few convinced signals of HOMs were detected. In the $NO_x$ experiments, due to the low ($O^1D$) in the PAM OFR, 2.5 slpm pure $N_2O$ was utilized instead of the previous one (350 sccm), whereas the total flow was kept the same as in the 1st-round experiments. The average [OH] in the PAM OFR was estimated with PAM_chem_v8 constrained by the precursor consumption and measured $O_3$ concentrations. The autoxidation reaction rate of BPR has been updated to be 0.078 s$^{-1}$ in the model as discussed below, and the results estimated by this model for the 1st-round experiments have also been updated using the updated autoxidation rate.

Contents in this part have been incorporated into **Section 2** and **the supplement** of the revised manuscript (version R2), which reads:

Line (146 - 174):

[revised manuscript text omitted]
. 2-7 with an atmospheric relevant [OH], as shown in Figure R1c (Figure 5c in the revised manuscript, version R2). From the perspective of molecular formula, $C_9H_{14}N_2O_{10}$ is also one of the most frequently observed multi-nitrogen-containing compound in polluted atmospheres, whose seasonal variations show a good correlation with [OH] (Guo et al., 2022; Yang et al., 2023).

We compared the difference in product distributions between Exp. 2-3 ([OH] = ~$1.69\times10^7$ molecule $cm^{-3}$) and Exp. 2-1 ([OH] = ~$1.03\times10^8$ molecule $cm^{-3}$), as well as between Exp. 2-3 and Exp. 1-12 ([OH] = ~$8.47\times10^8$ molecule $cm^{-3}$) in Figure R2 (also as Figure 6 in the revised manuscript, version R2). The normalized abundance was obtained by normalizing all the products to the most abundant one in each experiment, i.e., $C_{18}H_{26}O_{10}$ in Exp. 2-1 and Exp. 2-3, and $C_9H_{14}O_7$ in Exp. 1-12. The changes in the normalized abundance were obtained by subtracting the normalized abundance in Exp. 2-1 from that in Exp. 2-3, and Exp. 1-12 from Exp. 2-3. As the [OH] and OH exposure increased, there was a noticeable rise in the relative abundance of more oxygenated compounds, which can be attributed to the more intensive proportion of multi-generation OH oxidation in high OH exposure experiments. This comparison demonstrates the capacity and potential of multi-generation OH oxidation to reduce DBE and elevate the oxygenated levels of oxidation products.

The changes in the normalized abundance of C9 and C18  do not correspond proportionally to the observed alterations in their absolute abundance. For example, the signals of $C_9H_xO_{11}$ ($x$ = 12 - 15) products were actually higher in Exp. 2-1 than those in Exp. 2-3, but their normalized abundance is lower, which is due the faster increase in the signal of the most abundant compound $C_{18}H_{26}O_{10}$ in these experiments.

**(a)**

[Figure]

**(b)**

[Figure]

**(c)**

[Figure]

**Figure R1.** Distributions of C9 and C18 products detected by nitrate CI-TOF in (a) Exp. 2-3, (b) Exp. 2-4, and (c) Exp. 2-7. The reagent ion, $NO_3^-$, is omitted in the label for the molecular formula. Important radicals were labelled in pink. Note that no convinced signals of HOM dimers were observed in the 2nd-round experiments with $NO_x$.

[Figure]

[Figure]

**Figure R2.** The changes in the normalized abundance of C9 and C18 products observed by nitrate CI-TOF in (a) Exp. 2-3 relative to Exp. 2-1, and (b) Exp. 2-3 relative to Exp. 1-12. The reagent ion, $NO_3^-$, is omitted in the label. The normalized abundance was obtained by normalizing all the products to the most abundant one in each experiment, i.e., $C_{18}H_{26}O_{10}$ in Exp. 2-1 and Exp. 2-3, and $C_9H_{14}O_7$ in Exp. 1-12.

This part corresponds to the newly added **Section 3.3** in the revised manuscript (version R2).

We have revised our manuscript as:

Line (791 - 855):

[revised manuscript text omitted]

**R 1.3. Updates on the results of High [OH] experiments**

The estimation on the chemical fate of $RO_2$ in the high [OH] experiments has been updated, using the updated autoxidation reaction rate of BPR (0.078 s$^{-1}$). The radical concentrations estimated by the model have also been updated, due to the updates of kinetics and the input RH values. The obtained chemical fate of $RO_2$ in the high [OH] regime is compared with that in the low [OH] experiments, as shown in **R 1.4**.

**Section 3.1** in the revised manuscript (version R2) has been updated accordingly.

**R 1.4. Comparison of chemical regimes**

We acknowledge that the issue with our high [OH] experiments lies in the scaling of certain pathways, i.e., all the bimolecular reactions, to unrealistic levels, thereby facilitating observation of reaction products that are in fact minor in the ambient.

In our previous reply, the analysis on chemical regimes and the fate of $RO_2$ is intended to clarify that the decrease of DBE of products comes from the addition of OH to the stabilized products, and the hydroxyl termination reaction ($R3$) or hydroperoxyl termination reaction ($R5$) of $RO_2$. Though reaction $R6$ between OH and $RO_2$ can potentially generate ROOOH that is also a possible pathway to decrease DBE, its contribution is quite limited due to its low yield. Also, under our high [OH] experiments, [$HO_2$] and [$RO_2$] both increased with [OH], while the ratios of $HO_2$:OH:$RO_2$ were generally not far from those under ambient conditions. Hence, the ratios for monomeric termination products generated by those monomeric termination reactions, i.e., $R1$ - $R3$ and $R5$ - $R6$, are atmospheric relevant, which are able to alter the DBE of products without influencing their carbon chains.

The radical concentrations were estimated by the PAM_chem_v8 model to illustrate the chemical regimes in the 2$^{nd}$-round experiments (Table R2). The average [$HO_2$], [OH], and [$RO_2$] were $9.7 \times 10^7$, $1.64 \times 10^7$, and $1.69 \times 10^9$ molecule cm$^{-3}$, respectively, in Exp. 2-3, and were $6.7 \times 10^7$, $1.04 \times 10^7$, and $1.34 \times 10^9$ molecule cm$^{-3}$, respectively, in Exp. 2-4, both of which generally differ by no more than a factor of 3 from the summer daytime ambient ones in polluted atmospheres (Tan et al., 2017, 2018, 2019; Whalley et al., 2021; Lu et al., 2012). The average [$HO_2$], [OH], and [$RO_2$], as well as the NO and $NO_2$ concentrations in Exp. 2-7 are generally very close to those in the same environment (Tan et al., 2019).

**Table R2.** Summary of radical concentrations estimated by the PAM_chem_v8 model in the 2$^{nd}$-round experiments.

| | Estimated [OH] with PAM_chem_v8 based on the precursor consumption ($\times 10^7$ molecule cm$^{-3}$) | Estimated [$HO_2$] with PAM_chem_v8 based on the precursor consumption ($\times 10^8$ molecule cm$^{-3}$) | Estimated [$RO_2$] with PAM_chem_v8 based on the precursor consumption ($\times 10^9$ molecule cm$^{-3}$) |
|---|---|---|---|
| 2-1 | 10.3 | 3.73 | 4.05 |
| 2-2 | 4.54 | 2.14 | 2.80 |
| 2-3 | 1.64 | 0.97 | 1.69 |
| 2-4 | 1.04 | 0.67 | 1.34 |
| 2-5 | 5.80 | 13.1 | 1.55 |
| 2-6 | 3.69 | 11.5 | 1.39 |
| 2-7 | 1.69 | 8.38 | 1.07 |

To make the discussion clearer, in this reply we follow the sequential labels of different reactions of

RO$_2$ as below:

[revised manuscript text omitted]

This part corresponds to **Section 3.1** in the revised manuscript (version R2).

We have revised our manuscript (Line 328 - 539) as:

[revised manuscript text omitted]
 S4.** Summary of radical concentrations estimated by the PAM_chem_v8 model in the 2nd-round experiments.

| | Estimated [OH] with PAM_chem_v8 based on the precursor consumption ($\times 10^7$ molecule cm$^{-3}$) | Estimated [HO$_2$] with PAM_chem_v8 based on the precursor consumption ($\times 10^8$ molecule cm$^{-3}$) | Estimated [RO$_2$] with PAM_chem_v8 based on the precursor consumption ($\times 10^9$ molecule cm$^{-3}$) |
|---|---|---|---|
| 2-1 | 10.3 | 3.73 | 4.05 |
| 2-2 | 4.54 | 2.14 | 2.80 |
| 2-3 | 1.64 | 0.97 | 1.69 |

| 2-4 | 1.04 | 0.67 | 1.34 |
| 2-5 | 5.80 | 13.1 | 1.55 |
| 2-6 | 3.69 | 11.5 | 1.39 |
| 2-7 | 1.69 | 8.38 | 1.07 |

**Table S6.** The C9 and C18 products detected by nitrate CIMS in Exp.2-3. The exact mass is mass without adduct of a reagent ion of $NO_3^-$. "--" stands for the signal of compound was below the detection limit.

| Molecular formula | Molar mass (Th) | Contributions to total C9 and C18 products in Exp. 2-3 (%) | Contributions to total C9 and C18 products in Exp. 2-4 (%) | DBE |
|---|---|---|---|---|
| $C_9H_{13}O_4$ | 247.0698 | 0.28 | -- | 3.5 |
| $C_9H_{14}O_4$ | 248.0776 | 0.34 | -- | 3 |
| $C_9H_{12}O_5$ | 262.0569 | 1.03 | 2.03 | 4 |
| $C_9H_{13}O_5$ | 263.0647 | 0.36 | -- | 3.5 |
| $C_9H_{14}O_5$ | 264.0725 | 1.03 | 1.73 | 3 |
| $C_9H_{12}O_6$ | 278.0518 | 1.60 | 2.45 | 4 |
| $C_9H_{13}O_6$ | 279.0596 | 0.63 | 6.54 | 3.5 |
| $C_9H_{14}O_6$ | 280.0674 | 4.64 | 2.11 | 3 |
| $C_9H_{16}O_6$ | 282.0831 | 1.41 | 2.39 | 2 |
| $C_9H_{10}O_7$ | 292.0310 | 1.17 | 2.40 | 5 |
| $C_9H_{12}O_7$ | 294.0467 | 1.56 | 3.38 | 4 |
| $C_9H_{13}O_7$ | 295.0545 | 2.34 | 6.74 | 3.5 |
| $C_9H_{14}O_7$ | 296.0623 | 6.52 | 3.43 | 3 |
| $C_9H_{15}O_7$ | 297.0701 | 4.15 | 1.73 | 2.5 |
| $C_9H_{16}O_7$ | 298.0780 | 5.03 | 5.25 | 2 |
| $C_9H_{12}O_8$ | 310.0416 | 3.20 | 3.31 | 4 |
| $C_9H_{13}O_8$ | 311.0494 | 0.59 | -- | 3.5 |
| $C_9H_{14}O_8$ | 312.0572 | 3.39 | 4.42 | 3 |
| $C_9H_{15}O_8$ | 313.0651 | 2.40 | 2.08 | 2.5 |
| $C_9H_{16}O_8$ | 314.0729 | 7.31 | 8.70 | 2 |
| $C_9H_{12}O_9$ | 326.0365 | 2.22 | 2.54 | 4 |
| $C_9H_{13}O_9$ | 327.0443 | 1.30 | 1.52 | 3.5 |
| $C_9H_{14}O_9$ | 328.0522 | 2.02 | 2.55 | 3 |
| $C_9H_{15}O_9$ | 329.0600 | 0.78 | -- | 2.5 |
| $C_9H_{16}O_9$ | 330.0678 | 1.43 | 2.46 | 2 |
| $C_9H_{12}O_{10}$ | 342.0314 | 1.61 | 1.83 | 4 |
| $C_9H_{13}O_{10}$ | 343.0393 | 0.49 | -- | 3.5 |
| $C_9H_{14}O_{10}$ | 344.0471 | 1.07 | -- | 3 |
| $C_9H_{15}O_{10}$ | 345.0549 | 0.24 | -- | 2.5 |
| $C_9H_{16}O_{10}$ | 346.0627 | 0.37 | -- | 2 |
| $C_9H_{12}O_{11}$ | 358.0263 | 2.24 | 3.13 | 4 |
| $C_9H_{13}O_{11}$ | 359.0342 | 0.60 | -- | 3.5 |
| $C_9H_{14}O_{11}$ | 360.0420 | 0.46 | -- | 3 |
| $C_9H_{15}O_{11}$ | 361.0498 | 0.19 | -- | 2.5 |
| $C_{18}H_{24}O_8$ | 430.1355 | 0.56 | -- | 7 |
| $C_{18}H_{26}O_8$ | 432.1511 | 4.83 | 5.86 | 6 |
| $C_{18}H_{24}O_9$ | 446.1304 | 0.59 | -- | 7 |
| $C_{18}H_{26}O_9$ | 448.1461 | 1.35 | 2.53 | 6 |
| $C_{18}H_{24}O_{10}$ | 462.1253 | 0.45 | -- | 7 |
| $C_{18}H_{25}O_{10}$ | 463.1332 | 0.36 | -- | 6.5 |
| $C_{18}H_{26}O_{10}$ | 464.1410 | 9.58 | 9.36 | 6 |
| $C_{18}H_{28}O_{10}$ | 466.1566 | 2.97 | 2.42 | 5 |

| | | | | |
|---|---|---|---|---|
| $C_{18}H_{24}O_{11}$ | 478.1202 | 0.51 | -- | 7 |
| $C_{18}H_{26}O_{11}$ | 480.1359 | 2.28 | 2.15 | 6 |
| $C_{18}H_{27}O_{11}$ | 481.1437 | 1.09 | 1.42 | 5.5 |
| $C_{18}H_{28}O_{11}$ | 482.1515 | 2.44 | 1.70 | 5 |
| $C_{18}H_{26}O_{12}$ | 496.1308 | 1.61 | 1.84 | 6 |
| $C_{18}H_{28}O_{12}$ | 498.1465 | 1.13 | -- | 5 |
| $C_{18}H_{30}O_{12}$ | 500.1621 | 0.35 | -- | 4 |
| $C_{18}H_{24}O_{13}$ | 510.1101 | 0.33 | -- | 7 |
| $C_{18}H_{26}O_{13}$ | 512.1257 | 0.83 | -- | 6 |
| $C_{18}H_{27}O_{13}$ | 513.1335 | 0.51 | -- | 5.5 |
| $C_{18}H_{28}O_{13}$ | 514.1414 | 0.69 | -- | 5 |
| $C_{18}H_{30}O_{13}$ | 516.1570 | 0.33 | -- | 4 |
| $C_{18}H_{26}O_{14}$ | 528.1206 | 0.67 | -- | 6 |
| $C_{18}H_{27}O_{14}$ | 529.1284 | 0.17 | -- | 5.5 |
| $C_{18}H_{28}O_{14}$ | 530.1363 | 0.37 | -- | 5 |
| $C_{18}H_{30}O_{14}$ | 532.1519 | 0.30 | -- | 4 |
| $C_{18}H_{26}O_{15}$ | 544.1155 | 0.31 | -- | 6 |
| $C_{18}H_{27}O_{15}$ | 545.1234 | 0.24 | -- | 5.5 |
| $C_{18}H_{28}O_{15}$ | 546.1312 | 0.34 | -- | 5 |
| $C_{18}H_{30}O_{15}$ | 548.1469 | 0.21 | -- | 4 |
| $C_{18}H_{26}O_{16}$ | 560.1105 | 0.30 | -- | 6 |
| $C_{18}H_{28}O_{16}$ | 562.1261 | 0.30 | -- | 5 |

**Table S7.** The C9 products detected by nitrate CIMS in Exp.2-7.

| Molecular formula | Molar mass (Th) | Contributions to total C9 and C18 products signals (%) | DBE |
|---|---|---|---|
| $C_9H_{12}O_6$ | 216.0634 | 1.25 | 4 |
| $C_9H_{13}O_6$ | 217.0712 | 1.98 | 3.5 |
| $C_9H_{14}O_6$ | 218.0790 | 1.95 | 3 |
| $C_9H_{13}NO_6$ | 231.0743 | 3.97 | 3 |
| $C_9H_{12}O_7$ | 232.0583 | 1.59 | 4 |
| $C_9H_{13}O_7$ | 233.0661 | 2.92 | 3.5 |
| $C_9H_{15}O_7$ | 235.0818 | 0.97 | 2.5 |
| $C_9H_{13}NO_7$ | 247.0692 | 9.32 | 3 |
| $C_9H_{12}O_8$ | 248.0532 | 3.11 | 4 |
| $C_9H_{13}O_8$ | 249.0610 | 0.94 | 3.5 |
| $C_9H_{14}O_8$ | 250.0689 | 2.15 | 3 |
| $C_9H_{13}NO_8$ | 263.0641 | 21.25 | 3 |
| $C_9H_{13}O_9$ | 265.0559 | 1.57 | 3.5 |
| $C_9H_{15}NO_8$ | 265.0798 | 2.94 | 2 |
| $C_9H_{13}NO_9$ | 279.0590 | 5.45 | 3 |
| $C_9H_{12}O_{10}$ | 280.0430 | 1.91 | 4 |
| $C_9H_{14}NO_9$ | 280.0669 | 3.34 | 2.5 |
| $C_9H_{13}O_{10}$ | 281.0509 | 1.52 | 3.5 |
| $C_9H_{15}NO_9$ | 281.0747 | 2.51 | 2 |
| $C_9H_{14}N_2O_9$ | 294.0699 | 1.76 | 2 |
| $C_9H_{13}NO_{10}$ | 295.0539 | 3.52 | 3 |
| $C_9H_{14}N_2O_{10}$ | 310.0649 | 18.15 | 2 |
| $C_9H_{13}NO_{11}$ | 311.0489 | 3.11 | 3 |
| $C_9H_{12}N_2O_{11}$ | 324.0441 | 1.23 | 3 |
| $C_9H_{14}N_2O_{11}$ | 326.0598 | 1.60 | 2 |

*Also, I would advise the authors to be careful what they consider as an "ORF setup". It seems they have considered OFR to be everything where oxidation is studied, notwithstanding that several other setups have been previously meticulously characterized and assigned as well controlled flow reactors and environmental chambers that aim to investigate the processes under ambient relevant conditions. Especially the setups mentioned in the response letter, the Jülich JPAC chambers and the very carefully controlled free-jet flow reactor of Torsten Berndt in TROPOS, Leipzig. In JPAC (e.g., Garmash et al., 2020) the reaction time is much larger (ca. 50 minutes), surface-to-volume smaller and the oxidation mixture more dilute, leading to more ambient relevant conditions. In TROPOS (e.g., Wang et al., 2017) the reactions have been studied in short reaction times (at 7.9 s) in a practically wall-less setup with careful control over the reagent concentrations. Equating these setups and works with the PAM and the current work appears wrong.*

**R2.** We believe that in this comment the reviewer concerns about the usage/definition of PAM as an oxidation flow reactor (OFR). OFR is defined by the community as "flow reactors using oxidants with substantially (often orders of magnitude) higher concentrations than in the atmosphere to accelerate the oxidation chemistry in them" (Peng and Jimenez, 2020). The key aspects for an OFR are the flowing reaction manner of oxidation reactions and the higher concentrations of oxidants to accelerate oxidation reactions inside the reactor, which a PAM fulfills. In fact, the PAM manufactured by Aerodyne Research Inc. or flow reactors constructed with the same settings has been considered as an OFR by various groups for a long period of time (Avery et al., 2023; Lambe et al., 2011, 2015, 2017, 2019; Peng et al., 2015, 2016, 2018; Li et al., 2015; Kang et al., 2007; Watne et al., 2018). Meanwhile, our new experiments have shown that PAM can be used to study reactions under chemical regimes with atmospheric relevant [OH], i.e., $10^7$ cm$^{-3}$, but the total OH exposure in the flow tube under this condition is limited because of the pre-determined residence time in the PAM.

Nevertheless, if one is interested in reactions with extended OH exposures that is not straightforward to the current chamber/flow tube techniques, a possible solution is to increase [OH] to compensate for the residence time. As the reviewer has pointed out, one should be very careful with the obtained results because the unequally scaled radical concentrations and/or condensation of low-volatile products can both alter the product distribution even the production or not of a particular product. This is also why the concerns on the atmospheric relevance of chemistry in the OFR has persisted (Peng and Jimenez, 2020).

Although whether PAM can be considered as an OFR or not is absolutely relevant, the more important issue is whether results from this study is relevant to the ambient or used to help understand multigeneration oxidation mechanisms of aromatics and their stabilized products, which, we believe, is elaborated in **R1** of this reply and Session 3.1 of the revised manuscript (version R2).

*The chosen autoxidation rate of 7 s$^{-1}$ is very high, and not supported by the referenced study. The Authors state they based this on the expected autoxidation rate of ethylbenzene, yet this higher rate in the referenced work is due to a secondary hydrogen on the ethyl substituent, which is absent from methyl groups in the studied TMB molecule. Instead, the autoxidation rate from a methyl group was found 270 times slower (7 s$^{-1}$ / 0.026 s$^{-1}$). If one takes into account the symmetry with three methyl groups, then on could expect the 1,3,5-TMB rate as around 3 times higher, yet still likely less than 0.1 s$^{-1}$. Thus, this choice necessarily leads to severely overestimated fraction of autoxidized products. In fact, the Authors even state that "We arbitrarily set the autoxidation reaction rate…", and this seems to be critical for the analysis, leading to conclusion that 81% to 89.7% of the reaction products are due to autoxidation.*

**R3.** We follow quantum calculation results on the autoxidation reaction of a methyl group adjacent to the $RO_2$ functionality group (Wang et al. 2017), and time the suggested rate (0.026 s$^{-1}$) by 3 due to the symmetry with three methyl groups in our parent compound. The obtained autoxidation reaction rate is 0.078 s$^{-1}$, which has been updated in our PAM_chem_v8 model. The updated autoxidation reaction rate leads to changes in the calculated concentrations of BPR and $C_9H_{13}O_7\cdot$, but almost does not impact the sum of $[RO_2]$. Given an uncertainty range of 1 order magnitude for the autoxidation rate (Wang et al. 2017) that covers 0.0078 – 0.78 s$^{-1}$, the total $[RO_2]$ estimated by the model would vary by less than 2% even under the highest [OH] level.

The selection of this autoxidation rate is now updated in Section S1.

In addition, due to the update of the autoxidation rate, radical concentrations in the high [OH] experiments and radical profiles in Fig. S1 are updated.

Discussions on the 1$^{st}$-round experiments have also been updated, which has been included in **R1.2**.

We have revised our manuscript (Line 329 - 339), which reads,

Concentration profiles of OH, $RO_2$, and $HO_2$ as a function of OH exposures in our high [OH] experiments without $NO_x$, i.e., the 1$^{st}$-round experiments, are illustrated in Fig. S1a. According to the modified PAM_chem_v8, when [OH] increased from $9.32\times10^7$ to $1.03\times10^9$ molecule cm$^{-3}$, $[HO_2]$ increased from $7.25\times10^8$ to $2.79\times10^9$ molecule cm$^{-3}$, whereas $[RO_2]$ concentrations increased from $5.17\times10^9$ to $9.5\times10^9$ molecule cm$^{-3}$. The radical concentrations in high [OH] experiments with $NO_x$ (Fig. S1b) varied in a similar range, with $[RO_2]$ ranging from $4.38\times10^9$ to $9.13\times10^9$ molecule cm$^{-3}$, $HO_2$ ranging from $4.47\times10^9$ to $6.47\times10^9$ molecule cm$^{-3}$, and OH ranging from $3.86\times10^8$ to $7.82\times10^8$ molecule cm$^{-3}$, respectively. The ratios between $HO_2/OH$ and $RO_2/OH$ in the 1$^{st}$-round experiments were generally in the same order of magnitude with the ambient atmosphere (Whalley et al., 2021).

We have revised Supplementary Text S1, which reads,

"We follow quantum calculation results on the autoxidation reaction of a methyl group adjacent to the $RO_2$ functionality group (Wang et al. 2017), and time the suggested rate (0.026 s$^{-1}$) by 3 due to the symmetry with three methyl groups in our parent compound. The obtained autoxidation reaction rate is 0.078 s$^{-1}$. "

We have revised Figure S1, which is shown below

(a)

[Figure]

(b)

[Figure]

**Figure S1.** (a) Concentration profiles of OH, $HO_2$, BPR, and total $RO_2$ in the 1st-round experiments in absent of $NO_x$ in the PAM OFR as a function of OH exposures. The average total concentrations of $RO_2$ and BPR were scaled with a factor of 0.1 for a better visualization. (b) Concentration profiles of OH, $HO_2$, BPR, and total $RO_2$ in the 1st-round experiments in presence of $NO_x$ in the PAM OFR as a function of OH exposures. The average total concentrations of $RO_2$ were scaled with a factor of 0.9 for a better visualization.

*Similarly, as the pool of simulated RO2 consists only of the two prominent RO2 radicals (i.e., BPR and C9H13O7) which both have been assigned with a very high accretion rate, and then the whole pool of ROOR is counted based on these same two radicals. That would also seem to indicate an overestimated importance for the accretion reaction, or do I read it wrong?*

**R4.** Indeed, more $RO_2$ radicals than the two (BPR and $C_9H_{13}O_7\cdot$) currently explicitly mentioned and modeled exist in the PAM OFR tube, which come from the subsequent fragmentation of stabilized products as suggested by MCM and the multi-generation OH oxidation of stabilized products.

Accretion reaction rate constants for smaller $RO_2$ generated by the fragmentation of stabilized products are typically 1 - 2 orders of magnitude lower than those of BPR and $C_9H_{13}O_7\cdot$, as suggested in a previous study (Berndt et al., 2018). Accretion reaction rate constants of multi-generation $RO_2$ are currently unknown, but should be at least no lower than that of the smaller $RO_2$. However, reactions of these two categories of $RO_2$ with BPR and $C_9H_{13}O_7\cdot$, including $RO_2$ termination reactions (R1 – R3) and accretion reactions (R4), were not included in the model. Therefore, it is likely that the accretion channel of BPR and $C_9H_{13}O_7\cdot$ was not overestimated but underestimated.

PAM_chem_v8 model simulations here are conducted to characterize [OH], [HO_2], and [RO_2] inside the PAM OFR, not to really capture the detailed reactions, including the fragmentation, re-initiation of OH oxidation, and other chemical reactions. This model was developed to specifically simulate the formation and evolution of key radicals and oxidants, and quantify the radical budget for the OFR (Li et al., 2015). On the other hand, detailed reaction mechanisms e.g., the MCM-based 0-D box model, are still unlikely to estimate the exact significance of accretion reactions in the PAM OFR, as they do not cover $RO_2$ generated in the re-initiation of OH oxidation, whose interactions with BPR and $C_9H_{13}O_7\cdot$ are obviously unignorable based on our observations in both rounds of experiments.

The accretion reactions in the PAM have already been intensified a lot due to the much more abundant BPR and $C_9H_{13}O_7\cdot$ than those in the ambient. Therefore, the conclusion drawn from the high [OH] experiments is constrained to the proximity of monomeric termination products to those under ambient conditions as stated in **R1**. We focus on the monomeric reaction of $RO_2$ in the PAM OFR, and agree that the accretion reaction ($R4$) works as a magnified pathway of $RO_2$ in the laboratory. The HOM dimer data obtained from the high [OH] experiments serve as a symbolic illustration of the enhanced abundance of multi-generation OH oxidation in the laboratory, correlating with the elevated levels of radicals in the reactor. The uncertainty in the proportions of accretion reactions would not impact the existence of multi-generation OH oxidation.

*Furthermore, I still find the decreasing ROOR signal problematic, as under such short timescales it would seem to indicate either particle formation, or secondary reactions of the formed ROOR, neither which would be expected under atmospheric conditions at such short times. It is stated that no particle formation was observed with long SMPS, but I suppose the SMPS cut-off limit is so high that it would not see the fresh particles in any case. Or what was the smallest measurable particle size with the SMPS system?*

**R5.** The secondary oxidation of HOM dimers absolutely happened in the PAM OFR in both the 1st-round and the 2nd-round experiments, and their physical loss due to the generation of particles might have happened, too.

The long SMPS consisted of a long-DMA (TSI model 3081) and a CPC (TSI model 3787), covering a particle number size distribution from 13.6 nm to 736.5 nm. As suggested by the reviewer, the long SMPS cannot measure freshly formed particles. Thus, though not detected in this study, we cannot absolutely deny

the possibility that particles might have been generated with the increment of OH exposures, resulting in a larger physical loss of HOMs. We would like to add this information into the revised manuscript.

If the survival time permits, the secondary OH oxidation of stabilized HOM dimers should be feasible and the reaction rate constant should be at least comparable to those stabilized monomers. According to the general mechanisms of OH addition reactions to a C=C bond, the resulting $RO_2$ radical possesses one more hydrogen atom and three more oxygen atoms than the reactants. Then the most abundant dimer radical, $C_{18}H_{27}O_{11}\cdot$ (Figure R4), and the second most one, $C_{18}H_{27}O_{13}\cdot$, in the atmospheric relevant [OH] experiment of Exp. 2-3 exactly correspond to two previous reported HOM dimer products $C_{18}H_{26}O_8$ and $C_{18}H_{26}O_{10}$ (Berndt et al., 2018), respectively, which confirms the secondary OH oxidation of stabilized HOM dimers, though in Exp. 2-4 only $C_{18}H_{27}O_{11}\cdot$ was detected.

In the ambient, pre-existing particles can contribute to the physical loss of oxidation products. Atmospheric [OH] can also re-initiate secondary OH oxidation reactions. Nevertheless, the detailed timescale for the decrease of dimers should be different from what we have observed in the laboratory due to the complex environment in the ambient.

[Figure]

**Figure R4.** The fitting of averaged $C_{18}H_{27}O_{11}(NO_3)^-$ measured by nitrate CIMS in the Experiment 2-3 and 2-4.

We have revised our manuscript (Line 755 - 761), which reads,

"With the decrease of particulate pollution and thus condensation sinks in the polluted areas, the physical loss of HOMs might be lower and the chemical process can be more important. This series of experiments are not meant to specifically find out the detailed OH exposures when the maximum concentrations of HOM dimers will occur, but try to indicate how HOM dimers evolve with the increase of OH exposures. This work can be regarded as an indicator for the potential chemical fates of HOM dimers in the atmosphere if their survival time permitted."

We have revised our supplementary Text S1, which reads,

"In our experiments, measurement results by a long-SMPS show that the aerosol particles presented in the PAM OFR were few. The long SMPS consisted of a long-DMA (TSI model 3081) and a CPC (TSI model 3787), covering a particle number size distribution from 13.6 nm to 736.5 nm. Thus, though not detected in this study, we cannot absolutely deny the possibility that particles might have been generated with the increment of OH exposures, resulting in a larger physical loss of HOMs. This part of physical loss might be underestimated."

*The Authors seem to several times make the claim that the OH concentration in the work of Garmash et al., was higher, which is not the case. Even the highest [OH] = 4.5 x 10^8 cm^-3 was still considerably smaller than reported in the current study (i.e., 1.5 10^9 cm^-3). Moreover, in the work of Garmash et al., it was recognized that with such an unrealistically high OH, you can't really derive sound mechanistic conclusions, and thus no mechanisms were proposed. However, the fact that the rate coefficients increase together with the number of OH substituents was recognized during the study, which is why the potential for "multi-generation OH oxidation" in product formation was acknowledged as a complication for data analysis, and no other mechanistic description was provided due to the non-conventional reaction conditions.*

*This comparison seems in any case problematic, as the Authors talk about 0.7-6.9 hours of atmospheric aging at 2x10^6 cm^-3 [OH], yet the residence time in the setup is only 53 seconds. The exact [OH] used seemed to be unknown, yet with the response letter the Authors quote an OH concentration of 1.5x10^9 cm^-3. Now, in Garmash et al., the residence time is roughly 60 times longer and the maximum used [OH] is 4.5x10^8 cm^-3, naturally leading to longer aging timescale. What the Authors seem to miss here is that what is more important is the absolute concentrations of the reagents, as they push the chemistry to an unrealistic regime, and not the equivalent dose. And repeating from above, Garmash et al., did not derive mechanistic conclusion from the higher [OH] experiments, other than the sequential oxidation behavior.*

**R6.** Sorry for the unclarity in the previous comparison with the Garmash et al. study (2020). We agree with the reviewer that one should be very careful with the obtained results under laboratory high [OH] conditions because the unequally scaled radical concentrations and/or condensation of low-volatile products can both alter the product distribution even the production or not of a particular product.

Nevertheless, OFR methods provide an opportunity to investigate oxidation reactions with short residence times and reduced wall effects. Despite these advantages of OFR, concerns over the atmospheric relevance of the radical chemistry (and hence gas and aerosol products) in OFR have persisted.

Based on our previous analysis on the radical concentrations of high [OH] experiments, the product distributions of $RO_2$ monomeric termination reactions in the PAM are atmospheric relevant. We were trying to show that the subsequent oxidation reactions do exist for stabilized oxidation products of TMB, which could be deduced based on the DBE of detected compounds. Once a re-initiated OH oxidation of these first-generation products happened, we can unambiguously detect the secondary reaction based on the lower DBE of the secondary product. Besides, the comparison between the high [OH] experiments and the newly conducted low [OH] experiments confirms that these low DBE products are related to the increasing [OH].

Certainly, based on these results we cannot make conclusions that re-initiated OH oxidation of a certain compound in the ambient can be as important as in the lab, because of the different survival time, as discussed in **R1**.

We acknowledge that the mechanistic conclusion obtained in our high [OH] experiments is not robust enough by themselves, as the ratio of radicals will impact the distribution of products. Fortunately, in the laboratory low [OH] experiments, we still detected the low DBE products and critical radicals, i.e., $C_9H_{15}O_8\cdot$

and $C_9H_{15}O_9\cdot$, which are discussed in the proposed schemes (Scheme 2 and Scheme 3). This confirms that these reactions can happen and are still important in the atmospheric relevant [OH] conditions. Please refer to **R1.2**.

*So, my problem with the current documentation is that PAM, and the OFR approach, is really a methodology for reaching emission regulations, but it's ill-suited for detailing complex chemistry under atmospheric relevant concentration regimes. It would make an interesting comparison to see how the products change between atmospheric $10^6$ to $10^7$ $cm^{-3}$ to OFR $10^{-9}$ $cm^{-3}$, but unfortunately such an analysis was not provided. (To put it another way, if you would have written the story like "if HOMs would survive a day – what would happen", and not trying to insist this is strictly atmospherically relevant product distribution, then this would have made much more sense).*

**R7.** We acknowledge that the issue with high [OH] experiments lies in the scaling of certain reaction pathways to unrealistic levels, thereby facilitating the observation of reaction pathways that are minor in the ambient. Because of the limited survival time of reactants and competition of various reaction channels, laboratory products won't be identical to ambient ones after the same OH exposures. Even the same product can be generated in both the laboratory and the ambient, the relative significance of this products is not likely to be the same.

We agree that experiments under atmospheric radical concentrations are meaningful and helpful to probe the existence of certain oxidation reactions in the real atmosphere. Therefore, these atmospheric relevant [OH] experiments have been conducted, whose results and discussions are provided in **R1**.

We now discuss the chemical regimes for the investigated low [OH] experiments and show that the physical loss was biased to be lower in the PAM. Please refer to **R1.4**.

*Also, the Authors probably slightly misunderstood my comment about the potential influence of light: It would be expected to be a problem for the aromatic oxidation products undergoing photo-oxidation, not that the precursor TMB would be photolyzed. Also, I don't understand the comment that says: "Meanwhile, photolysis of HOMs can lead to decomposition, decreasing detected signals of HOMs, but unlikely to generate new HOMs." If the photolysis yields radical intermediates, as it does, then these new intermediates can continue oxidation to HOM as well. It's quite likely that this was occurring in the experiments.*

**R8.** We separated the influences of photolysis into two paragraphs, one on the precursors (**Line 177 - 184** in the previous revised manuscript with marks) and the other one on the oxidation products (**Line 185 - 189** in the previous revised manuscript with marks).

Since the photo intensity of 254 nm lamps can be estimated with the PAM_chem_v8 model, we can calculate the OH reaction rate and the photolysis rate of stabilized HOMs in the OFR, which together show that the proportion of photolysis was quite low. The calculations have been provided in the last reply.

On the other hand, according to the MCM (Saunders et al., 2003), the photolysis of oxygenated organic compounds, e.g. benzaldehyde, other complex carbonyl compounds, and hydroperoxides, will decompose the C-C bond adjacent to the oxygenated functionality. In the last version, we were trying to state that the photolysis, if really occurred, would mainly result in the decomposition of HOMs and generation of new compounds with fewer carbon atoms. In other words, photolysis is unlikely to diversify or augment the detected HOMs with 9 or 18 carbon atoms, which are the main subject of discussion of this paper. We are

sorry we did not clarity our discussion as C9 or C18 HOMs in **Line 185 - 189** in the previous revised manuscript with marks.

We have revised our manuscript (Line 193 - 207), which reads,

"Non-tropospheric VOC and OVOC photolysis is a typical issue that should be taken into account when evaluating the settings of OFR laboratory experiments, especially in the high UV light dose settings in the 1$^{st}$-round experiments. Photolysis of the precursor and HOMs were evaluated, showing that photolysis was not a contributor to our observation. The photolysis rate of 1,3,5-TMB can be estimated based on the absorption cross-sections of 1,3,5-TMB at 254 nm (Keller-Rudek et al., 2013) and UV photon fluxes estimated by a chemistry model discussed in the following sections. The ratio of photolysis-to-OH reaction in our 1$^{st}$-round experiments was merely 0.010 – 0.033. Hence, photolysis of 1,3,5-TMB was insignificant in the OFR. For stabilized products such as C9 and C18 HOMs, the cross sections of organic molecules are usually $\sim 3.9 \times 10^{-18}$ - $3.9 \times 10^{-17}$ cm$^2$ (Peng et al., 2016), while the reaction rate between OH and the stabilized first-generation products are estimated to be around $1.28 \times 10^{-10}$ molecule$^{-1}$ cm$^3$ s$^{-1}$, as suggested by MCM (Jenkin et al., 2003). Hence, the ratio of photolysis rates of C9 and C18 HOMs to their secondary OH oxidation rates is estimated to be merely around 0.020 – 0.056 in the 1$^{st}$-round experiments. In the 2$^{nd}$-round, the influences of photolysis should be even lower due to the much lower light intensity."

*Thanks for educating me that in PAM a ¼" Teflon tube is used. This is not adequate for measuring very condensable products. This is a problem I had overlooked in the common PAM methodology.*

**R9.** As stated in the **R1**, we have modified the sampling system of PAM OFR. A 30 cm-long Teflon tube with a 1/2 in. OD was utilized for the sampling of the new nitrate CI-TOF. Several new low volatile species have been detected in the 2$^{nd}$-round experiments, i.e., $C_9H_xO_{11}$ ($x$ = 12 - 15).

***Reviewer #2****: Wang et al. present laboratory results where they oxidized trimethylbenzene to investigate the role of autoxidation. The authors have done an exemplary job responding to both reviewers' comments, providing much needed calculations to indicate how their observations are similar to atmospheric conditions, softening the language when necessary, and improving the figures and story overall. The paper is ready for consideration for publication in ACP.*

We thank this reviewer for his/her positive view on our manuscript.

---

## Author Response (AR3)

**RE: A point-to-point response to reviewers' comments**

"Secondary OH reactions of aromatics-derived oxygenated organic molecules lead to plentiful highly oxygenated organic molecules within an intraday OH exposure" by Yuwei Wang, Chuang Li, Ying Zhang, Yueyang Li, Gan Yang, Xueyan Yang, Yizhen Wu, Lei Yao, Hefeng Zhang, Lin Wang (egusphere-2023-1702)

Dear Dr. Liggio,

We are very grateful for the comments from the reviewer and your suggestions to improve our work. To address the concerns raised by the reviewer and you, a point-to-point response to the comments, which are repeated in *italic*, is given below.

We are looking forward to your decision at your earliest convenience.

Best regards,

Lin Wang
Fudan University
lin_wang@fudan.edu.cn

*The manuscript has been improved to make it publishable, after some additional corrections as pointed out by the most recent review. In particular,*

*(1) A thorough language editing should be performed before the work is accepted for publication.*
**R1.** We are sorry for the language issue. We have carefully edited the texts to enhance the clarity and readability of this manuscript.

We have revised our manuscript (Line 73 – 74) as:

[revised manuscript text omitted]

(Line 643) as:
"as recommended by MCM,"

(Line 810 – 813) as:
"On the other hand, the loss pathways of HOM dimers were not exactly the same as the ambient as discussed in the last two paragraphs of Section 3.1. This …"

(Line 902 – 903) as:
"the larger proportion"

We have also revised the figure and table captions in the supplement.

We have revised the table caption of Table S2 as:

**"Table S2.** Reactions included in the modified PAM_chem_v8 model under the settings with only 254 nm UV lights on. For experiments in the absence of $NO_x$, the input value of $N_2O$ is 0 and all the $NO_x$-related reactions actually proceed with a zero rate. $RO_2$ is the sum of BPR and $C_9H_{13}O_7\cdot$ for simplification."

the table caption of Table S6 as:

**"Table S6.** The C9 and C18 products detected by nitrate CI-ToF in Exp. 2-3. The exact mass is the mass without an adduct of a reagent ion of $NO_3^-$. "--" stands for the signal of a compound was below the detection limit."

the figure caption of Figure S1 as:

**"Figure S1.** (a) Concentration profiles of OH, $HO_2$, BPR, and total $RO_2$ in the 1$^{st}$-round experiments in the absent of $NO_x$ in the PAM OFR as a function of OH exposures. The average total concentrations of $RO_2$ and BPR were scaled with a factor of 0.1 for a better visualization. (b) Concentration profiles of OH, $HO_2$, BPR, and total $RO_2$ in the 1$^{st}$-round experiments in the presence of $NO_x$ in the PAM OFR as a function of OH exposures. The average total concentrations of $RO_2$ were scaled with a factor of 0.9 for a better visualization."

and the figure caption of Figure S2 as:

**"Figure S2.** MS comparison of the (a) monomer products between Exp. 1-3 and Exp. 1-19, (b) dimer products between Exp. 1-3 and Exp. 1-19, (c) monomer product MS between Exp. 1-12 and Exp. 1-22, and (d) dimer products between Exp. 1-12 and Exp. 1-22. The signals of HOMs were raw ones in the nitrate CIMS."

We have also corrected the incorrect hyphens in our manuscript with En Dashes.

We have revised our manuscript (Line 203) as:

"$\sim 3.9\times10^{-18} - 3.9\times10^{-17}$"

(Line 438) as:

"and $(4 - 28)\times10^8$, $(0.8 - 2.4)\times10^7$"

(Line 457) as:

"(R1 – R3 and R5 – R9)"

(Line 482) as:

"of $R1 - R3$ and $R5 - R7$"

(Line 496) as:

"($R1 - R3$, $R5 - R6$, and $R8 - R9$)"

(Line 514 – 516) as:

"the proportions of $R8 - R9$ , i.e., the NO channel in the urban atmosphere were attributed to termination reactions of $R1 - R6$,"

(Line 853) as:

"($x = 12 - 15$)"

(Line 863) as:

"($m = 7 - 11$)"

(Line 877) as:

"($R8 - R9$)"

*(2) Somewhat strange statements persist in the text, like "Formation of HOMs is typically triggered by oxidation of VOCs in the gas phase." – how else they could be formed?*

**R2.** We are sorry for the unsuitable statements in the manuscript. We have revised our manuscript (Line 59) as:

"Formation of HOMs is triggered by oxidation of VOCs in the gas phase."

*(3) PAM name comes from "potential aerosol mass". Please correct.*

**R3.**

We have revised our manuscript (Line 29 – 30) as:

"with a potential aerosol mass oxidation flow reactor (PAM OFR)"

and (Line 148 – 149) as:

"investigated in a potential aerosol mass oxidation flow reactor (PAM OFR)"

*(4) Add important experimental details to the main paper*

**R4.** We have integrated Section S1 in the supplement into Section 2 in the main text of the manuscript.

We have revised our manuscript (Line 288 – 294) as:

"In this work, autoxidation and accretion of 1,3,5-TMB-derived BPR, as well as subsequent reactions of the autoxidation product of BPR, i.e., $C_9H_{13}O_7\cdot$, are newly implemented or modified in this model (Reaction No. 46 – 62 in Table S2). These two radicals were the most significant $RO_2$ in the system and represented the whole $RO_2$ pool in the PAM chemistry model simulation. The pathways of peroxy radicals and their kinetics are discussed below. $NO_x$-related reactions are also included in the model. When experiments without $NO_x$ are simulated, these $NO_x$-related reactions do not contribute to the simulation results.
"

(Line 304 – 388) as:

"The reaction rate constants for $RO_2$ in $R1 - R5$ are obtained by MCM or previous investigations (e.g., Jenkin et al., 2003; Berndt et al., 2018; Peng and Jimenez, 2020). We treat $R1 - R3$ as a total

reaction with a reaction rate constant of $8.8 \times 10^{-13}$ molecule$^{-1}$ cm$^3$ s$^{-1}$, and branching ratios of $R1 - R3$ of 0.6, 0.2, and 0.2, respectively, as suggested by MCM (Jenkin et al., 2003). The reaction rate constants of BPR and C$_9$H$_{13}$O$_7$· for $R4$ are $1.7 \times 10^{-10}$ and $2.6 \times 10^{-10}$ molecule$^{-1}$ cm$^3$ s$^{-1}$, respectively (Berndt et al., 2018). The reaction rate constants for $R5$ is $1.5 \times 10^{-11}$ 
[revised manuscript text omitted]

*(5) Please use consistent concentration units throughout the text.*

**R5.** We have used the concentration unit of molecular cm$^{-3}$ throughout the manuscript and supplement.

We have revised our manuscript (Line 166 – 168) as:

"leading to $7.08\times10^{11}$ – $1.54\times10^{12}$ molecule cm$^{-3}$ of 1,3,5-TMB in the 1$^{st}$-round experiments, and $7.55\times10^{11}$ or $8.45\times10^{11}$ molecule cm$^{-3}$ of 1,3,5-TMB in the 2$^{nd}$-round experiments …"

(Line 179 – 181) as:

"resulting in an initial ozone concentration of around $1.05\times10^{13}$ – $2.16\times10^{13}$ molecule cm$^{-3}$ in the OFR in the 1$^{st}$-round experiments and $3.01\times10^{12}$ – $3.72\times10^{12}$ molecule cm$^{-3}$ in the 2$^{nd}$-round experiments, respectively."

(Line 213 – 216) as:

"generally resulting in two NO$_x$ levels, $4.41\times10^{10}$ molecule cm$^{-3}$ NO + $1.72\times10^{12}$ molecule cm$^{-3}$ NO$_2$ (Exp. 1-41 – 1-54) and $1.18\times10^{11}$ molecule cm$^{-3}$ NO + $2.94\times10^{12}$ molecule cm$^{-3}$ NO$_2$ (Exp. 1-55 – 1-68) at the exit of the OFR."

(Line 218) as:

"were adjusted in a large range from $4.09\times10^{11}$ to $2.06\times10^{12}$ molecule cm$^{-3}$"

(Line 223 – 225) as:

"which also resulted in fluctuations in the NO concentrations ([NO]) from $3.19\times10^{10}$ to $1.74\times10^{11}$ molecule cm$^{-3}$ and the NO$_2$ concentrations ([NO$_2$]) from $2.70\times10^{11}$ to $9.31\times10^{11}$ molecule cm$^{-3}$."

(Line 403 – 406) as:

"53 s, 0.63%, $1.23\times10^{13}$ molecule cm$^{-3}$, and $1.23\times10^{12}$ molecule cm$^{-3}$, respectively, as measured directly. For the 2$^{nd}$-round experiments, the input parameters of O$_3$ concentration and the initial 1,3,5-TMB concentration were updated as $3.68\times10^{12}$ molecule cm$^{-3}$ and $7.55\times10^{11}$ molecule cm$^{-3}$, respectively."

(Line 521 – 524) as:

"as an example, [OH] = ~$6.77\times10^{8}$ molecule cm$^{-3}$, NO = ~$4.73\times10^{10}$ molecule cm$^{-3}$. NO$_2$ = ~$1.67\times10^{12}$ molecule cm$^{-3}$), low [OH] experiments (Exp. 2-7 as an example, [OH] = ~$1.69\times10^{7}$ molecule cm$^{-3}$, NO = ~$3.19\times10^{10}$ molecule cm$^{-3}$. NO$_2$ = ~$2.70\times10^{11}$ molecule cm$^{-3}$),"

and (Line 726 – 727) as:

"The averaged mass spectrometry of nitrate CIMS in the $4.41\times10^{10}$ molecule cm$^{-3}$ NO experiment and $1.18\times10^{11}$ molecule cm$^{-3}$ NO experiment …"

We have revised Table S1 in the supplement as:
"

**Table S1.** Summary of experimental conditions.

| No. | Initial concentration of 1,3,5-TMB | O$_3$ concentration ($\times10^{12}$ | NO concentration ($\times10^{10}$ molecule cm$^{-3}$) | NO$_2$ concentration ($\times10^{11}$ molecule cm$^{-3}$) | Estimated OH exposure based on the precursor |
|---|---|---|---|---|---|

| | ($\times 10^{11}$ molecule cm$^{-3}$) | molecule cm$^{-3}$)$^*$ | | | consumption ($\times 10^9$ molecule cm$^{-3}$ s) |
|---|---|---|---|---|---|
| 1-1 | 13.4 | 11.0 | 0 | 0 | 18.5 |
| 1-2 | 14.4 | 11.1 | 0 | 0 | 11.2 |
| 1-3 | 15.4 | 11.4 | 0 | 0 | 5.2 |
| 1-4 | 9.4 | 10.6 | 0 | 0 | 44.0 |
| 1-5 | 9.8 | 10.8 | 0 | 0 | 40.9 |
| 1-6 | 10.1 | 10.9 | 0 | 0 | 34.8 |
| 1-7 | 10.3 | 11.1 | 0 | 0 | 28.7 |
| 1-8 | 10.7 | 11.3 | 0 | 0 | 21.9 |
| 1-9 | 11.2 | 11.5 | 0 | 0 | 13.3 |
| 1-10 | 11.5 | 11.7 | 0 | 0 | 6.3 |
| 1-11 | 11.7 | 11.8 | 0 | 0 | 5.9 |
| 1-12 | 11.8 | 11.8 | 0 | 0 | 44.9 |
| 1-13 | 11.9 | 10.5 | 0 | 0 | 39.8 |
| 1-14 | 11.7 | 10.6 | 0 | 0 | 35.9 |
| 1-15 | 11.7 | 10.9 | 0 | 0 | 31.5 |
| 1-16 | 12.0 | 11.0 | 0 | 0 | 26.8 |
| 1-17 | 12.1 | 11.2 | 0 | 0 | 18.9 |
| 1-18 | 12.2 | 11.4 | 0 | 0 | 11.3 |
| 1-19 | 12.3 | 11.8 | 0 | 0 | 5.3 |
| 1-20 | 7.1 | 10.5 | 0 | 0 | 48.7 |
| 1-21 | 7.7 | 10.5 | 0 | 0 | 48.1 |
| 1-22 | 8.0 | 10.5 | 0 | 0 | 44.1 |
| 1-23 | 8.2 | 10.8 | 0 | 0 | 38.1 |
| 1-24 | 8.3 | 11.0 | 0 | 0 | 32.5 |
| 1-25 | 8.4 | 11.1 | 0 | 0 | 23.8 |
| 1-26 | 8.5 | 11.4 | 0 | 0 | 16.1 |
| 1-27 | 12.2 | 10.4 | 0 | 0 | 41.9 |
| 1-28 | 12.4 | 10.6 | 0 | 0 | 37.8 |
| 1-29 | 12.3 | 10.8 | 0 | 0 | 34.3 |
| 1-30 | 12.4 | 11.0 | 0 | 0 | 29.7 |
| 1-31 | 12.6 | 11.2 | 0 | 0 | 24.4 |
| 1-32 | 12.6 | 11.4 | 0 | 0 | 17.8 |
| 1-33 | 12.9 | 11.7 | 0 | 0 | 10.6 |
| 1-34 | 11.4 | 10.5 | 0 | 0 | 43.5 |
| 1-35 | 11.6 | 10.7 | 0 | 0 | 38.5 |
| 1-36 | 11.7 | 10.8 | 0 | 0 | 35.2 |
| 1-37 | 12.0 | 11.0 | 0 | 0 | 30.1 |
| 1-38 | 12.3 | 11.3 | 0 | 0 | 25.6 |
| 1-39 | 12.4 | 11.4 | 0 | 0 | 18.1 |
| 1-40 | 12.6 | 11.7 | 0 | 0 | 11.6 |
| 1-41 | 7.3 | 21.2 | 4.9 | 14.7 | 31.1 |
| 1-42 | 9.1 | 21.0 | 4.5 | 15.7 | 28.9 |
| 1-43 | 10.9 | 20.9 | 4.6 | 16.4 | 27.3 |
| 1-44 | 12.8 | 20.9 | 4.3 | 16.7 | 25.7 |
| 1-45 | 14.6 | 21.0 | 4.2 | 17.2 | 24.1 |
| 1-46 | 16.6 | 21.0 | 4.0 | 17.4 | 22.6 |
| 1-47 | 6.6 | 21.1 | 4.8 | 15.9 | 31.9 |
| 1-48 | 7.7 | 21.4 | 4.7 | 16.7 | 30.9 |
| 1-49 | 9.7 | 21.4 | 4.5 | 17.2 | 28.6 |
| 1-50 | 11.4 | 21.5 | 4.3 | 17.6 | 27.0 |
| 1-51 | 13.5 | 21.6 | 4.2 | 17.9 | 25.3 |
| 1-52 | 15.9 | 21.6 | 4.0 | 18.1 | 23.6 |
| 1-53 | 18.0 | 21.2 | 3.9 | 18.4 | 21.9 |

| | | | | | |
|---|---|---|---|---|---|
| 1-54 | 20.6 | 21.6 | 3.9 | 19.1 | 20.6 |
| 1-55 | 7.2 | 17.5 | 11.8 | 25.5 | 39.8 |
| 1-56 | 8.7 | 17.2 | 11.8 | 27.2 | 37.7 |
| 1-57 | 4.1 | 17.0 | 13.0 | 26.7 | 41.8 |
| 1-58 | 5.6 | 17.1 | 12.8 | 27.4 | 41.4 |
| 1-59 | 7.3 | 17.1 | 12.0 | 28.7 | 38.4 |
| 1-60 | 8.8 | 1.71 | 11.5 | 29.2 | 36.2 |
| 1-61 | 10.1 | 1.71 | 11.1 | 29.9 | 34.3 |
| 1-62 | 11.4 | 16.9 | 11.3 | 31.4 | 33.1 |
| 1-63 | 12.8 | 17.0 | 11.0 | 31.9 | 31.8 |
| 1-64 | 6.1 | 16.5 | 13.4 | 30.6 | 38.9 |
| 1-65 | 7.6 | 16.7 | 13.0 | 31.1 | 36.8 |
| 1-66 | 9.0 | 16.9 | 12.3 | 32.1 | 34.7 |
| 1-67 | 10.4 | 16.9 | 12.0 | 32.3 | 33.3 |
| 1-68 | 11.9 | 16.9 | 11.4 | 32.8 | 32.2 |
| 2-1 | 7.6 | 3.6 | 0 | 0 | 5.5 |
| 2-2 | 7.6 | 3.7 | 0 | 0 | 2.4 |
| 2-3 | 7.6 | 3.7 | 0 | 0 | 1.0 |
| 2-4 | 7.6 | 3.7 | 0 | 0 | 0.6 |
| 2-5 | 8.5 | 3.0 | 17.4 | 9.3 | 3.1 |
| 2-6 | 8.5 | 3.4 | 7.6 | 5.2 | 2.0 |
| 2-7 | 8.5 | 3.5 | 3.2 | 2.7 | 0.9 |

"

and the figure caption of Figure S3 as:

"**Figure S3.** Average MS of HOMs detected by nitrate CIMS in the $1^{st}$-round experiments with $NO_x$, presented with the averaged normalized signals in $4.41\times10^{10}$ molecule $cm^{-3}$ NO + $1.72\times10^{12}$ molecule $cm^{-3}$ $NO_2$ and $1.18\times10^{11}$ molecule $cm^{-3}$ NO + $2.94\times10^{12}$ molecule $cm^{-3}$ $NO_2$ experiments. For comparison, the MSunder the low $NO_x$ experiments is shown in opposite values.
"

*(6) Figures and Tables must be introduced before they appear.*

**R6.** We have moved the introduction of Figure 1 before its first appearance.

We have revised our manuscript (Line 433 – 444) as:

"We take Exp. 1-12 ([OH] = ~$8.47\times10^8$ molecule $cm^{-3}$ and $NO_x$ = 0) and Exp. 2-3 ([OH] = ~$1.64\times10^7$ molecule $cm^{-3}$ and $NO_x$ = 0) as representative examples and compare simulation results with those from the ambient atmosphere, since $NO_x$ in the ambient is believed not to impact relative ratios for $R1 – R3$, $R5$, and $R6$. In the ambient atmosphere, the average [$HO_2$], [OH], and [$RO_2$] were $2.7\times10^8$, $8.0\times10^6$, and $1.4\times10^9$ molecule $cm^{-3}$, respectively, around summertime noon in urban Beijing (Whalley et al. 2021), and (4 – 28)$\times10^8$, (0.8 – 2.4)$\times10^7$, and $1.2\times10^9$ molecule $cm^{-3}$ (modeled) at a suburban site in Yangtze River Delta (Ma et al. 2022). As shown in Figure 1a, for the most important $RO_2$, BPR, the fractions of monomeric termination reactions of $RO_2$ + $RO_2$ ($R1 – R3$), $RO_2$ + $HO_2$ ($R5$), and $RO_2$ + OH ($R6$) were 6.2%, 29.3%, and 64.5%, respectively, in Exp.1-12. In contrast, the fractions were 32.5%, 31.8%, and 35.7%, respectively, in Exp. 2-3, whereas the values were 20.3%, 66.6%, and 13.2%, respectively, for summertime, urban Beijing."

We have also moved Scheme 1 before its first appearance (Line 679 – 684).

*(7) Please explain the disappearance of dimers during NOx experiments, although the monomer signals are higher in this specific experiment.*

**R7.** The dramatic decrease in the abundance of HOM dimers after the introduction of $NO_x$ into the aromatic oxidation system have been reported in several previous studies (Garmash et al., 2020; Wang et al., 2020; Tsiligiannis et al., 2019), which might be a result from the competition of the $NO_x$ channel. Our current understanding on the kinetics and mechanisms of the formation of HOM dimers is still incomplete, which limits us from obtaining an accurate estimation of the accretion reactions for every $RO_2$. We can only deduce that this disappearance came from the significant decrease of the signals of HOM dimers, which were below the detection limit of our instruments, as we discussed in the last reply.

We have revised our manuscript (Line 872 – 875) as:

"in the $NO_x$-present experiments in the 2$^{nd}$-round experiments. Such a dramatic decrease in the abundance of HOM dimers after the introduction of $NO_x$ into the aromatic oxidation system has been reported in several previous studies (Garmash et al., 2020; Wang et al., 2020b; Tsiligiannis et al., 2019)."

*(8) It is apparent that the "tracked changes" version does not contain all the changes. For example, compare the lines 80-100 in this and the previous submission. Please address this issue.*

**R8.** We are very sorry for this mistake. The formulas of BPR and its autoxidation product came with typos in the first version and were corrected by us in the last version.

We have revised our manuscript (Line 76 – 81) as:

"previous studies suggest that the main products of OH-initiated oxidation of alkylbenzenes ($C_xH_{2x-6}$, x=7, 8, or 9), i.e., bicyclic peroxy radicals (BPR, $C_xH_{2x-5}O_5\cdot$, x=7, 8, or 9) (Jenkin et al., 2003), can undergo an autoxidation reaction and form a new peroxy radical, $C_xH_{2x-5}O_7\cdot$ (x=7, 8, or 9) (Wang et al., 2017).The autoxidation of BPR could be fast if it has a favorable structure, as found in a previous study (Wang et al., 2017). On the other hand, the structure of resulting $C_xH_{2x-5}O_7\cdot$ is strongly different from that of BPR,"

(Line 86 – 87) as:

"atoms in stabilized first generation monomer and dimer products, respectively, formed from $C_xH_{2x-5}O_7\cdot$ (Molteni et al., 2018; Wang et al., 2020b; Mentel et al., 2015; Berndt et al., 2018b)."

We added Reactions No. 2 – No. 5 into Table S2 in the last reply, which have been included in the model but not listed in the first version.

We have revised Table S2 as:
"

| No | Reactions | Reaction rate constants/photolysis rate (molecule$^{-1}$ cm$^3$ s$^{-1}$/ s$^{-1}$) |
|---|---|---|
| 1 | $HO_2 + h\nu$ ($\lambda = 254$ nm) = OH + O($^1$D) | $2.63 \times 10^{-19} \times \text{flux}_{254}$ |
| 2 | $O_3 + h\nu$ ($\lambda = 254$ nm) = OH + O($^1$D) | $1.03 \times 10^{-17} \times \text{flux}_{254}$ |
| 3 | $H_2O + O(^1D) = 2OH$ | $1.63 \times 10^{-10} \times e^{60/T}$ |
| 4 | $N_2 + O(^1D) = O(^3P)$ | $2.15 \times 10^{-11} \times e^{110/T}$ |
| 5 | $O_2 + O(^1D) = O(^3P)$ | $3.30 \times 10^{-11} \times e^{55/T}$ |
| 6 | $O_3 + O(^1D) = 2O_2$ | $1.20 \times 10^{-10}$ |

7     $O_3 + O(^1D) = O + O + O_2$        $1.20 \times 10^{-10}$

…"

We deleted the serial number in Table S5 in the last reply, but it cannot be shown in the "track changes" mode of Word.

We have checked the manuscript thoroughly and made sure all other changes have been included in the "tracked changes".

[revised manuscript text omitted]